

# Constraining metamorphic dome exhumation and fault activity through hydrothermal monazite-(Ce)

Christian A. Bergemann[1, 2], Edwin Gnos[2], Alfons Berger[3], Emilie Janots[4], and Martin J. Whitehouse[5]

[1]University of Geneva, Geneva, Switzerland
[2]Natural History Museum of Geneva, Switzerland
[3]University of Bern, Switzerland
[4]ISTerre University of Grenoble, France
[5]Swedish Museum of Natural History, Stockholm, Sweden

**Correspondence:** Christian Bergemann (christian.bergemann@unige.ch)

**Abstract.** Zoned monazite-(Ce) from Alpine fissures/clefts is used to gain new insights into the exhumation history of the Central Alpine Lepontine metamorphic dome, and timing of deformation along the Rhone-Simplon fault zone on the dome's western termination. These hydrothermal monazites-(Ce) directly date deformation and changes in physiochemical conditions through crystallization ages, in contrast to commonly employed cooling-based methods. The 480 SIMS measurement ages
from 20 individual crystals record ages over a time interval between 19 and 5 Ma, with individual grains recording ages over a lifetime of 2 to 7.5 Ma. The age range combined with age distribution and internal crystal structure help to distinguish between areas whose deformational history was dominated by distinct tectonic events or continuous exhumation. The combination of this age data with geometrical considerations and spatial distribution give a more precise exhumation/cooling history for the area. In the east and south of the study region, the units underwent monazite-(Ce) growth at 19-12.5 and 16.5-10.5 Ma, followed
by a central group of monazite-(Ce) ages at 15-10 Ma and the movements and related cleft monazites-(Ce) are youngest at the western border with 13-7 Ma. A last phase around 8-7 Ma is limited to clefts of the Simplon normal fault and related strike slip faults as the Rhone and Rhine-Rhone faults. The large data-set spread over significant metamorphic structures shows that the opening of clefts, fluid flow and monazite-(Ce) stability is direct linked to the geodynamic evolution in space and time.

## 1 Introduction

Metamorphic domes like the Lepontine area of the Central Alps often experienced a complex tectono-metamorphic evolution. In this case an interplay between exhumation and deformation during doming and activity of large fault systems that dominate the western parts of the area. Although much of the (thermo)chronological history of the area is well known, hydrothermal monazite-(Ce) ages complement existing cooling ages of zircon fission track, Rb-Sr in biotite and apatite fission track/apatite U-Th/He by providing crystallization and dissolution-precipitation ages that date low-T tectonic evolution.

Monazite, (LREE,Th,U)PO$_4$, is considered an excellent mineral for dating of geologic processes (*e.g.*, Parrish, 1990) that is highly resistant to radiation damage (*e.g.*, Meldrum et al., 1998, 1999, 2000) and shows negligible Pb loss through diffusion (Cherniak et al., 2004; Cherniak and Pyle, 2008). Nonetheless, monazite remains geologically reactive after crystallization. It



can experience dissolution-recrystallization, thereby recording new ages through mediation of hydrous fluids (*e.g.*, Seydoux-Guillaume et al., 2012; Janots et al., 2012; Grand′Homme et al., 2016).

Alpine fissures and clefts occasionally containing monazite-(Ce) are voids partially filled by crystals that crystallized on the cleft walls from hydrous fluids during late stage Alpine metamorphism (Mullis et al., 1994; Mullis, 1996). Dating such

mineralization is often difficult due to later overprinting along with multiple stages of fluid activity (Purdy and Stalder, 1973). Alpine fissures in some metasediments and metagranitoids have long been known to contain well-developed monazite-(Ce) crystals (Niggli et al., 1940), but it is only recently that some of these could be dated (Gasquet et al., 2010; Janots et al., 2012). Although other minerals like micas and adularia are common in alpine fissures, they are often affected by overpressure/excess argon, (*e.g.*, Purdy and Stalder, 1973), and it is not always clear if these ages represent crystallization or cooling

(*e.g.*, Rauchenstein-Martinek, 2014). The Alpine fissures and clefts in the Lepontine region formed after the metamorphic peak, in relation to extensional tectonic activity. In accordance with this tectonic activity, fissures and clefts are oriented roughly perpendicular to lineation and foliation of the host rock. The fluid that intruded during fissure formation (300-500°C; Mullis et al., 1994; Mullis, 1996) interacts with the wall rock. This triggered dissolution and precipitation of minerals in both host rock and fissure, causing the formation of a porous alteration halo in the surrounding wall rock. Complex growth domains are common

in hydrothermal monazites-(Ce) from such fissures showing both, dissolution and secondary growth (*e.g.*, Janots et al., 2012) as well as dissolution-reprecipitation reactions resulting in patchy grains (*e.g.*, Gnos et al., 2015). In contrast to metamorphic rocks, where monazite-(Ce) rarely exceeds 100 $\mu$m, cleft monazite-(Ce) is commonly mm-sized with large individual growth domains. This permits to date individual domains precisely by using SIMS (secondary ion mass spectrometry) and even resolve growth duration (Janots et al., 2012; Berger et al., 2013; Bergemann et al., 2017, 2018, 2019).

## 2 Geological setting

### 2.1 Evolution of the study area

The formation of the nappe stack of the Alps caused by the collision of the European and Adriatic plates was followed by the development of several metamorphic domes (Tauern and Rechnitz in Austria, and Lepontine in the western Alps; e.g. Schmid et al., 2004). Their formation was related to crustal shortening associated with coeval orogen parallel extension (*e.g.*,

Mancktelow, 1992; Ratschbacher et al., 1991; Ratschbacher et al., 1989; Steck and Hunziker, 1994). The Western and Central Alps with the Lepontine metamorphic dome have consequently had a complex tectonic and metamorphic history.

Early high-pressure metamorphism in the Western Alpine Sesia-Lanzo Zone during subduction below the Southern Alps is dated at 75-65 Ma (*e.g.* Ruffet et al., 1997; Duchêne et al., 1997; Rubatto et al., 1998). This was followed by underthrusting and nappe stacking from ca. 42 Ma on during continental collision linked with a transition from high-P to high-T metamorphism

(e.g. Köppel and Grünenfelder, 1975; Markley *et al.*, 1998; Herwartz et al., 2011; Boston et al., 2017). Peak metamorphic conditions in the Lepontine area in excess of 650°C in some regions were reached diachronously from south to north in time around 30-19 Ma and accompanied by limited magmatic activity from 33 Ma down to ca. 22 Ma (von Blanckenburg et al., 1991; Romer et al., 1996; Schärer et al., 1996; Oberli et al., 2004; Rubatto et al. 2009; Janots et al., 2009). Prograde metamorphism




was followed by staggered exhumation in the Ticino and Toce culminations of the Lepontine dome. Accelerated cooling below 500°C occurred at 26 Ma first in the central Lepontine (Hurford, 1986). This was followed in the east by a period of rapid cooling of the Ticino dome between 22 and 17 Ma (Steck and Hunziker, 1994; Rubatto et al., 2009) after which exhumation slowed down. To the west, the Toce dome experienced phases of accelerated cooling somewhat later in the time of 18-15 Ma and 12-10 Ma (Campani et al., 2014). The later cooling phase was related to detachment along the Rhone-Simplon Fault (Steck and Hunziker, 1994; Campani et al., 2014).

While most of the Lepontine area is marked by doming and associated deformation events, the western and southwestern limits of the study area are dominated by the Rhone-Simplon Fault system, its extensions to the Rhine-Rhone Line to the north along the Aar massif and the Centovalli Fault to the south. The extensional Simplon Fault zone (SFZ) was already active during thrusting in the external alpine domain (*e.g.*, Grosjean et al., 2004), with transpressional movements in the hanging wall of the dextral ductile Simplon shear zone occurring from ca. 32 Ma on (Steck, 2008). The ductile-brittle transition of the SFZ was constrained to the time between 14.5 and 10 Ma (Campani et al., 2010). Brittle deformation of the SFZ and Centovalli fault continued after this (Zwingmann and Mancktelow, 2004; Surace et al., 2011), with the youngest displacement activity dated to ca. 5-3 Ma (Campani et al., 2010).

## 2.2 The study Area

The study area comprises roughly half of the Lepontine metamorphic dome (Fig. 2), from the Tambo nappe, east of the Forcola fault, over the central Lepontine dome to the Val d'Ossola, south of the Centovalli Fault, and the southern Gotthard nappe and Aar massif to the north. See Fig. 1 for the tectonic position of the samples. The total number of 20 monazite-(Ce) samples dated in this study and 6 samples described in the literature (Janots et al., 2012; Berger et al., 2013; Bergemann et al., 2017) were divided into four groups roughly correlating to tectonic subdivisions of the area (Fig. 2). These are (1) the area to the east of the Forcola Fault (East; 2 samples), (2) the central Ticino dome and southern Gotthard nappe (Center; 7 samples), (3) the Toce dome, bounded by the Rhone-Simplon Fault to the west and adjacent south-western Gotthard nappe and parts of the Aar massif (West; 10 samples), and (4) the area to the south of the Centovalli and southern Simplon faults (South; 1 sample). Most of the samples were provided by mineral collectors, as hydrothermal cleft monazite-(Ce) is uncommon and often difficult to detect in the field when covered by dirt or chlorite. See Table 1 for location details.

## 3 Analytical techniques

Monazites-(Ce) were individually polished to the level of a central cross section and assembled in mounts of several grains. Backscatter electron (BSE) images were then obtained. Secondary ion mass spectrometry (SIMS) spot analyses (Fig. 4) were placed according to compositional domains visible in these images in order to capture the crystallization history. As far as possible, the placement of measurement spots located near cracks or holes was avoided, as the Th-Pb isotope system may be disturbed in these areas (Janots et al., 2012; Berger et al., 2013).



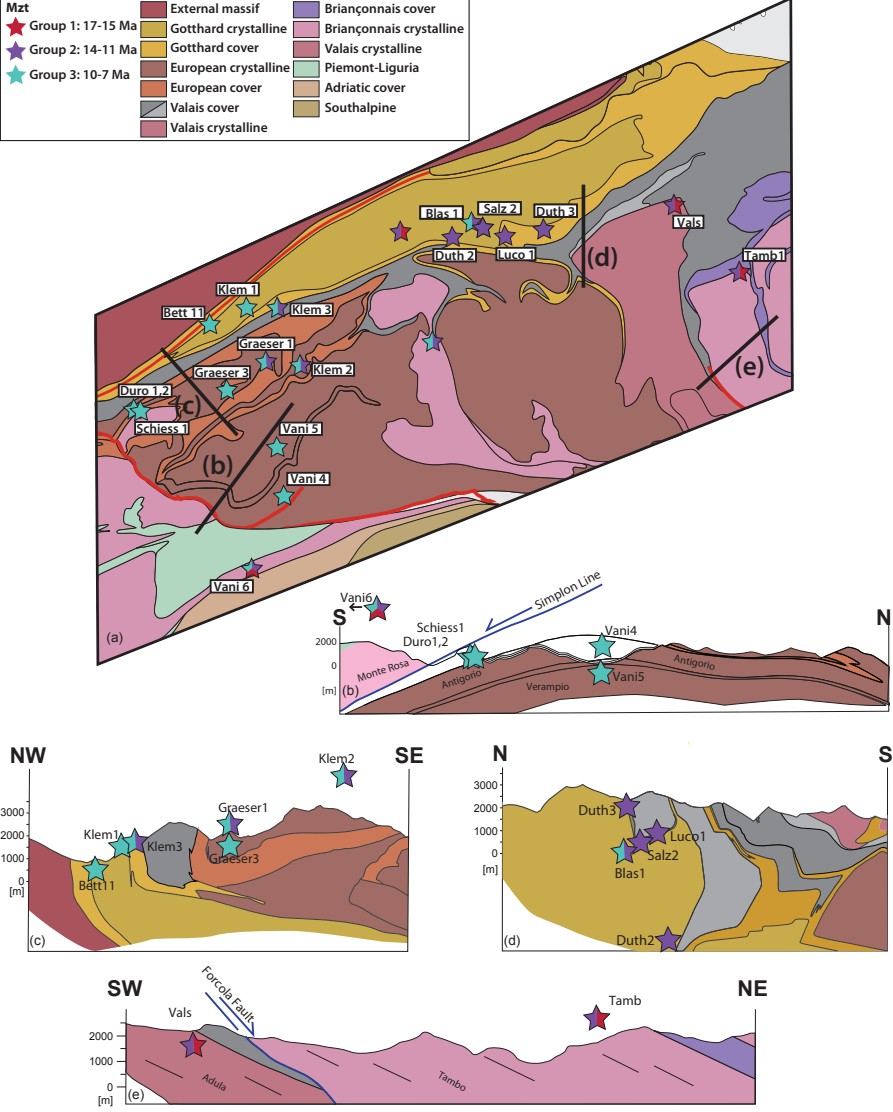

**Figure 1.** Geological-geometric situation of the study area. (a) Tectonic sketch map modified after Steck et al. (2013) and Schmid et al. (2004); (b) Tectonic section over the Simplon Fault zone into the western Lepontine, based on Campani et al. (2014); (c) Tectonic section through the western Northern Steep Belt, modified and extended after Leu (1986); (d) Tectonic section through the eastern Northern Steep Belt, redrawn after Wiederkehr et al. (2008); (e) Sketch of the situation at the Forcola normal fault, see also Meyre et al., (1998) and Berger et al. (2005).

Th-Pb analyses were conducted at the Swedish Museum of Natural History (NordSIM facility) on a Cameca IMS1280 SIMS instrument. Analytical methods and correction procedures followed those described by Harrison et al. (1995), Kirkland et al. (2009) and Janots et al. (2012), using a -13kV $O^{2-}$ primary beam of ca. 6nA and nominal 15$\mu$m diameter. The mass

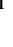
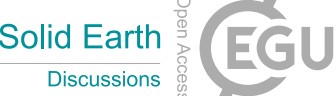

**Table 1.** Information on sample localities for all analyzed grains. Sample GRAESER1 has identification number NMBa 10226, VALS has NMBE43124.

| Sample | Locality | Latitude | Longitude | Altitude (m) |
|---|---|---|---|---|
| BETT11 | Bettelbach, Niederwald, Goms | 46°25.62' | 8°11.70' | 1460 |
| BLAS1 | Piz Blas, Val Nalps, Sedrun | 46°34.68' | 8°43.98' | 2790 |
| DURO1 | Doru, Gantertal, Simplon | 46°17.63' | 8°02.07' | 1160 |
| DURO2 | Doru, Gantertal, Simplon | 46°17.64' | 8°02.07' | 1160 |
| DUTH2 | Lago Sucro, Val Cadlimo | 46°33.80' | 8°41.50' | 2620 |
| DUTH3 | Lago Retica, Lagi di Campo Blenio | 46°34.45' | 8°53.57' | 2400 |
| DUTH6 | Pizzo Rüscada, Valle di Prato (Lavizzara) | 46°24.57' | 8°40.09' | 2420 |
| GRAESER1 | Lärcheltini, Binntal | ~46°22.25' | ~8°14.89' | ~1860 |
| GRAESER3 | Wannigletscher, Cherbadung, Binntal | ~46°19.46' | ~8°12.98' | ~2720 |
| KLEM1 | Grosses Arsch, Blinnental | 46°26.71' | 8°16.33' | ~1900 |
| KLEM2 | Alpe Devero, Val Antigorio | 46°22.16' | 8°18.44' | 2340 |
| KLEM3 | Griessgletscher | 46°26.59' | 8°19.46' | 2840 |
| LUCO1 | Lucomagno | 46°33.79' | 8°48.10' | 1915 |
| SALZ2 | Piz Scai | ~46°34.5' | ~8°45.75' | ~2740 |
| SCHIESS1 | Schiessbach/Simplon | 46°18.13' | 8°04.18' | 1760 |
| TAMB1 | Pizzo Tambo, Splügen | 46°30.48' | 9°18.35' | 2460 |
| VALS | Vals, Valsertal | ~46°37.33' | ~9°17.28' | ~3150 |
| VANI4 | Montecrstese | 46°09.60' | 8°19.18' | 370 |
| VANI5 | Crino Baceno | 46°15.13' | 8°19.14' | 710 |
| VANI6 | Cava Maddalena, Beura | 46°04.30' | 8°17.71' | 260 |

spectrometer was operated at +10kV and a mass resolution of ca. 4300 (M/ΔM, at 10% peak height), with data collected in peak hopping mode using an ion-counting electron multiplier. Unknowns were calibrated against monazite-(Ce) standard



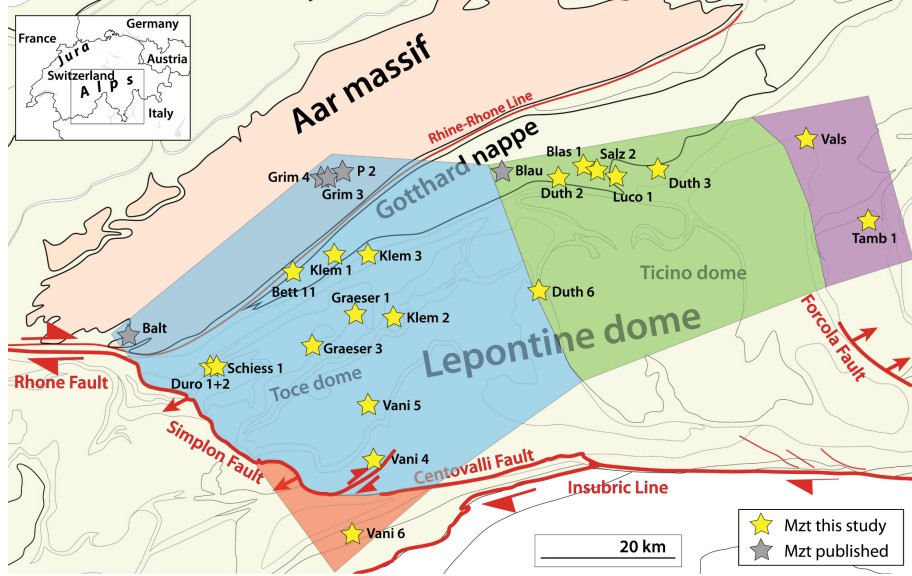

**Figure 2.** Map of the Lepontine metamorphic dome, modified from Steck *et al.* (2013) and Schmid *et al.* (2004). Colored areas mark areal division in the context of this study. Published monazite-(Ce) locations (grey stars) are from Janots *et al.* (2012), Berger *et al.* (2013) and Bergemann *et al.* (2017).

44069 (Aleinikoff et al., 2006). Lead isotope signals were corrected for common Pb contribution using measured $^{204}$Pb and an assumed present-day Pb isotope composition according to the model of Stacey and Kramers (1975). The measurement of $^{204}$Pb is subject to an unresolvable molecular interference by $^{232}$Th$^{143}$Nd$^{16}$O$_2^{++}$ (also affecting $^{206}$Pb and $^{207}$Pb to a lesser degree through replacement of $^{16}$O with heavier O-isotopes), which may result in an overestimation of common Pb concentrations. A

5   correction was applied whenever the $^{232}$Th$^{143}$Nd$^{16}$O$_2^{++}$ signal at mass 203.5 exceeded the average background signal on the ion-counting detector by three times its standard deviation. Age calculations use the decay constants recommended by Steiger and Jäger (1977). Th-Pb ages presented were corrected for common Pb and doubly charged $^{232}$Th$^{143}$Nd$^{16}$O$_2^{++}$ overlap and are given at $2\sigma$ uncertainties.

## 4   Results

10   The complete ion-probe data set is given in the data Supplement Table 1 (PANGEA, doi: still pending), see Tab. 2 for an overview and Figs. 4 and 7 for measurement positions and a graphical representation. As there are difficulties with the U-Pb system for hydrothermal monazite-(Ce) (Janots et al., 2012), only $^{208}$Pb/$^{232}$Th ages were used. For explanations on age patterns across the grains, grouping and weighted mean age determination, see the discussion in Chapter 5.2.



**Figure 3.** Back-scatter electron images of studied cleft monazite-(Ce). The zonation corresponds largely to variations in Th contents. Spots refer to SIMS analysis spots, with colours indicating chemical and age domains. The color of the frame indicates data for which it was possible to calculate weighted mean $^{208}Pb/^{232}Th$ ages.



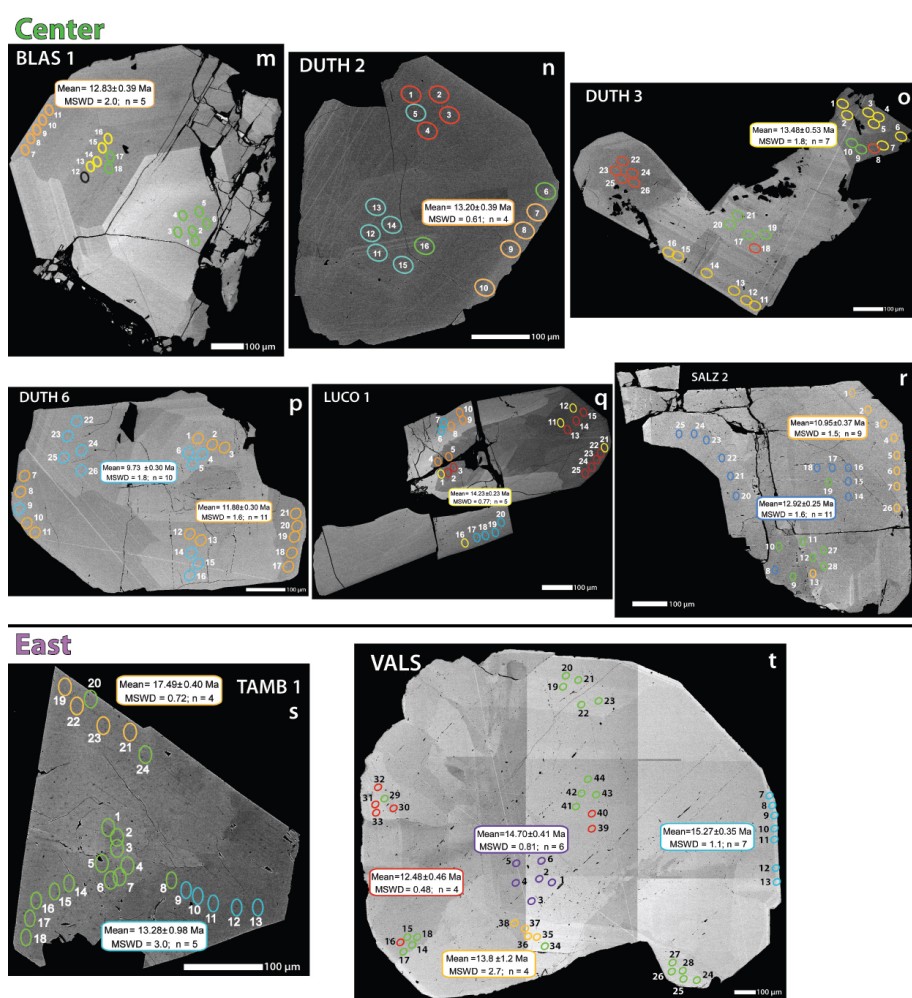

**Figure 4.** Back-scatter electron images of studied cleft monazite-(Ce). The zonation corresponds largely to variations in Th contents. Spots refer to SIMS analysis spots, with colours indicating chemical and age domains. The color of the frame indicates data for which it was possible to calculate weighted mean $^{208}$Pb/$^{232}$Th ages.





**Table 2.** Measurement spots AIGG1 1, 9, 11, BLANC2 7, 8, 23 and SALZ15 4 were excluded due to their location on cracks or signs of mineral inclusions.

| Sample | Figure | Weighted mean domain ages (Ma) | MSWD | Number of points | Sample spot age range (Ma) |
|---|---|---|---|---|---|
| VANI6 | 3a, 4a | $14.68 \pm 0.47$ | 2.8 | 5 | $16.80 \pm 0.31 - 10.62 \pm 0.18$ |
| BETT11 | 3b, 4c | $9.96 \pm 0.30$ | 1.5 | 9 | $10.55 \pm 0.33 - 7.34 \pm 0.26$ |
| | | $7.53 \pm 0.31$ | 0.53 | 3 | |
| DURO1 | 3c, 4c | $9.96 \pm 0.18$ | 0.30 | 7 | $10.82 \pm 0.26 - 8.21 \pm 0.20$ |
| | | $8.34 \pm 0.20$ | 0.29 | 4 | |
| DURO2 | 3d, 4d | $7.63 \pm 0.13$ | 0.55 | 8 | $11.48 \pm 0.28 - 7.02 \pm 0.18$ |
| | | $7.18 \pm 0.18$ | 0.50 | 4 | |
| GRAESER1 | 3e, 4e | $9.03 \pm 0.19$ | 0.28 | 5 | $12.14 \pm 0.30 - 7.57 \pm 0.19$ |
| | | $7.91 \pm 0.26$ | 1.7 | 7 | |
| GRAESER3 | 3f, 4f | | | | $15.60 \pm 0.61 - 6.36 \pm 0.39$ |
| KLEM1 | 3g, 4g | $8.43 \pm 0.20$ | 0.94 | 5 | $10.64 \pm 0.26 - 7.97 \pm 0.20$ |
| KLEM2 | 3h, 4h | $13.44 \pm 0.31$ | 0.57 | 5 | $13.65 \pm 0.33 - 9.47 \pm 0.40$ |
| | | $11.74 \pm 0.32$ | 0.83 | 5 | |
| | | $10.12 \pm 0.94$ | 2.6 | 4 | |
| KLEM3 | 3i, 4i | $12.64 \pm 0.38$ | 0.18 | 5 | $12.96 \pm 0.46 - 8.43 \pm 0.32$ |
| | | $11.99 \pm 0.66$ | 2.3 | 6 | |
| SCHIESS1 | 3j, 4j | $9.69 \pm 0.60$ | 2.4 | 4 | $9.94 \pm 0.25 - 6.78 \pm 0.18$ |
| VANI4 | 3k, 4k | $8.03 \pm 0.53$ | 3.3 | 9 | $9.27 \pm 0.43 - 6.89 \pm 0.37$ |
| | | $8.03 \pm 0.44$ | 2.2 | 7 | |
| VANI5 | 3l, 4l | $7.21 \pm 0.43$ | 1.9 | 6 | $8.07 \pm 0.36 - 4.86 \pm 0.24$ |
| | | $5.53 \pm 0.60$ | 3.5 | 5 | |
| BLAS1 | 3m, 4m | $12.83 \pm 0.39$ | 2.0 | 5 | $14.49 \pm 0.26 - 7.82 \pm 0.22$ |
| DUTH2 | 3n, 4n | $13.20 \pm 0.39$ | 0.61 | 4 | $14.34 \pm 0.41 - 11.15 \pm 0.43$ |
| DUTH3 | 3o, 4o | $13.48 \pm 0.53$ | 1.8 | 7 | $14.53 \pm 0.43 - 10.61 \pm 0.34$ |
| DUTH6 | 3p, 4p | $11.88 \pm 0.30$ | 1.6 | 11 | $12.60 \pm 0.37 - 9.33 \pm 0.32$ |
| | | $9.73 \pm 0.30$ | 1.8 | 10 | |
| LUCO1 | 3q, 4q | $14.23 \pm 0.23$ | 0.77 | 5 | $14.74 \pm 0.30 - 9.90 \pm 0.17$ |
| SALZ2 | 3r, 4r | $12.92 \pm 0.25$ | 1.6 | 11 | $14.28 \pm 0.74 - 10.51 \pm 0.39$ |
| | | $10.95 \pm 0.37$ | 1.5 | 9 | |
| TAMB1 | 3s, 4s | $17.49 \pm 0.40$ | 0.72 | 4 | $19.02 \pm 0.47 - 8.32 \pm 0.11$ |
| | | $13.28 \pm 0.98$ | 3.0 | 5 | |
| VALS | 3t, 4t | $15.27 \pm 0.35$ | 1.1 | 7 | $16.43 \pm 0.61 - 12.09 \pm 0.57$ |
| | | $14.70 \pm 0.41$ | 0.81 | 6 | |
| | | $12.48 \pm 0.46$ | 0.48 | 4 | |



**Figure 5.** Diagrams showing $^{208}$Pb/$^{232}$Th ages for all samples. The colours indicate chemical domains with weighted mean $^{208}$Pb/$^{232}$Th ages given where applicable.





**Figure 6.** Diagrams showing $^{208}$Pb/$^{232}$Th ages for all samples. The colours indicate chemical domains with weighted mean $^{208}$Pb/$^{232}$Th ages given where applicable.





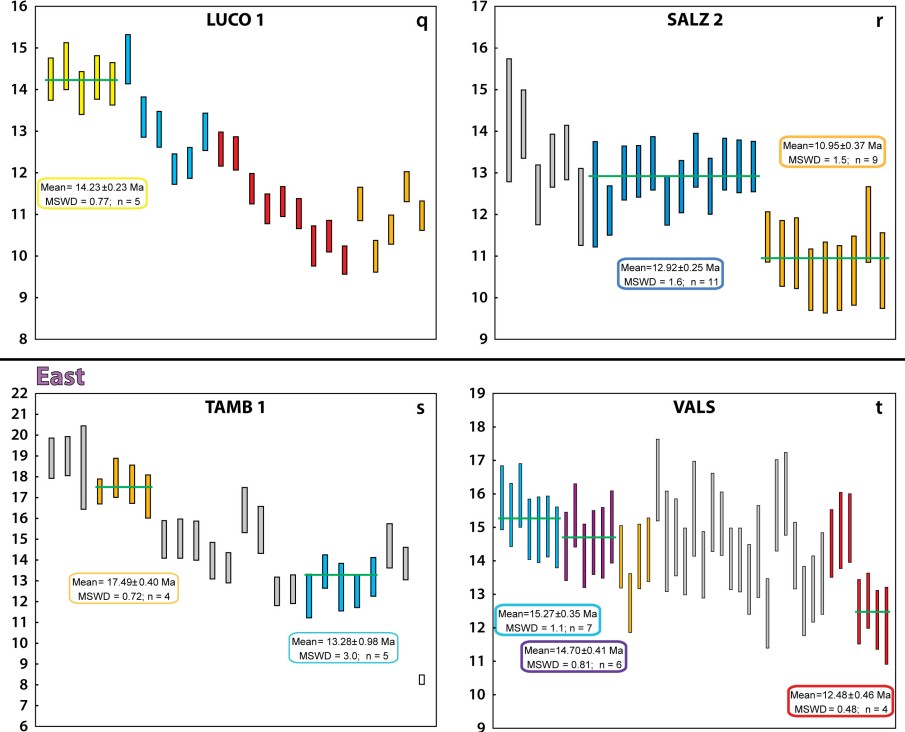

**Figure 7.** Diagrams showing $^{208}$Pb/$^{232}$Th ages for all samples. The colours indicate chemical domains with weighted mean $^{208}$Pb/$^{232}$Th ages given where applicable.

# 5 Discussion

## 5.1 Hydrothermal monazite-(Ce) crystallization

Hydrothermal fissure monazite-(Ce) typically crystallizes at temperatures below 350°C (Gnos et al., 2015; Bergemann et al., 2017, 2018) down to somewhere in the range of 200°C or slightly below (*e.g.* Townsend et al., 2000). Crystallization and

5   later reactions occur when the fissure fluid is brought into disequilibrium. This may be caused by tectonic events for a number of reasons: by volume changes due to deformation, partial collapse of the fissure walls bringing the fluid into contact with unaltered wallrock or the influx of new fluid.

After crystallization, monazite-(Ce) shows practically no U-Th-Pb diffusion (Cherniak and Pyle, 2008). However, replacement mechanisms that may be active in a hydrothermal environment may cause (re-)crystallization and possibly new growth

10   around an existing grain or dissolution-reprecipitation. From the fluid film, a secondary monazite-(Ce) phase precipitates at the surface of the primary phase. The self-sustaining reaction front propagates into the mineral for as long as the interfacial fluid retains a connection to a fluid reservoir. This dissolution-reprecipitation process may be initiated on any part of the crystal





in contact with the surrounding fluid. It is therefore not limited to grain rims, but commonly occurs along mineral inclusion interfaces, cracks and microcracks, that may be invisible in BSE images (Grand'Homme et al., 2018).

These processes may be active as long as conditions in the cleft stay within the monazite-(Ce) stability field. Therefore, several (re-)crystallization or dissolution-precipitation cycles may occur over the active lifespan of a monazite-(Ce) crystal. Later

reactions may be aided by secondary porosity and fracturing induced by the previous dissolution-reprecipitation/recrystallization events, by bringing an increased crystal volume into direct contact with the fluid.

### 5.2   Monazite-(Ce) Th-Pb single and weighted mean ages

As detailed above, SIMS spot analyses were placed across the samples according to growth domains visible in BSE images (Fig. 4). The derived spot ages were grouped together on the basis of chemical composition thought to represent crystallization

under homogeneous chemical conditions, and spatial distribution across the sample according to zonation visible on BSE images to calculate, whenever possible, weighted mean domain ages (Fig. 7). It appears that dissolution-precipitation may largely preserve the chemical composition of an affected crystal part, this would mean that areas with different chemical compositions may have reprecipitated simultaneously. Despite this, spots of different chemical groups were only in a few, clear cases grouped together for weighted mean age calculation. This is to avoid the risk of mistaking multiple mixing ages of

different chemical domains as a distinct event. In areas that experienced few and discrete tectonic events, this approach allows the calculation of domain ages for most analyzed spots of the dataset of a sample (*e.g.* Janots et al., 2012; Bergemann et al., 2017). However, large parts of the study area experienced more than two distinct deformation events and/or phases of prolonged activity. New growth on an existing crystal results in sharp boundaries between zones. But dissolution-reprecipitation processes may lead to irregularly shaped altered zones within a crystal, which may or may not be visible on a BSE image. If this happens

multiple times the limited number of analyses per grain will result in many individual ages being discarded. Meaning that events may not be recognized when looking only at the weighted mean ages. To avoid this, the entire dataset of each region was additionally plotted according to the number of ages per 0.5 Ma intervals to identify age clusters (Fig. 1, appendix). In the next step the peaks or plateaus of the age histogram were plotted according to their relative intensity. They were then combined with the weighted average ages (this study; Janots et al., 2012; Berger et al., 2013; Bergemann et al., 2017) to visualize distinct

events or phases of tectonic activity (Fig. 8). As only a limited number of analyses are possible to obtain for each grain, some weighted mean ages combine only a small number of individual ages. This is especially true for ages dating multiple late stage events that presumably happened at relatively low temperatures. In such cases only those weighted mean ages were kept whose geologic significance is also indicated by other dating techniques such as fault gouge dating, specifically close to the Rhone-Simplon line. Otherwise, these ages are included in the overall age range of the sample in question given in Tab. 2.

Another reason for a spread out age pattern may be a grain experiencing prolonged phases of low-intensity tectonic activity of multiple small deformation events during exhumation. In which case only small volumes of monazite-(Ce) would reprecipitate due to disequilibration during deformation. This leads tendentially to unclear crystal zonations that make it difficult to correctly





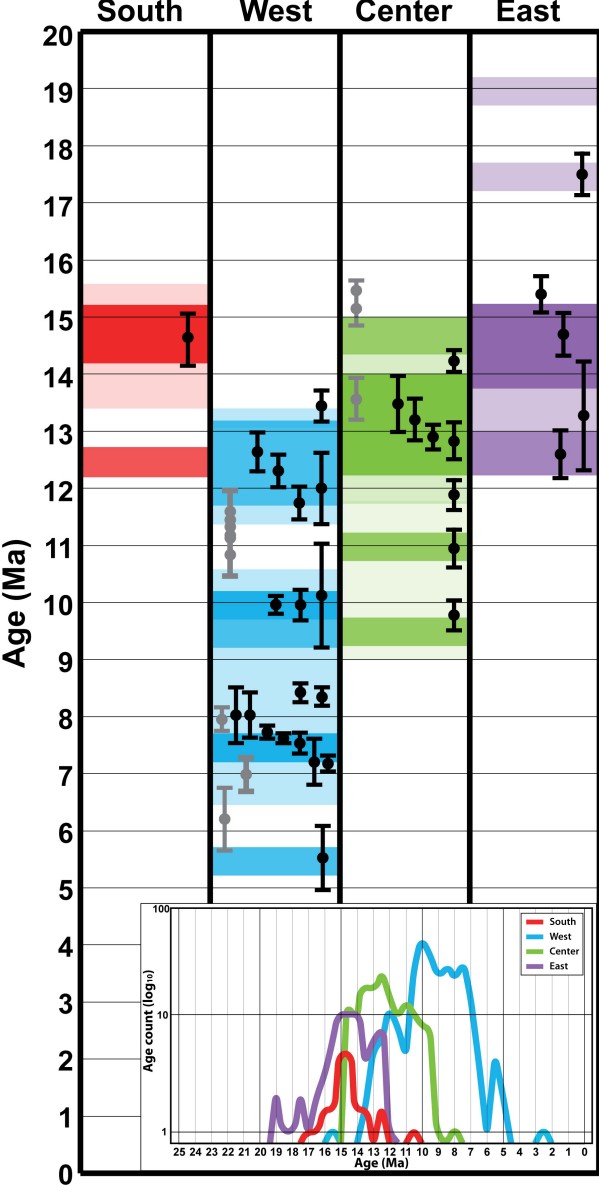

**Figure 8.** Time diagram combining identified peaks and weighted mean ages derived from the data displayed in Fig. 1, appendix, and weighted mean average ages. The color intensity indicates the amount of spot ages in this range. Lighter shades indicate fewer ages. Black error bars indicate weighted mean ages from this study, while grey bars indicate data from Janots et al. (2012), Berger et al. (2013) and Bergemann et al. (2017). The inset shows an age histogram representing the complete dataset of each region according to the number of ages per 0.5 Ma intervals.



identify growth zones on BSE images (compare Gnos et al., 2015; Bergemann et al., 2018). As opposed to areas where crystals record individual, stronger deformation events that tend to show a sharper zonation (compare Janots et al., 2012; Berger et al., 2013; Bergemann et al., 2017, 2019).

### 5.3 Monazite-(Ce) ages and Lepontine history

Hydrothermal cleft monazite-(Ce) crystallization and dissolution-reprecipitation occurred over time in different parts of the study region, as it passed through the monazite-(Ce) stability field. The time interval recorded within individual monazite-(Ce) crystals spans from 2.5 Ma to 7 Ma for individual grains (Fig. 7, Table **??**). The recorded time interval within individual grains is generally longer in the South and East regions of the study area (Fig. 2). The total age range covers the time from ca. 19 to 5 Ma. The monazite-(Ce) chronologic record can be seen to start in the eastern- and southernmost regions (Fig. 9). The recorded

activity then moves to the northeastern and central to the western area. Younger ages in the west progressively concentrate on the large fault systems of the Rhone-Simplon Fault and the proposed location of the Rhine-Rhone Line. The oldest recorded monazite-(Ce) ages of 19-17 Ma from the eastern edge of the study area (Figs. 7s, 9a) coincide with a phase of rapid exhumation and cooling between 22 and 17 Ma (Steck and Hunziker, 1994; Rubatto et al., 2009). At that time, temperatures in parts of the north-western area (northern Ticino Dome) were still prograde at 450-430 °C 19-18 Ma (Janots et al., 2009) as deduced

from allanite dating. After this, temperatures must have decreased to lower temperatures during exhumation, as hydrothermal monazite-(Ce) crystallization in the north(east)ern area started at around 16 Ma in the Valsertal (sample VALS, Fig. 7t) and then at the southern edge of the Gotthard nappe at 14-15 Ma (Fig. 9b). This may indicate crystallization during a deformation phase indicated by 17-14 Ma $^{40}$Ar/$^{39}$Ar biotite ages interpreted as dating recrystallization (Wiederkehr et al., 2009).

The monazite-(Ce) age record for the entire (north)eastern region continues until ca. 13 Ma after which the record ends for

the Valsertal where cooling below 180°C is dated at around 12 Ma (zircon U/Th-He; Price et al., 2018). The age range of the Valsertal sample of 16-12 Ma perfectly coincides with hydrothermal cleft monazite-(Ce) ages from within the Gotthard nappe of ca. 16-12 Ma (Janots et al., 2012; Ricchi et al., in review), after which monazite activity moved into the Lepontine dome south of the Gotthard nappe. Locally within in the dome, in the northern part of the western region, zircon fission track (ZFT) ages of 10-9 Ma in the border area of Ticino dome and Gotthard nappe (Janots et al., 2009) are equal to the last widely recorded

hydrothermal monazite-(Ce) ages of around 10 Ma. One sample records ages of 9-8 Ma (BLAS1; Fig. 4m) that is in agreement with K/Ar fault gouge data of 8.9 ± 0.2 to 7.9 ± 02 Ma close to BLAS1 and SALZ2 (Alp Transit tunnel; Zwingmann et al., 2010), as fault gouge ages seem to typically coincide with the end of monazite-(Ce) growth (see below; Bergemann et al., 2017).

While $^{40}$Ar/$^{39}$Ar cleft muscovite ages of 15.60 ± 0.30 to 14.71 ± 0.13 Ma (Rauchenstein-Martinek, 2014) slightly south

of LUCO1 coincide with the earliest monazite-(Ce) crystallization, this differs markedly from the situation further west or in the Aar and Mont Blanc massifs (Bergemann et al., 2017; 2019). There, as discussed below, ZFT ages predate or mirror primary monazite-(Ce) crystallization and are in turn predated by $^{40}$Ar/$^{39}$Ar white mica ages. The coincidence of these ZFT and $^{40}$Ar/$^{39}$Ar muscovite cooling ages with the hydrothermal monazite-(Ce) crystallization suggests slow cooling rates during



**Figure 9.** Overview maps of the study area showing the distribution of the monazite-(Ce) age record over time. Weighted mean average ages are given near the stars representing the corresponding sample locations. Note the shift over time from the outer regions of the Lepontine dome to the internal areas and then to the shear zones bounding its western limit.



continued deformation (Bergemann et al., 2018) for the time from around 15 Ma until ca. 9 Ma, as the systems closed only at the lower end of the closure temperature window in this case.

To the west, an early phase of accelerated cooling in the area was dated to 18-15 Ma (Steck and Hunziker, 1994; Campani et al., 2010), evidence of which is also preserved in the oldest monazite-(Ce) data of 17 Ma (Fig. 7a) from south of the Rhone-

Simplon Fault (RSF). Zircon fission track ages of 14-11 Ma (Hurford, 1986) and cleft adularia ages between $12.92 \pm 0.17$ Ma and $10.82 \pm 0.12$ Ma (Rauchenstein-Martinek, 2014) from south of the western Gotthard slightly predate to coincide with a later phase cooling and increased tectonic activity in the western Lepontine area. Primary monazite-(Ce) crystallization in parts of the northwestern and central Lepontine as well as the central Aar massif occurs at around 12 Ma, followed by monazite-(Ce) crystallization in the westernmost area around 11-10 Ma dating exhumation (Fig. 9 c,d). Multiple monazite-(Ce) samples

from locations in the Gotthard nappe and Aar-massif yield weighted average ages of 10 Ma. These age patterns are related to processes during backfolding of the northern steep belt (in the sense of Milnes 1974), dating it to ca. 10 Ma in this area (Steck, 1984; Steck and Hunziker, 1994; Campani et al., 2014).

The 12-10 Ma cooling phase of the western Lepontine was related to detachment movements along the Rhone-Simplon Fault. This time interval marks the end of the hydrothermal monazite-(Ce) age record in the hanging wall of the Rhone-

Simplon Fault. Correspondingly, 12-10 Ma also marks the beginning of monazite-(Ce) crystallization to the east of the fault, first in the vicinity of the Aar massif (Figs. 7 c, d, j) and then also further south (Fig. 7k). Primary monazite-(Ce) crystallization ages along the eastern side of the RSF are tendentially predating, but still in close agreement with zircon fission track ages in this area. In the case of sample VANI6 from south of the RSF (Fig. 7a) ZFT ages of this area show a scatter from 12 to 7 Ma (Keller et al., 2005) that overlap with the youngest monazite-(Ce) age spots. Monazite-(Ce) ages of 9-7.5 Ma indicate continued

exhumation of the western region and the central areas leave the hydrothermal monazite-(Ce) stability field at this time (Fig. 9f). The number of weighted mean ages (i.e. clear age patterns within the crystals) staggered over a relatively short time (Fig. 8), suggest deformation pulses during brittle tectonics along the Rhone-Simplon/Centovalli Faults and corroborates evidence of continued deformation along the southern RSF and the Centovalli Fault (Zwingmann and Mancktelow, 2004; Surace et al., 2011). The youngest widely recorded monazite-(Ce) age group for the western Lepontine dates to around 7 Ma (Figs. 7 b, d, j-l;

9f). This coincides with young fault gouge data of 8-6 in this region (Zwingmann and Mancktelow, 2004, Surace et al., 2011) Overall, the 10-7 Ma time interval is characterized by phases of strike-slip deformation along the extended Rhone-Simplon fault system. This is recorded through hydrothermal monazite-(Ce) and fault gouge illite crystallization that was not restricted to the south-western Lepontine but also recorded in faults bounding the Mont Blanc massif (Bergemann et al., 2019). The ages of 8-7 Ma of the sample with the youngest recorded age (VANI5) among the studied monazites-(Ce) are concurrent with the

youngest recorded ages of all other samples along the Rhone-Simplon fault system. The sample comes from an area where hydrothermal gold mineralization occurred and the youngest age group of VANI5 give a weighted mean age of $5.53 \pm 0.60$ Ma that coincides with ZFT ages of 6.4-5.5 Ma (Keller et al., 2005). The area also has a muscovite $^{40}\text{Ar}/^{39}\text{Ar}$ age of 10.56 $\pm 0.31$ Ma (Pettke et al., 1999) that postdates other white mica ages of the area by 4-5 Ma (see summary in Campani et al., 2010), similar to the difference between the youngest recorded ages for monazite-(Ce) samples from the same area.



# 6 Conclusions

Hydrothermal fissure monazite-(Ce) always dates crystallization and not cooling due to system closure and often shows complex recrystallization features. It provides an important record of the shifting tectonic activity associated with the regions exhumation history within the monazite stability field. A comparison between hydrothermal monazite-(Ce) samples from different parts of the Lepontine metamorphic dome shows that age clusters within individual crystals from a simply exhuming area have a less clear age distribution than samples from fault zone areas, or fast exhuming areas. Monazite-(Ce) (re)crystallization/ dissolution-reprecipitation during exhumation is in these areas connected to repeated tectonic activity of small intensity, while distinct events or short periods of intense tectonic activity of fault zones appear to result in larger, more homogenous crystal zones that are easier to date.

The $^{232}$Th-$^{208}$Pb monazite-(Ce) crystallization data records prolonged hydrothermal activity between 19 and 5 Ma contribute to the understanding of the tectonic evolution of the Central Alps in a temperature range of ca. 350-200°C. The oldest ages of 19-17 Ma come from the eastern- and southernmost regions of the study area (Fig. 1), in the hanging wall of the Forcola and Rhone-Simplon faults defining the borders of the metamorphic dome. Within the dome, monazite-(Ce) crystallization started in the northern Ticino dome and eastern Gotthard nappe around 15 Ma and show signs of slow exhumation. Further west, in the Toce dome, primary crystallization occurred in the western Gotthard nappe and the central Aar massif at 12-10 Ma. Younger ages of 9-7 Ma in the west of the study area record the progressive concentration of tectonic activity along the large fault systems of the Rhone-Simplon Fault and the Rhine-Rhone Line.

*Competing interests.* No competing interests are present.

*Acknowledgements.* This work was funded by the Swiss National foundation, projects 200021-143972 and 200020-165513. We greatly appreciate the help of M. Andres, R. Duthaler, M. Flepp, S. Graeser, L. Klemm, B. Hofmann, A. Salzmann, F. Vanini, and M. Walter in organizing monazite-(Ce) samples for this study.

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
