# Peer review of "Constraining metamorphic dome exhumation and fault activity through hydrothermal monazite-(Ce)"

_Solid Earth, 2019_

## Referee Comment (RC2)

Review of Bergemann, Gnos, Berger, Janots and Whitehouse, Solid Earth Ms. Feb 2019: Constraining metamorphic dome exhumation and fault activity through hydrothermal monazite-(Ce).
Reviewer: Fraukje Brouwer, VU Amsterdam, Netherlands, 11 March 2019

**General assessment**
The manuscript presents and extensive new dataset of cleft monazite ages that are an important addition to exiting geochronological work in the Alps. In addition, the study presents an interesting analysis of the relationship between the duration of tectonic events and the spread in ages recorded in individual monazite crystals. The paper certainly falls within the scope of Solid Earth, but has significant shortcomings in its presentation and therefore **I recommend that it undergoes major revision before being accepted for publication in Solid Earth**.
The manuscript in its present state has three major shortcomings: 1) The data are presented and grouped in multiple ways that are not always clarified to the reader, which makes it impossible for the reader to judge whether the interpretations are sound. 2) The figure numbers appear to have been switched around several times during the preparation of the manuscript leaving many incorrect references, including a non-existing figure in the electronic supplement, making it nearly impossible to find the correct data. 3) Section 5.3 is not clearly argued and organised and needs to be revised to clarify the reasoning of the authors.
The abstract suggests the results of the study include major new findings, but if fact the results mostly confirm existing age information. To me, the value of the paper is more in the applicability of cleft monazite ages and the different expression of faster and slower tectonic processes in this dataset.

Note: This review was performed after the review of Dr. M. Rahn became available. I have tried to avoid duplication. I agree with most of his comments and suggestions.

Numbers between brackets below (1) are marked in the appended annotated manuscript.

**Specific comments**
Throughout: figure numbers and references to them are a mess throughout the manuscript. This needs thorough checking.
* The title is too general and not entirely on-topic. Metamorphic dome is rather unspecific. Please add an indication of location and perhaps time (Alpine). Given that the applicability of the method is not restricted to metamorphic domes, it may be better to rephrase the title altogether.
(4) It would be good to add a sentence or two at the end of the introduction that elaborates on the aims of the study.
(6) The more generally interested reader may have no idea where we are. I suggest to move Fig. 2 here and to add a reference to this figure to section 2.1. As indicated by Dr. Rahn, Figs. 1 and 2 need to be completed with coordinates, an indication of North, etc.
(10) The samples were grouped "roughly correlating to tectonic subdivisions" is a vague statement and leaves the reader unable to judge the criteria that were applied. This might give an unfortunate impression of arbitrary grouping, which renders the paper less persuasive.

(12) Regarding Figs 3 and 4, it would be good to mention briefly in the text, and not only in the figure caption what characteristic causes the zoning and how that is thought to be related to age information.

(13) See annotated manuscript for necessary edits to figure 1. The term "Geological-geometric in the caption is unclear. Perhaps best replaced by "Geometry".

Pages 7 and 8

(15) The images and all lettering in figures 3 and 4 should be enlarged so the reader is better able to assess the placement of the spots. Dr. Rahn mentions justified concerns regarding the placement of spots across boundaries between compositional domains. Some of the spots within one apparent compositional zone have different colours and it is not clear why that is the case (e.g., grains Duro2 and Klem1). The caption mentions "the color of the frame" but it is not entirely clear what that refers to. Is it the box around each weighted mean age result? Please clarify.

Pages 10-12

(18) In addition to Dr Rahn's comments. Please add spot numbers so the ages can be matched to the spots in figures 3 and 4. Enlarge lettering for readability; 6 pts at full size printing is usually considered minimal. I printed the pdf to A4 and most figures are too small in one way or another. The meaning of grey bands in these figures is not clear to me. Are the colours matched with those in figs 3-4?

(19) The content of section 5.1 is more fitting for the introduction than for the discussion.

(22) The decisions behind the groupings are not really explained and therefor the reason has no way to judge whether these decisions are sound, or not. In addition, as indicated in figure 3 some spots within the same apparent chemical domain (based on BSE, other compositional data that may have been used is not available to the reader) are marked with different colours and therefore apparently assigned to different groups for reasons not indicated. The groupings need to be argued more clearly to convince the reader.

(23) "to calculate, whenever possible, weighted mean domain ages (Fig. 7)." Should this be figure 8? It is unclear to me what determines whether a weighted domain age can be calculate or, in fact, how this is done. This needs more explanation. It seems that some of this explanation is actually in the paragraph following this reference. It would be better to first explain the procedure and then present the calculated ages.

(24) "It appears that if dissolution-precipitation may largely preserve the chemical composition of an affected crystal part, this would mean that areas with different chemical compositions may have reprecipitated simultaneously." What is the basis for the assumption of preservation? Has this been shown in the literature? Or do the data somehow suggest this? This needs to be explained better. For the second part of the sentence, I do not understand the reasoning either. I am not an expert on monazite dating, but if the authors want the reader to trust the validity of their interpretations, they need to argue their assumptions and decisions more clearly.

(26) There is no figure in the appendix. Has this figure been moved to the inset of Figure 8? Please correct accordingly.

(29) This is certainly not clear from Fig 2 or 7, and perhaps refers to Fig 8. If so, the statement that the age ranges within grains are generally longer in the Eastern and Southern domain does not appear to be supported. This could also refer to figs 5-7 (I now note that the panels are numbered continuously through figures 5-7, which is rather confusing), but there I do not see a consistency

in the graphs to support this statement either. This leaves me at a loss as to the basis of this this statement. This needs to be clarified.

(31) The shadings in Figure 9 render the ages illegible and this figure needs editing for clarity. It also seems that the age ranges are idealised to an extent: in 9b a 13.6 +/- 0.4 age is included in the 15-14 Ma range and in 9c a 13.4 +/- 0.3 age further West is included in the 13-11 Ma range. The 13-11 Ma area in 9c includes the area coloured in 9b, which contains almost exclusively ages >13 Ma. The colouring is persuasive but the averaged ages do not appear to match the areas all that closely.

From the caption it seems that the shaded areas are based on all ages from each sample, but the weighted mean average ages are based on a selection of those. Such, presumably unintentional juggling with the data makes it almost impossible for the reader to judge the value of the results and interpretations, which is very unfortunate. The authors need to do a better job in presenting their results to convince me that their interpretations are valid and can be used to underpin a tectonic scenario.

(36) The first sentence of the conclusions is a bit awkward. Please rephrase.

(37) "age clusters within individual crystals from a simply exhuming area have a less clear age distribution than samples from fault zone areas, or fast exhuming areas." This apparently main conclusion is new here and was not that clearly presented in the discussion. It would be good to add a couple of sentences specifying the argument and its conclusions. The same goes for the next sentence.

(38) The conclusions presented here paint a much clearer picture than section 5.3. The regional references (to the various faults and domes) are less clear in 5.3. Section 5.3 needs a thorough rewrite, and perhaps splitting in two sections to present the arguments more clearly. The first part could argue the conclusions about slow vs. punctuated events leading to broader and narrower age ranges, respectively, whilst the second part would present the tectonometamorphic development of the study area (leading to the conclusions in the second paragraph of section 6).

**Technical corrections**

Many suggested corrections for spelling and grammar and indicated in the annotated manuscript. In addition, please consider the following numbered comments.

(1) Earlier in the Abstract the authors argue that using cleft Mz is superior to other dating techniques because it is not cooling based. It then seems somewhat inconsistent to highlight cooling in line 8. Better to say exhumation only.

(2) The final sentence is very general and it would be better to be more specific as to what kind of information can be derived by dating of cleft-Mz.

(3) Alpine is used a lot here (once also without capital), but the processes considered in the study are not likely to be restricted to Alpine orogenesis or the Alps orogeny. It would be better to phrase this a bit more generally.

(5) This sentence is very vague. Either be more specific, or leave out.

(7) I am not sure what is meant by 'staggered' exhumation.

(8) This reference to Central Lepontine is unclear, because the next sentences refer to more specific areas that are indicated in Fig. 2.

(9) Number figures in order of mention in the text. The current figures 1 and 2 should be swapped. This is consistent with comment (6) above.

(11) Please add the groupings to Table 1.

(14) Please check figure references. This should probably Figs 5 through 7.

Pages 7 and 8

(16) On both pages colour and color is used in the same sentence. Please use either British or American English spelling consistently.

(17) The caption of Table 2 does not describe its content. Please correct. The text that is now in the caption is in fact a note.

(20) Suggestion to rephrase: "…existing grain. Alternatively, dissolution-reprecipitation may cause precipitation of a secondary monazite-(Ce) phase from the fluid film at the surface of the primary phase."

(21) The reference to Grand'Homme et al., 2018 is not in the bibliography.

(25) It would be good to add a reference supporting these statements.

(27) "another reason" is confusing here, because in lines 17-18 prolonged tectonic activity is already mentioned as a possible reason for age spread. In addition, the description "…prolonged phases of low-intensity tectonic activity of multiple small deformation events…" is very vague. Please revise to address both these issues.

(28) Correct references. Presumably Fig. 8 and Table 2.

(30) Clarify the location of the Rhine-Rhone line in the text and give it the same font size in figure 9 as all other faults

(32) panels a and b are in figure 5, j and k are in 6. All figure references need to be thoroughly checked and corrected.

(33) "clear age patterns within the crystals" is a very vague criterion, which can not be judged by the reader. Please be as specific as possible.

(34) Again, "staggered" is used in a sense that is not entirely clear to me. It would help if the authors clarify to which part of Fig 8 this refers.

(35) The mention of hydrothermal gold mineralisation is very random and appears to have little relationship with the rest of the study. Consider leaving this out.

Pages 19 and further - Bibliography
Missing from the reference list:
Milnes (1974)

Not referenced in the paper:
Frisch (1979)
Frisch et al (2000)
Glotzbach
Keller et al (2006)
Kralik et al

Putnis 2002 and 2009
Schmid et al (1996)

Possible mistakes:
Grand'Homme et al. 2016 or 2018?
Steiger and Jaeger: Title is "Subcommittee on Geochronology: Convention …."
Wiederkehr et al. 2008 or 2009?

[revised manuscript text omitted]

---

## Referee Comment (RC1) · Meinert Rahn (Referee) · 10 Mar 2019

The study of Bergemann and co-workers presents 480 single spot ages and 33 weighted mean ages from 19 locations and their cleft monazites within the northern Lepontine Dome (and adjacent to it). These ages are used to decipher the exhumation and tectonic history of the Lepontine dome, as the ages are compared with other geochronological data supposed to represent the Neogene cooling history.

To me, there is no doubt that the provided data are interesting for publication in Solid Earth. However, for the moment the manuscript and submitted material has for the moment several critical shortcomings that I would recommend to fix prior to becom-

ing acceptable, as I consider them critical, if the paper wants to have the impact the presented topic deserves and the general title promises. My major concerns are the following:

1. The title of the manuscript suggests that the monazite data provide new constraints on the tectonic and exhumation history of the Lepontine dome, while the discussion of the data mostly refer your data to already existing constraints of the dome exhumation and Tectonics. As such the focus of the paper is more on methodical aspects of monazite dating (e.g. monazite formation temperatures, relationship to other dating techniques and their closure temperatures).

2. There is throughout the paper a mess with the figure numbers. My assumption is that the authors may have changed these numbers shortly before submission of the manuscript. I invite the authors to check carefully all figure numbers when revising their submitted material. I also note that at several occasions the authors refer to figure 1 in the appendix, which I was unable to locate.

3. For the moment, the chapter "Results" is ultra-short and lacks important information. In your discussion chapter, you tend to describe your results at several places, which should be done in the "Results" chapter. The "Results" chapter should also be used to clarify, what data you will discuss in the "Discussion" chapter and which data will not further be discussed.

4. My major concern is that the authors are rather vague with their methodical descriptions. Some of these details should be part of the "Introduction" chapter, of a new methodical chapter or part of the "Results" chapter. Let me summarize this in five points that I would expect the authors to provide more information about:

4a. The authors talk about the "monazite stability field" (e.g. p. 13, line 3; p. 15, line 6; p. 17, line 20), however, they never discuss, what they mean with "stability field". Note that the authors on p. 12, line 5, talk about "disequilibrium", without clarifying what kind of "disequilibrium" they refer to. I would assume that this is not a "thermodynamic stability", but they rather consider a kind of temperature window, in which the cleft monazites were formed. If correct, it might more correctly speak about the "monazite formation temperature window". This aspect is important, because in the "Discussion" chapter you compare the formation of monazite with the closure temperatures of low-temperature thermochronology methods (which seems to suggest some kind of closure-T for cleft monazites).

4b. The authors present BSE images for each on the investigated monazite crystals (their figures 3 and 4). However, it remains unclear what the visible colour changes mean within each individual crystals (no chemical data are given except for a few selected elements in the supplementary data file) and how the authors have chosen their analytical spots on these crystals. The only information is that the authors state that they have placed the SIMS spots were placed "according to compositional domains" (p. 3, line 29). Accordingly, we would expect that spots of same colour rings in figures 3 and 4 would always represent areas of same gray colour in the BSE image. This clearly is not the case for e.g.in the DURO1 crystal the yellow spots seem to only roughly follow a lighter lamella, but overlap with darker areas around, in the DUTH2 crystal the orange spots lie within a lighter rim, but spread into the darker centre next to it. The authors have to state clearly their criteria in how to assure that spots are not mixtures between to different generations of monazite formation.

4c. The authors state that they have avoided measurements next to cracks and holes (p. 15, line 30). This statement is in contradiction to e.g. the red spots in BETT11, the blue spots in VANI6, the red spots in VANI5 etc. I assume that the criteria is more likely defined by the analysis itself showing a deficit in elements rather than the geometric vicinity. The authors have to clarify this issue.

4d. The authors have to clarify on the basis of which criteria they have chosen the weighted mean ages out of the spot analyses. In Figure 5a (VANI6), it seems obvious that the orange group weighted mean age is formed out of all orange spots. Agewise, however, these spots seem to overlap with the gray spots. So, how have the authors

separated between orange and gray? In figure 5b (BETT11), the four red spots show age overlap, but they are not combined to one weighted mean age. Why not? In figure 5c (DURO1), the four blue spots form a weighted mean age, but the gray spot next to it (same age) is not part of it. Why not? I could continue the same way for most of the diagrams in the figures 5 to 7. I am sure that there are good reasons for the authors' choice of the weighted mean ages, but for the moment, this choice cannot at all be assessed by the reader and looks very arbitrary, not scientifically founded. The authors have to explain to the readership their selection criteria, and for such purpose, it may be needed to better illustrate the different compositional variations among the individual monazite analyses.

4e. According to figure 1, there are three age groups (with some samples showing more than one). In figures 5 to 7, however, the authors have several samples with more than two weighted mean ages, in figure 8, the three age groups are no longer visible, and in your discussion chapter, you discuss a much finer distinction among the age groups (see also figure 9). We would recommend to the authors to clarify this issue of age groups in an early stage (e.g. in the results chapter and then stick to it throughout the entire discussion chapter. For the moment, the reader gets lost due to the many age groups and the inconsistency between the figures.

4f. Figure 8 shows the ages again, but in probability density plots. Up to here (in particular in the figures 2 to 7, the reader has gained the impression that single spot data are clustered to weighted mean ages. Here, however, the authors seem to have split the ages again in single spot ages to form new curves and density plots. The same is true in the "Discussion" chapter on pages 15 and 17: Sometimes, the authors refer to single spot ages and sometimes they refer to weighted mean ages. I do not understand why the authors refer to weighted mean ages at all, if they afterwards selectively use the information that fits best their arguments. The authors have to clarify their strategy in interpreting their results. They have to clarify the meaning of their "weighted mean ages" in that sense. They also have to explain how uncertainties were calculated for

the different types of ages.

4g. Figure 8 shows a kind of clustering of the single spot ages. In this plot the authors also show previous literature data (in gray), but these are not included in their clustering pattern (we do not know, whether this is the case for the curves in the inset below). In Figure 9, however, their interpretation includes all the literature data (e.g. for the Gotthard nappe and the Aar Massif). This is inconsistent. Either you use all data or you do not. The authors have to lay out their strategy on what data are to be interpreted and then stick to it.

4h. Figure 2 shows nicely how the authors divise their samples into regional groups. However, in the "Discussion" chapter, their division seem to not make sense in many respects as they tend to again subdivide their division. I make two examples: (1) On p. 15, line 19, the authors refer to "the entire (north)eastern region that seem to act differently than the rest of the region. This "sub-region" is not well defined. (2) Figure 2 places sample DUTH6 to the edge of the "Center" region, but in figure 9, this sample rather behaves like the samples in region "West", so why DUTH6 is part of the "Center" area?

4i. In chapter 5.3, the authors compare their data with data from other thermochronometers. However, this comparison is incomplete in that sense that sometimes ages are quoted, sometimes not, sometimes the authors only refer to the interpretation of the previous workers without referring to the geochronological evidence. This should be done in a more careful, systematic and transparent way. I recommend e.g. that they authors clearly state what time and methodical information they use for their discussion (e.g. they refer to K/Ar ages, ZFT and ZHe ages, but they do not use AFT or AHe ages.

4j. The "Discussion" chapter starts with an interesting subchapter on hydrothermal monazite crystallisation. This is exactly the information needed to understand methodically the authors' strategies. However, as far as I understand, this chapter is not a

"result" but a initially chosen "strategy" on how the monazite ages are to be interpreted (it looks therefore misplaced in the "Discussion" chapter). The authors should somewhere clarify their strategy of the understanding on how monazite is formed.

5. From the title of the paper, the reader expects some new information about exhumation and tectonics within the Lepontine Dome. However, in such respect, the "Discussion" chapter has been disappointing for me. The authors support existing cooling/exhumation paths and tectonic events, but they have no courage to suggest any new "events". I agree that the paper title could be understood as "Confirming metamorphic dome exhumation", and I also agree that the problem with monazite dating is the fact that the ages cannot be related to a temperature value (closure temperature) in contrast to other methods. Nevertheless, I also see potential about the information of the monazite ages that the authors seems to keep untouched. What e.g. is the function of the Rhone-Rhine line (e.g. in figure 9e, f)? Where do the new results show an extension of previous time windows or a focussing on smaller windows for existing phases of tectonic activity? In the end, the "Discussion" chapter does not seem to provide any new information.

Looking through these comments (and the detailed comments below) I would recommend to the authors to thoroughly revise their manuscript (major revisions). For me, there is no doubt that this study would be an excellent contribution to Solid Earth. However, for the moment, publication of the extensive data set would fail to gain credibility among the readers, because so many methodical details are only vaguely described and therefore lack credibility.

For detailed comments to the manuscript, see attached pdf file.

Please also note the supplement to this comment:
https://www.solid-earth-discuss.net/se-2019-10/se-2019-10-RC1-supplement.pdf
* * *
[Figure]

**Supplement:**

**Constraining metamorphic dome exhumation and fault activity through hydrothermal monazite-(Ce)**

Christian A. Bergemann, Edwin Gnos, Alfons Berger, Emilie Janots, Martin J. Whitehouse

by Meinert Rahn

The study of Bergemann and co-workers presents 480 single spot ages and 33 weighted mean ages from 19 locations and their cleft monazites within the northern Lepontine Dome (and adjacent to it). These ages are used to decipher the exhumation and tectonic history of the Lepontine dome, as the ages are compared with other geochronological data supposed to represent the Neogene cooling history.

To me, there is no doubt that the provided data are interesting for publication in Solid Earth. However, for the moment the manuscript and submitted material has for the moment several critical shortcomings that I would recommend to fix prior to becoming acceptable, as I consider them critical, if the paper wants to have the impact the presented topic deserves and the general title promises. My major concerns are the following:

1. The title of the manuscript suggests that the monazite data provide new constraints on the tectonic and exhumation history of the Lepontine dome, while the discussion of the data mostly refer your data to already existing constraints of the dome exhumation and Tectonics. As such the focus of the paper is more on methodical aspects of monazite dating (e.g. monazite formation temperatures, relationship to other dating techniques and their closure temperatures).

2. There is throughout the paper a mess with the figure numbers. My assumption is that the authors may have changed these numbers shortly before submission of the manuscript. I invite the authors to check carefully all figure numbers when revising their submitted material. I also note that at several occasions the authors refer to figure 1 in the appendix, which I was unable to locate.

3. For the moment, the chapter "Results" is ultra-short and lacks important information. In your discussion chapter, you tend to describe your results at several places, which should be done in the "Results" chapter. The "Results" chapter should also be used to clarify, what data you will discuss in the "Discussion" chapter and which data will not further be discussed.

4. My major concern is that the authors are rather vague with their methodical descriptions. Some of these details should be part of the "Introduction" chapter, of a new methodical chapter or part of the "Results" chapter. Let me summarize this in five points that I would expect the authors to provide more information about:

   a. The authors talk about the "monazite stability field" (e.g. p. 13, line 3; p. 15, line 6; p. 17, line 20), however, they never discuss, what they mean with "stability field". Note that the authors on p. 12, line 5, talk about "disequilibrium", without clarifying what kind of "disequilibrium" they refer to. I would assume that this is not a "thermodynamic stability", but they rather consider a kind of temperature window, in which the cleft monazites were formed. If correct, it might more correctly speak about the "monazite formation temperature window". This aspect is important, because in the "Discussion" chapter you compare the formation of monazite with the closure temperatures of low-temperature thermochronology methods (which seems to suggest some kind of closure-T for cleft monazites).

   b. The authors present BSE images for each on the investigated monazite crystals (their figures 3 and 4). However, it remains unclear what the visible colour changes mean within each individual crystals (no chemical data are given except for a few selected elements in the supplementary data file) and how the authors have chosen their analytical spots on these crystals. The only information is that the authors state that they have placed the SIMS spots were placed "according to compositional domains" (p. 3, line 29). Accordingly, we would expect that spots of same colour rings in figures 3 and 4 would always represent areas of same gray colour in the BSE image. This clearly is not the case for e.g.in the DURO1 crystal the yellow spots seem to only roughly follow a lighter lamella, but overlap with darker areas around, in the DUTH2 crystal the orange spots lie within a lighter rim, but spread into the darker centre next

to it. The authors have to state clearly their criteria in how to assure that spots are not mixtures between to different generations of monazite formation.

c. The authors state that they have avoided measurements next to cracks and holes (p. 15, line 30). This statement is in contradiction to e.g. the red spots in BETT11, the blue spots in VANI6, the red spots in VANI5 etc. I assume that the criteria is more likely defined by the analysis itself showing a deficit in elements rather than the geometric vicinity. The authors have to clarify this issue.

d. The authors have to clarify on the basis of which criteria they have chosen the weighted mean ages out of the spot analyses. In Figure 5a (VANI6), it seems obvious that the orange group weighted mean age is formed out of all orange spots. Agewise, however, these spots seem to overlap with the gray spots. So, how have the authors separated between orange and gray? In figure 5b (BETT11), the four red spots show age overlap, but they are not combined to one weighted mean age. Why not? In figure 5c (DURO1), the four blue spots form a weighted mean age, but the gray spot next to it (same age) is not part of it. Why not? I could continue the same way for most of the diagrams in the figures 5 to 7. I am sure that there are good reasons for the authors' choice of the weighted mean ages, but for the moment, this choice cannot at all be assessed by the reader and looks very arbitrary, not scientifically founded. The authors have to explain to the readership their selection criteria, and for such purpose, it may be needed to better illustrate the different compositional variations among the individual monazite analyses.

e. According to figure 1, there are three age groups (with some samples showing more than one). In figures 5 to 7, however, the authors have several samples with more than two weighted mean ages, in figure 8, the three age groups are no longer visible, and in your discussion chapter, you discuss a much finer distinction among the age groups (see also figure 9). We would recommend to the authors to clarify this issue of age groups in an early stage (e.g. in the results chapter and then stick to it throughout the entire discussion chapter. For the moment, the reader gets lost due to the many age groups and the inconsistency between the figures.

f. Figure 8 shows the ages again, but in probability density plots. Up to here (in particular in the figures 2 to 7, the reader has gained the impression that single spot data are clustered to weighted mean ages. Here, however, the authors seem to have split the ages again in single spot ages to form new curves and density plots. The same is true in the "Discussion" chapter on pages 15 and 17: Sometimes, the authors refer to single spot ages and sometimes they refer to weighted mean ages. I do not understand why the authors refer to weighted mean ages at all, if they afterwards selectively use the information that fits best their arguments. The authors have to clarify their strategy in interpreting their results. They have to clarify the meaning of their "weighted mean ages" in that sense. They also have to explain how uncertainties were calculated for the different types of ages.

g. Figure 8 shows a kind of clustering of the single spot ages. In this plot the authors also show previous literature data (in gray), but these are not included in their clustering pattern (we do not know, whether this is the case for the curves in the inset below). In Figure 9, however, their interpretation includes all the literature data (e.g. for the Gotthard nappe and the Aar Massif). This is inconsistent. Either you use all data or you do not. The authors have to lay out their strategy on what data are to be interpreted and then stick to it.

h. Figure 2 shows nicely how the authors divise their samples into regional groups. However, in the "Discussion" chapter, their division seem to not make sense in many respects as they tend to again subdivide their division. I make two examples: (1) On p. 15, line 19, the authors refer to "the entire (north)eastern region that seem to act differently than the rest of the region. This "sub-region" is not well defined. (2) Figure 2 places sample DUTH6 to the edge of the "Center" region, but in figure 9, this sample rather behaves like the samples in region "West", so why DUTH6 is part of the "Center" area?

i. In chapter 5.3, the authors compare their data with data from other thermochronometers. However, this comparison is incomplete in that sense that sometimes ages are quoted, sometimes not, sometimes the authors only refer to the interpretation of the previous workers without referring to the geochronological evidence. This should be done in a more careful, systematic and transparent way. I recommend e.g. that they authors clearly state what time and methodical information they use for their discussion (e.g. they refer to K/Ar ages, ZFT and ZHe ages, but they do not use AFT or AHe ages.

j. The "Discussion" chapter starts with an interesting subchapter on hydrothermal monazite crystallisation. This is exactly the information needed to understand

methodically the authors' strategies. However, as far as I understand, this chapter is not a "result" but a initially chosen "strategy" on how the monazite ages are to be interpreted (it looks therefore misplaced in the "Discussion" chapter). The authors should somewhere clarify their strategy of the understanding on how monazite is formed.

5. From the title of the paper, the reader expects some new information about exhumation and tectonics within the Lepontine Dome. However, in such respect, the "Discussion" chapter has been disappointing for me. The authors support existing cooling/exhumation paths and tectonic events, but they have no courage to suggest any new "events". I agree that the paper title could be understood as "Confirming metamorphic dome exhumation", and I also agree that the problem with monazite dating is the fact that the ages cannot be related to a temperature value (closure temperature) in contrast to other methods. Nevertheless, I also see potential about the information of the monazite ages that the authors seems to keep untouched. What e.g. is the function of the Rhone-Rhine line (e.g. in figure 9e, f)? Where do the new results show an extension of previous time windows or a focussing on smaller windows for existing phases of tectonic activity? In the end, the "Discussion" chapter does not seem to provide any new information.

Looking through these comments (and the detailed comments below) I would recommend to the authors to thoroughly revise their manuscript (major revisions). For me, there is no doubt that this study would be an excellent contribution to Solid Earth. However, for the moment, publication of the extensive data set would fail to gain credibility among the readers, because so many methodical details are only vaguely described and therefore lack credibility.

Detailed comments:

The following comments are sorted according to page numbers and text lines (in the pdf provided). They were continuously gathered while reading. Many of them have later been clustered to major comments (see above). With respect to grammar or English style corrections, this reviewer confesses to not be a native English speaking person; accordingly, suggestions concerning grammar and phrasing are to be seen as suggestions. I also highlight, if a comment is thought to be a suggestion (for e.g. clarifying a statement or shortening the text).

Title — The authors always refer to "monazite-(Ce)". This is ok as the IMA points out that monazites could be dominated by different REE in their formula. However, the authors never confirm that Ce is the dominating RE element in the formula of their 19 samples. The paper has no compositional data except for those in the supplementary data set. I assume that much of the division of the analysis spots (figures 3 and 4) is due to compositional arguments. However, the reader has no chance to assess this division. The reader has not even a chance to find out, whether the "monazite-(Ce)" in the title is correct or not.

Title — Looking back to the "Discussion" chapter, this reviewer has rather the impression as if the authors do not "constrain" but only confirm previous information on the exhumation and tectonic history of the Lepontine Dome. I suggest to thoroughly revise the manuscript and then decide on whether to change the title of the paper. The same might apply for the "Abstract".

Title — The term "hydrothermal monazite-(Ce)" is varied in the manuscript. Sometimes, the authors refer to "cleft monazite-(Ce)", sometimes to "hydrothermal cleft monazite-(Ce)", and the reader gets puzzled about the different expressions. It would help to clearly state somewhere in the manuscript that these monazites all come from clefts. I am not so happy about the term hydrothermal as it suggests that these monazites have formed in a flush of "hot" fluid, the temperature of which might not have been in equilibrium with the surrounding rock. If so, any later comparison with closure temperatures of thermochronologic systems may not be useful, as the temperature of the monazite formation may have been completely different from the temperature controlling closure of other geochronologic methods.

Line 2 — The authors refer to the "Central Alpine Lepontine metamorphic dome", which is a rather unusual term. First, figures 1, 2 and 9 show that the study area does not cover the south of the dome (along the Insubric Line), but its entire E-W extension (so, "Central" may not be needed). Second, the Lepontine Dome is an Alpine structure, so, I would suggest that there is no need to use the term "Alpine" here.

| | |
|---|---|
| Line 4 | Suggestion: delete "The" at beginning of sentence |
| Line 5 | There seem to be a double space between "between and "19" |
| Line 6 | The authors use the abbreviation "Ma" (= mega annum) for both, specific moments in time and timer periods. Please check the instructions for authors that suggest to use "Ma" for time spots", however, "Myr" for time periods. I fully support such instruction. Accordingly, the authors should write here "lifetime of 2 to 7.5 Myr" |
| Line 6 | Suggestion to put "combined with age distribution" into brackets. |
| Line 9 | Suggestion: "In the east and south of the northern Lepontine dome" |
| Line 9 | Suggestion: delete "the" before "units" |
| Line 9 | Suggestion: add ", respectively," after "Ma" |
| Line 10 | Suggestion: "at 15-10 Ma. Cleft monazites…" |
| Lines 10/11 | There seem to be a mismatch with the statement "youngest" in line 10 and "A last phase" in line 11. "Youngest" means "last phase", thus these phase should be the same… Furthermore, I would suggest using the term "age signal" instead of "phase" as the latter already suggests that this has been a distinct tectonic or thermal "event". |
| Line 11 | Suggestion: "along the Simplon Fault". One further note: If you quote names of domes and faults, make sure that they are constantly with upper case or lower case. So, either Lepontine Dome or Lepontine dome, Simplon Fault or Simplon fault. |
| Lines 11/12 | Add hyphen: "strike-slip" |
| Line 12 | Suggestion: "faults along the Rhine and Rhine-Rhone faults" |
| Line 13 | Do you mean "stability" or "formation"? |
| Line 13 | "directly" instead of "direct" |
| Line 15 | "experience" instead of "experienced" |
| Line 16 | Suggestion: Start line with "For the Lepontine Dome, this evolution is an interplay …" |
| Line 16 | Suggestion: "motion along" instead of "activity of" |
| Line 16 | Suggestion: "define" instead of "dominate" |
| Line 17 | Suggestion: "the western edge of the dome" instead of "the western parts of the area". Note that throughout the text, you frequently use "in this area", but you are not very specific, what area exactly you are referring to. I would expect to get some references here. |
| Line 17 | The statement "Although much of the (thermo)chronological history  of the area is well known" is rather cryptic. Does this refer to the fact that there are a lot of data concerning the post-peak metamorphic history? Or does this mean that the tectonic history is known in detail? Since you do not provide any references and the statement is rather broad, the reader cannot assess the meaning of this statement. |
| Line 18/19 | Interestingly, here you refer to AFT and AHe ages, while in the Discussion chapter, you do not refer to any of the existing studies including such data and you do nowhere cite any of them. This is ok with me (as e.g. AFT and AHe ages may already be influenced by topographic evolution. Nevertherless, there is a mismatch between this statement and the discussion of your data afterwards. Finally one suggestion for a rewording: "existing cooling ages of the Rb-Sr, fission track (FT) and (U-Th)/He systems". |
| Line 20 | Suggestion: "(e.g. Parrish, 1990). It is highly resistant…" |
| Line 21 | "by diffusion" instead of "through diffusion" |
| Line 22 | not clear, what "geologically reactive" means |

| | |
|---|---|
| Line 1 | "by" instead of "through". |
| Line 1 | What do you mean by "mediation"? |
| Line 3 | Suggestion: "occasionally contain monazite-(Ce). They represent voids…" |
| Line 7 | Suggestion: "were" instead of "could be" |
| Line 11 | Suggestion: "…tectonic activity. Accordingly, fissures and clefts are …" |
| Line 13 | "interacted" instead of "interacts" |
| Line 13 | Suggestion: Start new sentence with "Dissolution and precipitation…" |
| Line 14 | Suggestion: "led to" instead of "causing" |
| Line 18 | Suggestion: "by using secondary ion mass spectrometry (SIMS)" |
| Line 19 | At the end of this introduction, I miss a paragraph that presents the aims of this study. I addition, I miss an introduction to the methodology of monazite dating, not the technical issues, but the prerequisites that define the strategy of the study. |
| Line 22 | Add "European" to "Alps |
| Line 23 | This bracket lists examples of metamorphic domes, but their allocation is once to a country ("Austria") and once to the part of an orogeny ("western Alps"). I would suggest referring either to Switzerland/Austria or to eastern and western Alps. |
| Line 25 | The references listed here have a strange ordering, either alphabetic or with decreasing age. Please check the author's instructions. |
| Line 26 | The statement that the Western and Central Alps had a "complex tectonic and metamorphic history" is not very convincing. "Complex" with respect to what? Orogens tend to be complex anyway (fortunately, this keeps us going to find out more details of this history!), but the statement is too vague, too general. You may start the chapter with such a statement. Furthermore, I refer to the detail that Schmid et al. (2004) did not use the term "Central Alps", but divided the Alps into western and eastern Alps. |
| Line 31 | You may put "in excess of 650°C in some regions" into brackets. |

| | |
|---|---|
| Line 1 | not clear, what you mean by "staggered exhumation", I am doing research in the field of exhumation for 20 years, but have never heard of this expression. |
| Line 1 | I would recommend to add references to this statement |
| Line 4 | Suggestion: "later in time at 18-15 Ma |
| Line 5 | I would recommend to use the term "normal faulting, rather than "detachment", as the term "detachment" has no direction. |
| Line 9 | Here, the term "Simplon Fault zone" is used, but in line 11 "Simplon shear zone". If these terms describe different zones, then the authors should explain the difference. |
| Line 10 | "Alpine" instead of "alpine" |
| Line 14 | From figures 1 and 2 it is evident that the study area is the northern Lepontine dome. Nevertheless, the exhumation of the Lepontine Dome cannot be described without the Insubric Line. (15 km of differential vertical exhumation!). The authors should consider including the Insubric Line in their "Geological setting". |

| Line 15 | "area" instead of "Area" |
|---|---|
| Line 16 | Suggestion: "The study area comprises the northern half of the Lepontine …" |
| Line 17 | "Forcola Fault" |
| Line 17 | The "Val d'Ossola" is not visible in any of the figures. |
| Line 18 | Suggestion: "Aar massif to the north (see Fig. 1 for the tectonic position of the samples)…" |
| Line 19 | Should not Janots et al. 2009 be added to this list, as it also contains monazite ages? |
| Line 20 | The statement that the four groups correlate "to tectonic subdivisions" cannot fully be assessed by the reader: First, there is no tectonic subdivision between central and western part (the Ticino and Toce domes are separated by what tectonic feature?). Second, In figures 1, 2 and 9, the Forcola Fault ends south of the study area, and if the authors refer to the extension of the Forcola Fault to the north, the sample VALS would be located in the central part. How about separating along the western rim of the Adula nappe? |
| Line 22 | "bound" instead of "bounded", as it is related to "to bind" |
| Lines 23-25 | I would shift the last sentence to the beginning of the next chapter. |
| Line 28 | Note that there are different spellings of "backscatter". In figures 3 and 4, you write "back-scatter". |
| Line 28 | Suggestion: "Backscatter electron (BSE) images were used to define spots suitable for analytical investigations." It is, however, unclear to me, how you defined the different groups (of different colours) and how you have chosen the analytical spots. This needs more information. |
| Line 29 | The statement "according to compositional domains" cannot be understood without further information: How do you define a compositional domain? What compositional data were available? I refer to the fact that a change in RE element (but otherwise constant composition of the monazite) may not really show a difference in colour in the BSE image… It would be helpful to know what analytical data were available, and which of these data were used to "define the compositional domains". |
| Line 30 | Suggestion: "As far as possible, spot measurements next to cracks or holes were avoided". |
| Line 30 | Looking at e.g. figures 5a, 5j, or 5l, this statement cannot be understood. In the corresponding crystals, there are measurement spots directly next to cracks… |
| Line 31 | Suggestion: "in such areas" |

| Figure 1 | This figure is not yet finished. First and foremost, there are no coordinates, there is no scale and there is no north direction. I would recommend to enlarge the map on top (a), and enlarge the labels (b) to (e). The legend contains the abbreviation "mzt" that is not explained in the caption. According to the thick black line in map (a), the profile (c) should end at the border to the Aar Massif, but in (c), it extends into it. Profile (d) has gray units that do not show up in the map (a). Each of the profiles has its own scale, thus, scales should be added to figure parts (b) to (e). Faults are not labelled.

One methodical problem is that the authors in this figure's legend refer to three age groups, but in the discussion chapter (and in figure 9), there are more age groups. This is inconsistent. Do you need to mark the different groups in this figure already? |
|---|---|
| Figure caption | Line 1: switch the two references, as they are not in correct order, line 4: There is no Wiederkehr et al. 2008 reference in your reference list. Do you mean the 2009 reference? Line 4: Why does the caption uses a different text for profile (e)? |

Table 1    In this table, all samples are given names with upper case spelling. However, in figures 1 to 7, the authors sometimes use lower case or upper case. In addition, sometimes, there is a gap between name and number, sometimes not. I recommend adapting the names throughout the paper and figures and tables, and I would use a space between name and number, as several names end with "I", which then might be interpreted as a "1".

For some of the samples, you note coordinates with "~". What is the difference? If the location of the sample is not sure; I would rather recommend to reduce the digits after the comma of the angle minutes (e.g. sample SALZ2) rather than adding a "~".

Figure 2    The frame in the inset does not fit to the area of the large map. Similar to figure 1, coordinates and north direction are missing, the abbreviation "mzt" should be explained, The label of the "Centovalli Fault" is partly covered with blue colour.

Line 4      Suggestion: *by" instead of "with

Line 7      delete "presented"

Line 8      Suggestion: "reported" instead of "given"

Lines 10/11  Suggestion: "… pending). Table 2 provides an overview and Figures 3 and 4 show measurement positions and the division of the analysis spots into different age groups, represented by different colours.

Lines 11/12  The statement "As there are difficulties with the U-Pb system for hydrothermal monazite-(Ce)" is very cryptic. What difficulties do you mean? And how have the authors dealt with these difficulties?

Line 13     "see discussion in chapter 5.2".

Line 13     I note that the "Results" chapter only contains these four lines. I strongly recommend describing the results. What should the reader see? The next pages are figures 3 to 7 and table 2, and if the reader goes through these many data, there is no guarantee that the reader will come to the same conclusions as the authors about what is important and what is not. As noted in the general comments above, I have rather collected doubts about the usefulness of the chosen strategy of the authors and the credibility in their chosen analytical spots, derived weighted mean ages etc. I would argue that a careful description of the results would help much not losing the reader half way. You should add a description on how the spots have been selected, grouped in the diagrams of figures 5 to 7, how the weighted mean ages were afterwards grouped. It may help starting with figure 8 instead that shows an overview of all ages.

Figure 3    Many of the colours of the spots are hard to be distinguished (e.g. figure e). All sample names should be similar to tables 1 or 2. For several sub-figures, the numbers are too small to read (perhaps enlarge figure to full page). You should clarify that this BSE images show all spots made for this study.

Figure caption   "Back-scatter" or "Backscatter"? (see comment to page 3, line 28). You have to add the information that the grains shown are from the south and west areas only, e.g. "Backscatter electron images of all studied cleft monazite-(Ce) grains from the South and West areas." Line 1: "content" instead of "contents", "Coloured ovals" instead of "Spots". Line 2: "different colours indicate different chemical …". Lines 2/3: "indicates those data, for which a weighted mean 208Pb/232Th age could be calculated.

Figure 4    For this figure, see comments to figure 3 and figure 3 caption. Again, it should be clarified in the caption that these samples are from the Center and East areas.

Table 2    The caption of this table does not fit. The information given here should be placed elsewhere. Most of the numbers in the "Figure" column are wrong. Obviously, the authors have changed the figure numbers in a late step and not adapted text and tables. The last column to the right lists the range of single grain ages. I do not understand why there are empty lines in this column as the range can also be given to those age groups that are combined to a weighted mean age. For sample "GRAESER1", the number of points for the older weighted mean age should be 6 instead of 5 (see figure 3e), and the second weighted mean age is missing in figure 5e. I would recommend to draw lines between the four different areas (South, West, Center, East).

Page 10-12

Figures 5-7    Looking at these diagrams, I have many questions about how you define a weighted mean age. In some cases, all spots of the same colour form such a weighted mean age (e.g. 5a, 5b, 5g). More frequently, the weighted mean ages are calculated for a selection of ages of the same colour only (see e.g. 5b, 5e). In some cases, it is obvious that only one colour bar iw far off (e.g. in 5e, red), but in other cases (e.g. 5b, red), I do not know, why one age has not been included, even though the vertical bars overlap. Finally, there are examples, where age bars overlap, but were not used to calculate a weighted mean age (e.g. in 5e, young yellow ages, in figure 7s, gray bars in the figure centre). The authors have to explain their selection procedure in detail, otherwise there is no credibility to their ages at all.

In figure 6p, one vertical bar has a lighter blue colour; why?, In figure 7s to the left, there is a short bar without colour.

Line 2    To me the content of chapter 5.1 is misplaced. The reasoning given here is not the result of the study, but a strategy that the authors have chosen beforehand, as far as I understand. If there are observations, features, analyses that support your reasoning, then they should be presented as results and used to interpret your ages properly. For the moment, the content of this chapter should be transferred elsewhere, e.g. in a new chapter on methodology.

If you argue that "later reactions may be aided by secondary porosity and fracturing induced by the previous dissolution-reprecipitation/recrystallization events", then show evidence for it. Show what you mean, add another figure like figs 3 or 4 to show corresponding features. For the moment, the entire chapter 5.1 is a black box to the reader, as the authors do not present any evidence for dissolution, reprecipitation, recrystallization. The authors should better define, what they mean with these processes and present examples for it.

Line 5    Suggestion: "later reactions may occur"

Line 5    The term "disequilibrium" is hard to understand. "Disequilibrium" with respect to what?

Line 5    Suggestion: "The may be caused by a tectonic event for a …"

Line 10    Suggestion: "phase may precipitate"

| Line 12 | Suggestion: "fluid remains connected to a …" |
| | |
| Line 12 | Suggestion: Start sentence with "A dissolution …" as this process has not been defined. |

| Line 2 | There is no 2018 reference for Grand'Homme et al. in the reference list. Do you mean the 2016 reference? |
| | |
| Line 8 | The statement "placed … according to growth domains" cannot be assessed by the reader. The authors should show their arguments, e.g. with one example. How do you define a growth domain? How do you define what is old, what is younger? Do ages from different growth domains fit to the measured ages (same age order)? Have you tested the order? |
| | |
| Line 9 | The statement "on the basis of chemical composition" cannot be assessed by the reader. What chemical information did the authors gather and what chemical arguments did they use? This information is of major importance if forthcoming studies should apply the same methodology. |
| | |
| Line 11 | The reference to "Fig. 7" should probably be "Fig. 5-7". |
| | |
| Lines 11/12 | The statement "It appears that dissolution-precipitation m ay largely preserve the chemical composition of an affected crystal part" cannot be assessed by the reader. Is this an assumption? If not, what is the evidence you found? What are the conclusions drawn from it? |
| | |
| Line 12 | "areas of" instead of "areas with" |
| | |
| Line 13 | Not clear, what you mean by "this" ("Despite this,….") |
| | |
| Lines 13/14 | The statement "only in a few, clear cases" cannot be assessed by the reader. Which cases? And what means "clear"? |
| | |
| Line 14 | Suggestion: "age calculation, to avoid …" |
| | |
| Line 15 | Suggestion: "single" instead of "distinct" |
| | |
| Line 18 | Suggestion: "New growth on an existing crystal results in sharp chemical (or colour) boundaries between growth zones." Note however, that this is only the case, if the growth zones differ in composition. Thus, it is an argument, but it does not need to be… |
| | |
| Lines 18/19 | I wonder whether this discussion on the dissolution-reprecipitation should be shifted to the chapter 5.1, where these issues are raised already. |
| | |
| Lines 20/21 | Suggestion: Start sentence with "Accordingly, events of monazite growth may not…" |
| | |
| Line 21 | Suggestion: "if looking at the weighted mean ages only." |
| | |
| Line 21 | Suggestion: "To avoid age mixture" |
| | |
| Line 22 | What is the reason for a 0.5 Ma time interval? And use 0.5 "Myr" (instead of "Ma"), |
| | |
| Line 22 | Does the content in the bracket mean: "see at figure 1 and the appendix"?. Or is there a figure in the appendix (which I have no access to)? |
| | |
| Lines 22/23 | Suggestion: "In a subsequent step…" |
| | |
| Line 23 | What do you mean by "plateau"? Could the "plateau be simply made of several distinct intervals that a time step of 0.5 myr is not sufficient to resolve individual age peaks? Is there geological evidence that would support a plateau due to continuous monazite formation? Can we see this in the BSE images (e.g. continuous colour gradients)? |
| | |
| Line 24 | The authors should clarify, whether their interpretation is based on their ages only, on their ages and the already published ages together, or on the weighted mean ages only. |

From here on, there is mix between all types, which does not enhance the credibility of your interpretation.

Lines 24/25    The statement "to visualize distinct events or phases of tectonic activity (Fig. 8) is difficult to understand. First, the visualization is dependent on your chosen time interval (see comment above). Second, you have to clarify for figure 8, what you do with the gray ages (literature data): Are they included or not? If not, why not? Third, you have to specify the different colours used in figure 8, there are colours of different intensity (in the West area three shades, but no explanation).

Line 25    Suggestion: "are possible to be obtained for each grain"

Line 2

Line 29    "listed" instead of "given"

Line 30    Suggestion: Start sentence with "An alternative reason …"

Line 30    I suggest that the authors speak of a "plateau" if they claim a "spread out age pattern". The nomenclature should be clarified.

Lines 30/31    Suggestion: "tectonic activity by multiple small…"

Line 31    "In such a case …"

Line 31    Do you really mean "reprecipitate" or only "precipitate"?

Line 32    Suggestion: "This may lead to …

Line 32    Can you specify more precisely, what you mean by "unclear crystal zonations"?

Figure 8    In this figure, there are black and gray data. It is obvious that the gray data were not used for the colour coding of the denser and less dense age areas. The authors should clarify on what data set they base their interpretation. Does the inset (with the curves) show the ages of this study only or all data? The authors talk about the "compete data set". What is the complete data set?

What are we expected to see? What is the meaning of the different colour shades (only the West area has three different shades (why?). Howe were they defined (what are the boundaries between the shades)?

Figure caption    Line 4: Suggestion: "frequency" instead of "number". The inset shows a probability density plot, not a histogram.

Line 2    Suggestion: „tend to show sharper zonation"

Line 6    The "monazite stability field" has not been defined. If the authors talk about a "field", there should be at least two parameters that define the stability "field". But I assume that they rather mean a temperature range in which monazite in clefts are formed. It has nothing to do with the thermodynamic stability of monazite.

Line 7    "2.5 to 7 Myr"

Line 7    The bracket should refer to Figs. 5-7 and table 2.

Line 9    In the bracket I would more specifically refer to Fig. 9a and b.

| | |
|---|---|
| Line 10 | If the authors start here to separate another area (the "northeastern" area), the reader wonders why they do not specify such area in figure 2… |
| Line 13 | For the sake of completeness: Tony Hurford (in Hunziker et al. 1992) has produced a zircon FT age for the Splügenpass, not far from your sample locality, which is 20. 3 Ma, There are also AFT ages for this locality in Hunziker et al. (1992, 15.3 Ma) and Rahn (2005, 16.9 Ma). |
| Line 13 | Start new paragraph after "Rubatto et al. 2009)." Start new paragraph with "At the same time…" |
| Line 12-18 | In this discussion you sometimes refer to geochronological data, sometimes to references that quote time intervals for a specific "event". I would argue that such time intervals are also based on geochronologic data. Thus, the authors may as well quote those data or at least refer to the dating method used. The presentation so far is rather inconsistent. |
| Line 14 | The sentence starts with "At that time…" (or my suggested change) but then a time interval of 19-18 Ma is quoted. Is this the same time interval? Is this the relict of a former statement? |
| Line 15 | It is not clear, what the authors mean by "After this". What does "this" refer to? |
| Line 15 | Suggestion: "Afterward, temperature decrease due to exhumation, …" |
| Line 16 | The "Valsertal" is not visible in any of the figures. Thus, a reader not familiar with Swiss geography is lost. My suggestion: "started ar 16 Ma near sample VALS (Figs. 2, 7t) …" |
| Line 18 | Suggestion: "biotite recrystallization ages (Wiederkehr et al., 2009)." |
| Lines19/20 | Suggestion: "after which the record ends around sample VALS where cooling below…" By the way, I refer to the apatite FT age by Rahn (2005), which for this locality is 8.6 Ma. |
| Lines 20/21 | Suggestion: "The sample VALS age range of 16-12 Ma…" |
| Line 21 | I do not understand the "perfectly", in particular if the time interval in line 22 is only "ca.". |
| Line 23 | "within the dome" |
| Line 25 | "Fig. 6m" instead of "Fig. 4m" |
| Line 27 | The statement "as fault gouge ages seem to typically coincide with the end of monazite-(Ce) growth" is rather strange. This should be carefully illustrated. If you have a fault gouge, you are in a process of brittle deformation. But to form cleft monazites you have to be in a brittle deformation state anyway. Is this an important statement? If yes, the authors have to show more evidence for it or more specifically refer to a study that has shown such a coincidence. Here I do not know where "below" is. |
| Line 29 | Start paragraph with "40Ar/39Ar cleft muscovite ages…" |
| Line 30 | Stop sentence after "crystallization" and restart with "Further west…" |
| Line 31 | reduce to "2017, 2019), however, ZFT ages predate …" |
| Line 33 | The statement "suggests slow cooling rates" cannot be assessed by the reader. What is the arguments for slow cooling? |

| | |
|---|---|
| Figure 9 | I like this figure very much. To me, it is the heart of this study. Note that similar to figure 2 (same inset), the frame in the inset does not fit to the chosen map outline. Line 2 of caption: "quoted" instead of "given". Line 2: Since the study area mostly extends in a E-W direction, the term "outer region" is difficult to understand. Metamorphically, the inner region would be in the South towards the Insubric Line, not towards the Simplon Line. |

Line 1    Suggestion: „continued deformation and monazite formation". Is this what you mean?

Line 1    The statement "as the systems closed" is not clear to me. What "systems" are you talking about?

Line 2    The term "the lower end of the closure temperature window" is wrong. The closure temperature sensu Dodson (1973) is the moment when a geochronologic system closes, and for one cooling history, there is only one closure temperature, not a closure temperature window. You may argue that the closure temperature of a geochronologic system may vary depending on the cooling rate (see e.g. Bernet 2009 for the ZFT system), but then you are talking about different cooling histories. Do not mix the term closure temperature with the terms partial retention (He methods) or partial annealing zone (FT methods).

Line 4    double space before "17"

Line 4    Suggestion for bracket: "(VANI 6, Fig. 5a)"

Line 5    The authors make a jump in their discussion from S of the RSF to N of the RSF, which is hard to understand and follow.

Line 9    "Fig. 9c, d"

Lines 14/15    I do not understand this statement. Next to the RSF, there are weighted mean ages as low as 7.2 Ma (sample VANI 5).

Line 15    Suggestion: "Correspondingly, the 12-10 Ma phase also marks…"

Line 16    The first bracket should be "(Figs. 5c, d and 6j)", the second "(Fig. 6k)".

Line 17    Suggestion: along the eastern side of the RSF tend to predate, but still are in agreement with…"

Line 18    Suggestion: "In the vicinity of sample VANI 6 south of the RSF (Figs, 2, 5a), ZFT ages show a scatter …"

Line 20    The statement "leave the hydrothermal stability field …" should be avoided. Fact is that you have no younger monazite. This should not be mixed up with monazite no longer being stable.

Line 20/21    Here you should refer to "Fig. 9e and f"

Line 21    Suggestion to start a new paragraph before "The number"

Line 21    The "clear age patterns within the crystals" remain completely cryptic to me (see comments further above). I do not know, how the authors have sorted out what is a cluster to be combined to a weighted mean age. They have to provide this information in the "Results" chapter to gain credibility for their study.

Line 21    Rather than "staggered", I recommend using the word "stacked" or "clustered".

Lines 24/25    The content in the bracket should be "(Figs 5b, d and 6j-l)"

Lines 27/28    I do not know why the authors here refer to results from the Mont Blanc Massif. The only reason might be that they want to point out that there is a tectonic link by the Rhone Line, which starts in their study area and reached west to the Mont Blanc Massif. If yes, they should make this link. Otherwise, I would suggest deleting the entire sentence. Alternative suggestion: "Overall, the 10-7 Ma time interval …along the extended RSF system, as far as to the Mont Blanc Massif (…"

Line 29    Suggestion to add "Fig. 6l" to the bracket.

Line 31    Add reference after "occurred"

| Line 32 | The ages are "6.4-5-4 Ma". |
|---|---|
| Line 32 | The reference to a "muscovite age" is cryptic. What muscovite has been dated? From the host rock or from clefts or fault gouges? |
| Line 33 | The statement "of the area" is cryptic as well. What area? |
| Line 34 | The statement in this line ("similar…") is not clear. What does this statement tell us? I do not understand your reasoning. Note, this is your last sentence of the discussion. Do not stop with a loose end… |

| Lines 2/3 | Is this first sentence a conclusion out of this study? My suggestions would be to start with: "Hydrothermal cleft monazite-(Ce) provides an important record…" You should not use the word "fissure" here, after having used the word "cleft" only so far… I would have preferred "fissure". |
|---|---|
| Lines 3/4 | Suggestion: "provides a important record of shifting tectonic activity associated with the regional exhumation history." Delete "within the monazite stability field" as this is a term that has not been defined. |
| Lines 5/6 | The statement "that age clusters within individual crystals from a simple exhuming area have a less clear age distribution than samples from fault zone areas" is not clear to me. First of all, you state that the Lepontine Dome has a "complex metamorphic and tectonic history" (see above), thus you should not refer to "a simple exhuming area". Second, this statement, if quoted in the "Conclusions" chapter, should have been prepared in the "Discussion" chapter, but I cannot see such a discussion. |
| Line 6 | Here, the authors talk about "fast exhuming area". What is considered to be "fast"? The authors did not distinguish between slow and fast exhumation before. Why do they do this in the "Conclusions" chapter? |
| Line 6 | Do you really mean "recrystallization"? I realize that the authors have not explained where the monazites are crystallized and where they underwent recrystallization. Therefore, the statement remains cryptic to me. The authors should carefully define the different processes and explain, what pattern they generate and what this means for the age interpretation. |
| Line 7 | If you state "in these areas", the reader does not know where. Try to be more specific. |
| Line 7-9 | These statements have not been discussed in the previous chapter. |
| Line 10 | This is the only place where the authors use the term "$^{232}Th$-$^{208}Pb$ monazite-(Ce)", Why only here? |
| Line 10 | double space before "19" |
| Line 11 | The temperature range given here is not a result of this study. If yes, this should be presented with more clarity in your "Discussion" chapter. |
| Line 12 | double space before "19" |
| Line 12 | The bracket content should be "(Fig. 9)" |
| Line 13 | Suggestion: "Within the Lepontine Dome…" |
| Line 14 | Here you refer to the "eastern Gotthard nappe", but on p. 15, line 17, you refer to the "southern edge" of it… |
| Line 14 | What do you consider to be "slow exhumation" (see also comment further above concerning "fast exhumation")? |
| Line 15 | The sentence here seems to be incomplete, something does not fit, at least I do not understand. |

Line 19    "Swiss National Science Foundation"

Line 21    Suggestion: "in providing monazite-(Ce) material for this study."

References    There are several references listed that are not in the text: Frisch 1979, Frisch et al. 2000, Glotzbach et al. 2010, Keller et al. 2006, Kralik et al. 1992, Putnis 2002, Schmid et al. 1996.

References    There are some type errors, e.g. at Keller et al. 2005 and 2006, check spaces between first names, e.g. with Steiger and Jäger 1977 (What is "S.o."?), with Townsend et al. 2000 (is the name "DAndrea" correct?).

Meinert Rahn, March 10, 2019

---

## Author Comment (AC1) · 15 Jul 2019

Dear editors, dear reviewers,

I would first like to apologize for the chaos concerning the figure numbers and figure references in the text of the first submission, and the understandable confusion and frustration this caused during review of the manuscript. The problem was not due to a last minute rearrangement of the figures and I would like to give a short explanation how it happened:

The figures 3 and 4 were supposed to be continuing over two and three pages, respectively. Unfortunately, I did not notice during my final review before submitting the pdf, that the LaTex command I normally use for this did not work. This resulted in figures 3 and 4 ending up being figures 3 to 7, and pushing back any subsequent figures by three numbers. I noticed the strange figure numbers referenced in the text and eventually ended changing them by hand, but unfortunately did not realize that the problem were not the references to the figures, but the figure numbers themselves. The problem could be solved and the numbers and references are now in order.

All reviewer comments were taken into careful consideration and implemented to the best of our ability. This resulted in a near complete reorganization of the manuscript, with the greater part of the text rewritten or newly added, including changed and new figures. There are now several chapters that discuss the basis of the study approach as well as the way of interpreting the data. One of the new figures combines backscatter images with age data, adds chemical plots, and should thereby make the reasoning behind the age grouping better visible to the reader.

The discussion is now more focussed. The study results are highlighted and the discussion split into two chapters for better readability. One chapter compares hydrothermal monazite dating to thermochronometers, while the other discusses the results in a regional geological context.

We hope that the improved and restructured version of the submitted manuscript meets your aproval and look forward to any further suggestions you may have.

Sincerely yours,
Christian Bergemann

Review of
**Constraining metamorphic dome exhumation and fault activity**
**through hydrothermal monazite-(Ce)**
Christian A. Bergemann, Edwin Gnos, Alfons Berger, Emilie Janots, Martin J. Whitehouse
by Meinert Rahn

The study of Bergemann and co-workers presents 480 single spot ages and 33 weighted mean ages from 19 locations and their cleft monazites within the northern Lepontine Dome (and adjacent to it). These ages are used to decipher the exhumation and tectonic history of the Lepontine dome, as the ages are compared with other geochronological data supposed to represent the Neogene cooling history.

To me, there is no doubt that the provided data are interesting for publication in Solid Earth. However, for the moment the manuscript and submitted material has for the moment several critical shortcomings that I would recommend to fix prior to becoming acceptable, as I consider them critical, if the paper wants to have the impact the presented topic deserves and the general title promises. My major concerns are the following:

1. The title of the manuscript suggests that the monazite data provide new constraints on the tectonic and exhumation history of the Lepontine dome, while the discussion of the data mostly refer your data to already existing constraints of the dome exhumation and Tectonics. As such the focus of the paper is more on methodical aspects of monazite dating (e.g. monazite formation temperatures, relationship to other dating techniques and their closure temperatures).

2. There is throughout the paper a mess with the figure numbers. My assumption is that the authors may have changed these numbers shortly before submission of the manuscript. I invite the authors to check carefully all figure numbers when revising their submitted material. I also note that at several occasions the authors refer to figure 1 in the appendix, which I was unable to locate.

3. For the moment, the chapter ‰Results+ is ultra-short and lacks important information. In your discussion chapter, you tend to describe your results at several places, which should be done in the ‰Results+ chapter. The ‰Results+ chapter should also be used to clarify, what data you will discuss in the ‰Discussion+ chapter and which data will not further be discussed.

4.
**a.**My major concern is that the authors are rather vague with their methodical descriptions. Some of these details should be part of the ‰Introduction+ chapter, of a new methodical chapter or part of the ‰Results+ chapter. Let me summarize this in five points that I would expect the authors to provide more information about: a. The authors talk about the ‰monazite stability field+ (e.g. p. 13, line 3; p. 15, line 6; p. 17, line 20), however, they never discuss, what they mean with ‰stability field+. Note that the authors on p. 12, line 5, talk about ‰disequilibrium+, without clarifying what kind of ‰disequilibrium+ they refer to. I would assume that this is not a ‰thermodynamic stability+, but they rather consider a kind of temperature window, in which the cleft monazites were formed. If correct, it might more correctly speak about the ‰monazite

formation temperature window+. This aspect is important, because in the "Discussion+ chapter you compare the formation of monazite with the closure temperatures of lowtemperature thermochronology methods (which seems to suggest some kind of closure-T for cleft monazites).

**b.** The authors present BSE images for each on the investigated monazite crystals (their figures 3 and 4). However, it remains unclear what the visible colour changes mean within each individual crystals (no chemical data are given except for a few selected elements in the supplementary data file) and how the authors have chosen their analytical spots on these crystals. The only information is that the authors state that they have placed the SIMS spots were placed "according to compositional domains+(p. 3, line 29). Accordingly, we would expect that spots of same colour rings in figures 3 and 4 would always represent areas of same gray colour in the BSE image. This clearly is not the case for e.g.in the DURO1 crystal the yellow spots seem to only roughly follow a lighter lamella, but overlap with darker areas around, in the DUTH2 crystal the orange spots lie within a lighter rim, but spread into the darker centre next to it. The authors have to state clearly their criteria in how to assure that spots are not mixtures between to different generations of monazite formation.

**c.** The authors state that they have avoided measurements next to cracks and holes (p. 15, line 30). This statement is in contradiction to e.g. the red spots in BETT11, the blue spots in VANI6, the red spots in VANI5 etc. I assume that the criteria is more likely defined by the analysis itself showing a deficit in elements rather than the geometric vicinity. The authors have to clarify this issue.

**d.** The authors have to clarify on the basis of which criteria they have chosen the weighted mean ages out of the spot analyses. In Figure 5a (VANI6), it seems obvious that the orange group weighted mean age is formed out of all orange spots. Agewise, however, these spots seem to overlap with the gray spots. So, how have the authors separated between orange and gray? In figure 5b (BETT11), the four red spots show age overlap, but they are not combined to one weighted mean age. Why not? In figure 5c (DURO1), the four blue spots form a weighted mean age, but the gray spot next to it (same age) is not part of it. Why not? I could continue the same way for most of the diagrams in the figures 5 to 7. I am sure that there are good reasons for the authorsq choice of the weighted mean ages, but for the moment, this choice cannot at all be assessed by the reader and looks very arbitrary, not scientifically founded. The authors have to explain to the readership their selection criteria, and for such purpose, it may be needed to better illustrate the different compositional variations among the individual monazite analyses.

**e.** According to figure 1, there are three age groups (with some samples showing more than one). In figures 5 to 7, however, the authors have several samples with more than two weighted mean ages, in figure 8, the three age groups are no longer visible, and in your discussion chapter, you discuss a much finer distinction among the age groups (see also figure 9). We would recommend to the authors to clarify this issue of age groups in an early stage (e.g. in the results chapter and then stick to it throughout the entire discussion chapter. For the moment, the reader gets lost due to the many age groups and the inconsistency between the figures.

**f.** Figure 8 shows the ages again, but in probability density plots. Up to here (in particular in the figures 2 to 7, the reader has gained the impression that single spot

data are clustered to weighted mean ages. Here, however, the authors seem to have split the ages again in single spot ages to form new curves and density plots. The same is true in the "Discussion" chapter on pages 15 and 17: Sometimes, the authors refer to single spot ages and sometimes they refer to weighted mean ages. I do not understand why the authors refer to weighted mean ages at all, if they afterwards selectively use the information that fits best their arguments. The authors have to clarify their strategy in interpreting their results. They have to clarify the meaning of their "weighted mean ages" in that sense. They also have to explain how uncertainties were calculated for the different types of ages.

**g.** Figure 8 shows a kind of clustering of the single spot ages. In this plot the authors also show previous literature data (in gray), but these are not included in their clustering pattern (we do not know, whether this is the case for the curves in the inset below). In Figure 9, however, their interpretation includes all the literature data (e.g. for the Gotthard nappe and the Aar Massif). This is inconsistent. Either you use all data or you do not. The authors have to lay out their strategy on what data are to be interpreted and then stick to it.

**h.** Figure 2 shows nicely how the authors divise their samples into regional groups. However, in the "Discussion" chapter, their division seem to not make sense in many respects as they tend to again subdivide their division. I make two examples: (1) On p. 15, line 19, the authors refer to "the entire (north)eastern region that seem to act differently than the rest of the region. This "sub-region" is not well defined. (2) Figure 2 places sample DUTH6 to the edge of the "Center" region, but in figure 9, this sample rather behaves like the samples in region "West", so why DUTH6 is part of the "Center" area? i. In chapter 5.3, the authors compare their data with data from other thermochronometers. However, this comparison is incomplete in that sense that sometimes ages are quoted, sometimes not, sometimes the authors only refer to the interpretation of the previous workers without referring to the geochronological evidence. This should be done in a more careful, systematic and transparent way. I recommend e.g. that they authors clearly state what time and methodical information they use for their discussion (e.g. they refer to K/Ar ages, ZFT and ZHe ages, but they do not use AFT or AHe ages.

**j.** The "Discussion" chapter starts with an interesting subchapter on hydrothermal monazite crystallisation. This is exactly the information needed to understand methodically the authors' strategies. However, as far as I understand, this chapter is not a "result" but a initially chosen "strategy" on how the monazite ages are to be interpreted (it looks therefore misplaced in the "Discussion" chapter). The authors should somewhere clarify their strategy of the understanding on how monazite is formed.

5. From the title of the paper, the reader expects some new information about exhumation and tectonics within the Lepontine Dome. However, in such respect, the "Discussion" chapter has been disappointing for me. The authors support existing cooling/exhumation paths and tectonic events, but they have no courage to suggest any new "events". I agree that the paper title could be understood as "Confirming metamorphic dome exhumation", and I also agree that the problem with monazite dating is the fact that the ages cannot be related to a temperature value (closure temperature) in contrast to other methods. Nevertheless, I also see potential about

the information of the monazite ages that the authors seems to keep untouched. What e.g. is the function of the Rhone-Rhine line (e.g. in figure 9e, f)? Where do the new results show an extension of previous time windows or a focussing on smaller windows for existing phases of tectonic activity? In the end, the "Discussion" chapter does not seem to provide any new information. Looking through these comments (and the detailed comments below) I would recommend to the authors to thoroughly revise their manuscript (major revisions). For me, there is no doubt that this study would be an excellent contribution to Solid Earth. However, for the moment, publication of the extensive data set would fail to gain credibility among the readers, because so many methical details are only vaguely described and therefore lack credibility.

**Detailed comments:**

The following comments are sorted according to page numbers and text lines (in the pdf provided). They were continuously gathered while reading. Many of them have later been clustered to major comments (see above). With respect to grammar or English style corrections, this reviewer confesses to not be a native English speaking person; accordingly, suggestions concerning grammar and phrasing are to be seen as suggestions. I also highlight, if a comment is thought to be a suggestion (for e.g. clarifying a statement or shortening the text).

Title The authors always refer to "monazite-(Ce)". This is ok as the IMA points out that monazites could be dominated by different REE in their formula. However, the authors never confirm that Ce is the dominating RE element in the formula of their 19 samples. The paper has no compositional data except for those in the supplementary data set. I assume that much of the division of the analysis spots (figures 3 and 4) is due to compositional arguments. However, the reader has no chance to assess this division. The reader has not even a chance to find out, whether the "monazite-(Ce)" in the title is correct or not.

Title Looking back to the "Discussion" chapter, this reviewer has rather the impression as if the authors do not "constrain" but only confirm previous information on the exhumation and tectonic history of the Lepontine Dome. I suggest to thoroughly revise the manuscript and then decide on whether to change the title of the paper. The same might apply for the "Abstract".

Title The term "hydrothermal monazite-(Ce)" is varied in the manuscript. Sometimes, the authors refer to "cleft monazite-(Ce)", sometimes to "hydrothermal cleft monazite-(Ce)", and the reader gets puzzled about the different expressions. It would help to clearly state somewhere in the manuscript that these monazites all come from clefts. I am not so happy about the term hydrothermal as it suggests that these monazites have formed in a flush of "hot" fluid, the temperature of which might not have been in equilibrium with the surrounding rock. If so, any later comparison with closure temperatures of thermochronologic systems may not be useful, as the temperature of the monazite formation may have been completely different from the temperature controlling closure of other geochronologic methods.

Line 2 The authors refer to the "Central Alpine Lepontine metamorphic dome", which is a rather unusual term. First, figures 1, 2 and 9 show that the study area does not cover the south of the dome (along the Insubric Line), but its entire E-W extension

(so, "Central" may not be needed). Second, the Lepontine Dome is an Alpine structure, so, I would suggest that there is no need to use the term "Alpine" here.

Changed

Line 4 Suggestion: delete "The" at beginning of sentence

Deleted

Line 5 There seem to be a double space between "between" and "19"

This was due to an error in the command for ~, that was now replaced in the entire document

Line 6 The authors use the abbreviation "Ma" (= mega annum) for both, specific moments in time and timer periods. Please check the instructions for authors that suggest to use "Ma" for time spots", however, "Myr" for time periods. I fully support such instruction. Accordingly, the authors should write here "lifetime of 2 to 7.5 Myr"

This was changed

Line 6 Suggestion to put "combined with age distribution" into brackets.

Changed

Line 9 Suggestion: "In the east and south of the northern Lepontine dome"

Changed to "In the north-east and south-west of the Lepontine dome"

Line 9 Suggestion: delete "the" before "units"

Deleted

Line 9 Suggestion: add "respectively," after "Ma"

Added

Line 10 Suggestion: "at 15-10 Ma. Cleft monazites…"

Changed

Lines 10/11 There seem to be a mismatch with the statement "youngest" in line 10 and "a last phase" in line 11. "youngest" means "last phase", thus these phase should be the same… Furthermore, I would suggest using the term "age signal"

instead of "phase" as the latter already suggests that this has been a distinct tectonic or thermal "event".

"Youngest" was changed to "younger". "Phase" was changed to "age group"

Line 11 Suggestion: "along the Simplon Fault". One further note: If you quote names of domes and faults, make sure that they are constantly with upper case or lower case. So, either Lepontine Dome or Lepontine dome, Simplon Fault or Simplon fault.

Lines 11/12 Add hyphen: "strike-slip"

Added

Line 12 Suggestion: "faults along the Rhine and Rhine-Rhone faults"

Line 13 Do you mean "stability" or "formation"?

Line 13 "directly" instead of "direct"

Changed

Line 15 "experience" instead of "experienced"

Changed

Line 16 Suggestion: Start line with "For the Lepontine Dome, this evolution is an interplay õ "

Changed

Line 16 Suggestion: "motion along" instead of "activity of"

Changed

Line 16 Suggestion: "define" instead of "dominate"

Changed

Line 17 Suggestion: "the western edge of the dome" instead of "the western parts of the area".

Changed

Note that throughout the text, you frequently use "in this area", but you are not very specific, what area exactly you are referring to. I would expect to get some references here.

Line 17 The statement "Although much of the (thermo)chronological history of the area is well known" is rather cryptic. Does this refer to the fact that there are a lot of data concerning the post-peak metamorphic history? Or does this mean that the tectonic history is known in detail? Since you do not provide any references and the statement is rather broad, the reader cannot assess the meaning of this statement.

Line 18/19 Interestingly, here you refer to AFT and AHe ages, while in the Discussion chapter, you do not refer to any of the existing studies including such data and you do nowhere cite any of them. This is ok with me (as e.g. AFT and AHe ages may already be influenced by topographic evolution. Nevertherless, there is a mismatch between this statement and the discussion of your data afterwards. Finally one suggestion for a rewording: "existing cooling ages of the Rb-Sr, fission track (FT) and (U-Th)/He systems".

Good point, this was left over from an earlier version. The references were taken out for exactly that reason, the sentence has been changed now to avoid confusion.

Line 20 Suggestion: "(e.g. Parrish, 1990). It is highly resistant…"

Changed

Line 21 "by diffusion" instead of "through diffusion"

Changed

Line 22 not clear, what "geologically reactive" means

Changed to "Nonetheless, monazite remains reactive after crystallization, as it can experience dissolution-recrystallization facilitated through hydrous fluids..."

Line 1 "by" instead of "through".

Changed

Line 1 What do you mean by "mediation"?

Changed to "facilitated through", meaning that the monazite remains susceptible to these changes as long as the cleft remains fluid filled and within a certain temperature range.

Line 3 Suggestion: "occasionally contain monazite-(Ce). They represent voids" +

Changed

Line 7 Suggestion: "were" instead of "could be" +

Changed

Line 11 Suggestion: "to tectonic activity. Accordingly, fissures and clefts are " +

Changed

Line 13 "interacted" instead of "interacts" +

Changed

Line 13 Suggestion: Start new sentence with "Dissolution and precipitation" +

Line 14 Suggestion: "led to" instead of "causing" +

Changed

Line 18 Suggestion: "by using secondary ion mass spectrometry (SIMS)" +

Changed

Line 19 At the end of this introduction, I miss a paragraph that presents the aims of this study. I addition, I miss an introduction to the methodology of monazite dating, not the technical issues, but the prerequisites that define the strategy of the study.

Such a paragraph was added and further explanations added in a later chapter.

Line 22 Add "European" to "Alps"

Added

Line 23 This bracket lists examples of metamorphic domes, but their allocation is once to a country ("Austria") and once to the part of an orogeny ("western Alps"). I would suggest referring either to Switzerland/Austria or to eastern and western Alps.

Changed to "Eastern Alps"

Line 25 The references listed here have a strange ordering, either alphabetic or with decreasing age. Please check the author's instructions.

This was adjusted

Line 26 The statement that the Western and Central Alps had a "complex tectonic and metamorphic history" is not very convincing. "Complex" with respect to what? Orogens tend to be complex anyway (fortunately, this keeps us going to find out more details of this history!), but the statement is too vague, too general. You may start the chapter with such a statement.

Furthermore, I refer to the detail that Schmid et al. (2004) did not use the term "Central Alps", but divided the Alps into western and eastern Alps.

True, there is no consistent terminology in this case. If consulting for example A. O. Pfiffner's "Geology of the Alps" (2014) there is the division into Western, Central and Eastern Alps.

Line 31 You may put "in excess of 650°C in some regions" into brackets.

Done

Line 1 not clear, what you mean by "staggered exhumation". I am doing research in the field of exhumation for 20 years, but have never heard of this expression.

Staggered was used here in its meaning of one process starting somewhat later than the other. To avoid confusion, it was replaced by "exhumation starting first in the Ticino and then the...".

Line 1 I would recommend to add references to this statement

Line 4 Suggestion: "later in time at 18-15 Ma

Changed

Line 5 I would recommend to use the term "normal faulting", rather than "detachment", as the term "detachment" has no direction.

Changed

Line 9 Here, the term "Simplon Fault zone" is used, but in line 11 "Simplon shear zone". If these terms describe different zones, then the authors should explain the difference.

Line 10 "Alpine" instead of "alpine"

Changed

Line 14 From figures 1 and 2 it is evident that the study area is the northern Lepontine dome. Nevertheless, the exhumation of the Lepontine Dome cannot be described without the Insubric Line. (15 km of differential vertical exhumation!). The authors should consider including the Insubric Line in their "Geological setting".

A sentence was added here

Line 15 "area" instead of "Area"

Changed

Line 16 Suggestion: "The study area comprises the northern half of the Lepontine õ "

This only true for the eastern part of the Lepontine, while the west is completely covered, leaving out the south and south-west. If it is preferred, the statement could be changed to "... the western and northern parts of..."

Line 17 "Forcola Fault"

Changed

Line 17 The "Val d'Ossola" is not visible in any of the figures.

The place name was removed from the text to avoid cluttering on the map.

Line 18 Suggestion: "Aar massif to the north (see Fig. 1 for the tectonic position of the samples)õ "

Changed

Line 19 Should not Janots et al. 2009 be added to this list, as it also contains monazite ages?

Janots et al. 2009 was omitted, since it contains ages of metamorphic monazite instead of hydrothermal. However, due to changes in the text later on to avoid confusion, the references to Janots et al 2012 and Bergemann et al 2017 were removed here, since the samples are discussed, but sampling sites lie outside the Lepontine dome.

Line 20 The statement that the four groups correlate "to tectonic subdivisions" cannot fully be assessed by the reader: First, there is no tectonic subdivision between central and western part (the Ticino and Toce domes are separated by what tectonic feature?). Second, In figures 1, 2 and 9, the Forcola Fault ends south of the study area, and if the authors refer to the extension of the Forcola Fault to the north, the sample VALS would be located in the central part. How about separating along the western rim of the Adula nappe?

Changed

Line 22 "bound" instead of "bounded", as it is related to "to bind"

Changed

Lines 23-25 I would shift the last sentence to the beginning of the next chapter.

The sentence was moved

Line 28 Note that there are different spellings of "backscatter". In figures 3 and 4, you write "back-scatter".

Changed in all cases to "backscatter"

Line 28 Suggestion: "Backscatter electron (BSE) images were used to define spots suitable for analytical investigations." It is, however, unclear to me, how you defined the different groups (of different colours) and how you have chosen the analytical spots. This needs more information.

Explanations were added

Line 29 The statement "according to compositional domains" cannot be understood without further information: How do you define a compositional domain? What compositional data were available? I refer to the fact that a change in RE element (but otherwise constant composition of the monazite) may not really show a difference in colour in the BSE image.  It would be helpful to know what analytical data were available, and which of these data were used to "define the compositional domains".

This was an error, the sentence should have stated "according to visible domains" and has been corrected. These should represent differences in concentration of the heavier elements.

Line 30 Suggestion: "As far as possible, spot measurements next to cracks or holes were avoided".

Changed

Line 30 Looking at e.g. figures 5a, 5j, or 5l, this statement cannot be understood. In the corresponding crystals, there are measurement spots directly next to cracksõ

That is unfortunately, despite best efforts, not always avoidable. The grains are gold covered for the SIMS measurements, due to the large grain size not visible in their entirety when placing the measurement spots, and viewed at an angle due to the machine construction. Smaller cracks and inclusions may therefore be covered up and orientation points lie outside the visible area during measurement spot placement. If the data appears undisturbed, the measurement is still used.

Line 31 Suggestion: %n such areas+

Changed

Figure 1 This figure is not yet finished. First and foremost, there are no coordinates, there is no scale and there is no north direction. I would recommend to enlarge the map on top (a), and enlarge the labels (b) to (e). The legend contains the abbreviation %nzt+ that is not explained in the caption. According to the thick black line in map (a), the profile (c) should end at the border to the Aar Massif, but in (c), it extends into it. Profile (d) has gray units that do not show up in the map (a). Each of the profiles has its own scale, thus, scales should be added to figure parts (b) to (e). Faults are not labelled. One methodical problem is that the authors in this figures legend refer to three age groups, but in the discussion chapter (and in figure 9), there are more age groups. This is inconsistent. Do you need to mark the different groups in this figure already?

The requested changes were made and the reference to age groups taken out.

Figure caption
Line 1: switch the two references, as they are not in correct order, line 4: There is no Wiederkehr et al. 2008 reference in your reference list. Do you mean the 2009 reference? Line 4: Why does the caption uses a different text for profile (e)?

References were switched, the caption text for (e) was made identical to the others.

Table 1 In this table, all samples are given names with upper case spelling. However, in figures 1 to 7, the authors sometimes use lower case or upper case. In addition, sometimes, there is a gap between name and number, sometimes not. I recommend adapting the names throughout the paper and figures and tables, and I would use a space between name and number, as several names end with %t, which then might be interpreted as a %ot. For some of the samples, you note coordinates with %ot. What

is the difference? If the location of the sample is not sure; I would rather recommend to reduce the digits after the comma of the angle minutes (e.g. sample SALZ2) rather than adding a ‰.

The sample names were standardized to upper case with a space between name and number.

Figure 2 The frame in the inset does not fit to the area of the large map. Similar to figure 1, coordinates and north direction are missing, the abbreviation "mzt" should be explained, The label of the "Centovalli Fault" is partly covered with blue colour.

Line 4 Suggestion: "by" instead of "with"

Changed

Line 7 delete "presented"

Changed

Line 8 Suggestion: "reported" instead of "given"

Changed

Lines 10/11 Suggestion: "… pending). Table 2 provides an overview and Figures 3 and 4 show measurement positions and the division of the analysis spots into different age groups, represented by different colours.

Changed

Lines 11/12 The statement "As there are difficulties with the U-Pb system for hydrothermal monazite-(Ce)" is very cryptic. What difficulties do you mean? And how have the authors dealt with these difficulties?

Explanations were added to the text.

Line 13 "see discussion in chapter 5.2".

Line 13 I note that the "Results" chapter only contains these four lines. I strongly recommenddescribing the results. What should the reader see? The next pages are figures 3 to 7and table 2, and if the reader goes through these many data, there is no guarantee that the reader will come to the same conclusions as the authors about what is important and what is not. As noted in the general comments above, I have rather collected doubts about the usefulness of the chosen strategy of the authors and the credibility in their chosen analytical spots, derived weighted mean ages etc. I would argue that a careful description of the results would help much not losing the reader half way. You should add a description on how the spots have been selected,

grouped in the diagrams of figures 5 to 7, how the weighted mean ages were afterwards grouped. It may help starting with figure 8 instead that shows an overview of all ages.

The results chapter was expanded and the figures combined into a larger figure which now also includes a chemical plot for each sample.

Figure 3 Many of the colours of the spots are hard to be distinguished (e.g. figure e). All sample names should be similar to tables 1 or 2. For several sub-figures, the numbers are too small to read (perhaps enlarge figure to full page). You should clarify that this BSE images show all spots made for this study.

A remark that the images show all spots made for the study was added.

Figure caption "Back-scatter" or "Backscatter"? (see comment to page 3, line 28). You have to add the information that the grains shown are from the south and west areas only, e.g. "Backscatter electron images of all studied cleft monazite-(Ce) grains from the South and West areas." Line 1: "content" instead of "contents", "Coloured ovals" instead of "Spots".

Changed

Line 2: "different colours indicate different chemical …". Lines 2/3: "indicates those data, for which a weighted mean 208Pb/232Th age could be calculated.

Changed

Figure 4 For this figure, see comments to figure 3 and figure 3 caption. Again, it should be clarified in the caption that these samples are from the Center and East areas.

The same changes as former figure 3 wered made.

Table 2 The caption of this table does not fit. The information given here should be placed elsewhere. Most of the numbers in the "Figure" column are wrong. Obviously, the authors have changed the figure numbers in a late step and not adapted text and tables.
 The last column to the right lists the range of single grain ages. I do not understand why there are empty lines in this column as the range can also be given to those age groups that are combined to a weighted mean age.

The age range given does not correspond to the age groups, but is of the entire sample. To avoid confusion this column was moved.

 For sample %GRAESER1+, the number of points for the older weighted mean age should be 6 instead of 5 (see figure 3e), and the second weighted mean age is missing in figure 5e.

Changed

I would recommend to draw lines between the four different areas (South, West, Center, East).

Page 10-12

Figures 5-7 Looking at these diagrams, I have many questions about how you define a weighted mean age. In some cases, all spots of the same colour form such a weighted mean age (e.g. 5a, 5b, 5g). More frequently, the weighted mean ages are calculated for a selection of ages of the same colour only (see e.g. 5b, 5e). In some cases, it is obvious that only one colour bar iw far off (e.g. in 5e, red), but in other cases (e.g. 5b, red), I do not know, why one age has not been included, even though the vertical bars overlap. Finally, there are examples, where age bars overlap, but were not used to calculate a weighted mean age (e.g. in 5e, young yellow ages, in figure 7s, gray bars in the figure centre). The authors have to explain their selection procedure in detail, otherwise there is no credibility to their ages at all.

Explanations were added to the text and the figures combined into a larger figure which now also includes a chemical plot for each sample.

In figure 6p, one vertical bar has a lighter blue colour; why?, In figure 7s to the left, there is a short bar without colour.

Changed

Line 2 To me the content of chapter 5.1 is misplaced. The reasoning given here is not the result of the study, but a strategy that the authors have chosen beforehand, as far as I understand. If there are observations, features, analyses that support your reasoning, then they should be presented as results and used to interpret your ages properly. For the moment, the content of this chapter should be transferred elsewhere, e.g. in a new chapter on methodology.

In light of the comments of both reviewers, this seems like a good idea. Accordingly, chapters 5.1 and 5.2 were moved to the methodology section and combined to give a better introduction into the approach used.

If you argue that "later reactions may be aided by secondary porosity and fracturing induced by the previous dissolution-reprecipitation/recrystallization events", then show evidence for it. Show what you mean, add another figure like figs 3 or 4 to show corresponding features. For the moment, the entire chapter 5.1 is a black box to the reader, as the authors do not present any evidence for dissolution, reprecipitation, recrystallization. The authors should better define, what they mean with these processes and present examples for it.

The chapters were moved, reorganized and expanded

Line 5 Suggestion: "later reactions may occur"

Changed

Line 5 The term "disequilibrium" is hard to understand. "Disequilibrium" with respect to what?

The sentence was changed to state "… brought into chemical disequilibrium with the hydrothermal mineral phases and the surrounding wall rock."

Line 5 Suggestion: "The may be caused by a tectonic event for a …"

Changed to "These may be..."

Line 10 Suggestion: "phase may precipitate"

Changed

Line 12 Suggestion: "fluid remains connected to a …"

Changed

Line 12 Suggestion: Start sentence with "A dissolution …" as this process has not been defined.

Changed, and the previous sentence was moved to follow this one.

Line 2 There is no 2018 reference for Grand¢Homme et al. in the reference list. Do you mean the 2016 reference?

The reference was adde to the bibliography

Line 8 The statement "placed … according to growth domains" cannot be assessed by the reader. The authors should show their arguments, e.g. with one example. How

do you define a growth domain? How do you define what is old, what is younger? Do ages from different growth domains fit to the measured ages (same age order)? Have you tested the order?

Line 9 The statement "on the basis of chemical composition" cannot be assessed by the reader. What chemical information did the authors gather and what chemical arguments did they use? This information is of major importance if forthcoming studies should apply the same methodology.

Line 11 The reference to "Fig. 7" should probably be "Fig. 5-7".

Explanations were added to the text and the figures combined into a larger figure which now also includes a chemical plot for each sample.

Lines 11/12 The statement "It appears that dissolution-precipitation may largely preserve the chemical composition of an affected crystal part" cannot be assessed by the reader. Is this an assumption? If not, what is the evidence you found? What are the conclusions drawn from it?

References were added, An explanation on the implications of this was added.

Line 12 "areas of" instead of "areas with"

Changed

Line 13 Not clear, what you mean by "this" ("Despite this, …")

Changed

Lines 13/14 The statement "only in a few, clear cases" cannot be assessed by the reader. Which cases? And what means "clear"?

The text was adjusted

Line 14 Suggestion: "age calculation, to avoid …"

Changed

Line 15 Suggestion: "single" instead of "distinct"

Changed

Line 18 Suggestion: "New growth on an existing crystal results in sharp chemical (or colour) boundaries between growth zones." Note however, that this is only the case, if the growth zones differ in composition. Thus, it is an argument, but it does not need to be…

This is indeed only the case if the growth zones differ in composition. It is, however, safe to assume that the fluid composition will change at least slightly in almost every case.

Lines 18/19 I wonder whether this discussion on the dissolution-reprecipitation should be shifted to the chapter 5.1, where these issues are raised already.

Moved

Lines 20/21 Suggestion: Start sentence with "Accordingly, events of monazite growth may not…"

Changed

Line 21 Suggestion: "… looking at the weighted mean ages only."

Changed

Line 21 Suggestion: "To avoid age mixture"

Changed

Line 22 What is the reason for a 0.5 Ma time interval? And use 0.5 "Myr" (instead of "Ma"),

Changed to Myr

Line 22 Does the content in the bracket mean: "see at figure 1 and the appendix"?. Or is there a figure in the appendix (which I have no access to)?

Lines 22/23 Suggestion: "In a subsequent step…"

Changed

Line 23 What do you mean by "plateau"? Could the plateau be simply made of several distinct intervals that a time step of 0.5 myr is not sufficient to resolve individual age peaks? Is there geological evidence that would support a plateau due to continuous monazite formation? Can we see this in the BSE images (e.g. continuous colour gradients)?

This argument can be made for all dating techniques applied to any type of mineral grain that underwent stepwise crystallization. With respect to the data presented here, when considering the precision of the individual measured ages an interval size of 0.5 Myr is reasonable. A smaller interval size would indeed likely result in the identification of more age peaks, which would, however, suggest a higher accuracy

than supported by the precision of the data. While it is likely that such smaller events exist, as also argued in the text, they cannot be resolved in time with the currently availabe techniques. Currently, the only way to identify phases that closely followed one another, is through changes in the crystal chemistry.

Line 24 The authors should clarify, whether their interpretation is based on their ages only, on their ages and the already published ages together, or on the weighted mean ages only. From here on, there is mix between all types, which does not enhance the credibility of your interpretation.

This was adjusted

Lines 24/25 The statement "to visualize distinct events or phases of tectonic activity (Fig. 8) is difficult to understand. First, the visualization is dependent on your chosen time interval (see comment above). Second, you have to clarify for figure 8, what you do with the gray ages (literature data): Are they included or not? If not, why not? Third, you have to specify the different colours used in figure 8, there are colours of different intensity (in the West area three shades, but no explanation).

This was adjusted

Line 25 Suggestion: "are possible to be obtained for each grain"

Changed

Line 29 "listed" instead of "given"

Changed

Line 30 Suggestion: Start sentence with "An alternative reason õ "

Changed

Line 30 I suggest that the authors speak of a "plateau" if they claim a "spread out age pattern". The nomenclature should be clarified.
Lines 30/31 Suggestion: "tectonic activity by multiple smallõ "

Changed

Line 31 "In such a case õ "

Changed

Line 31 Do you really mean "reprecipitate" or only "precipitate"?

Both, changed to "(re-)precipitate".

Line 32 Suggestion: "This may lead to …

Changed

Line 32 Can you specify more precisely, what you mean by "unclear crystal zonations"?

This was adjusted

Figure 8 In this figure, there are black and gray data. It is obvious that the gray data were not used for the colour coding of the denser and less dense age areas. The authors should clarify on what data set they base their interpretation. Does the inset (with the curves) show the ages of this study only or all data?

Both, figure and inset, show only the data of this study, as the literature data is from the Aar Massif and the interior Gotthard Nappe and was only included for additional information. After careful consideration, the references and data points were removed from this figure.

The authors talk about the "compete data set". What is the complete data set? What are we expected to see?

"Complete" was deleted to avoid confusion

What is the meaning of the different colour shades? Howe were they defined (what are the boundaries between the shades)?

An explanation has been adde to the figure caption: "Darker colors represent peaks or plateaus, that indicate individual events or phases of monazite formation/alteration due to tectonic activity. Lighter shades indicate fewer ages recorded, due to either only reduced tectonic activity or mixing ages."

(only the West area has three different shades (why?).

This was a mistake and has been corrected to two shades

Figure caption
Line 4: Suggestion: "frequency" instead of "number". The inset shows a probability density plot, not a histogram.

Changed

Line 2 Suggestion: "tend to show sharper zonation"

Changed

Line 6 The "monazite stability field" has not been defined. If the authors talk about a "field", there should be at least two parameters that define the stability "field". But I assume that they rather mean a temperature range in which monazite in clefts are formed. It has nothing to do with the thermodynamic stability of monazite.

Line 7 "2.5 to 7 Myr"

Changed

Line 7 The bracket should refer to Figs. 5-7 and table 2.

Changed

Line 9 In the bracket I would more specifically refer to Fig. 9a and b.

Changed

Line 10 If the authors start here to separate another area (the "northeastern" area), the reader wonders why they do not specify such area in figure 2…

Change to "eastern and central"

Line 13 For the sake of completeness: Tony Hurford (in Hunziker et al. 1992) has produced a zircon FT age for the Splügenpass, not far from your sample locality, which is 20. 3 Ma, There are also AFT ages for this locality in Hunziker et al. (1992, 15.3 Ma) and Rahn (2005, 16.9 Ma).

Added to the text and in a new figure

Line 13 Start new paragraph after "Rubatto et al. 2009)." Start new paragraph with "At the same time…"

Line 12-18 In this discussion you sometimes refer to geochronological data, sometimes to references that quote time intervals for a specific "event". I would argue that such time intervals are also based on geochronologic data. Thus, the authors may as well quote those data or at least refer to the dating method used. The presentation so far is rather inconsistent.

The discussion was reorganized

Line 14 The sentence starts with "At that time" (or my suggested change) but then a time interval of 19-18 Ma is quoted. Is this the same time interval? Is this the relict of a former statement?

Sentence was changed

Line 15 It is not clear, what the authors mean by "After this". What does "this" refer to?
Line 15 Suggestion: "Afterward, temperature decrease due to exhumation, "

Changed to "After this time"

Line 16 The "Valsertal" is not visible in any of the figures. Thus, a reader not familiar with Swiss geography is lost. My suggestion: "started ar 16 Ma near sample VALS (Figs. 2, 7t) "

Changed

Line 18 Suggestion: "biotite recrystallization ages (Wiederkehr et al., 2009)."

Changed

Lines19/20 Suggestion: "after which the record ends around sample VALS where cooling below" y the way, I refer to the apatite FT age by Rahn (2005), which for this locality is 8.6Ma.

Changed

Lines 20/21 Suggestion: "The sample VALS age range of 16-12 Ma"

Changed

Line 21 I do not understand the "perfectly", in particular if the time interval in line 22 is only "ca.".

"perfectly" was taken out

Line 23 "within the dome"

Changed

Line 25 "Fig. 6m" instead of "Fig. 4m"

Line 27 The statement "as fault gouge ages seem to typically coincide with the end of monazite-Ce) growth" is rather strange. This should be carefully illustrated. If you have a fault gouge, you are in a process of brittle deformation. But to form cleft monazites you have to be in a brittle deformation state anyway. Is this an important statement? If yes, the authors have to show more evidence for it or more specifically

refer to a study that has shown such a coincidence. Here I do not know where "below" is.

Line 29 Start paragraph with "40Ar/39Ar cleft muscovite ages…"

Changed

Line 30 Stop sentence after "crystallization" and restart with "Further west…"

Changed

Line 31 reduce to "2017, 2019), however, ZFT ages predate …"

Changed

Line 33 The statement "suggests slow cooling rates" cannot be assessed by the reader. What is the arguments for slow cooling?

A chapter on this was added

Figure 9 I like this figure very much. To me, it is the heart of this study. Note that similar to figure 2 (same inset), the frame in the inset does not fit to the chosen map outline.

The inset frame was adjusted

Line 2 of caption: "quoted" instead of "given".

Changed

Line 2: Since the study area mostly extends in a EW direction, the term "outer region" is difficult to understand. Metamorphically, the inner region would be in the South towards the Insubric Line, not towards the Simplon Line.
Changed to " Note the shift over time from the southern and eastern regions of the Lepontine dome to the central and western areas and finally to the areas close to the shear zones bounding its western limit."

Line 1 Suggestion: "continued deformation and monazite formation" Is this what you mean?
Line 1 The statement "as the systems closed" is not clear to me. What "systems" are you talking about?
Line 2 The term "the lower end of the closure temperature window" is wrong. The closure temperature sensu Dodson (1973) is the moment when a geochronologic system closes, and for one cooling history, there is only one closure temperature, not a closure temperature window. You may argue that the closure temperature of a

geochronologic system may vary depending on the cooling rate (see e.g. Bernet 2009 for the ZFT system), but then you are talking about different cooling histories. Do not mix the term closure temperature with the terms partial retention (He methods) or partial annealing zone (FT methods).

This part of the discussion was rewritten

Line 4 double space before "17"

Adjusted

Line 4 Suggestion for bracket: "(VANI 6, Fig. 5a)"

Line 5 The authors make a jump in their discussion from S of the RSF to N of the RSF, which is hard to understand and follow.

The discussion was changed

Line 9 "Fig. 9c, d"

Lines 14/15 I do not understand this statement. Next to the RSF, there are weighted mean ages as low as 7.2 Ma (sample VANI 5).

VANI 5 is located in the foot wall and not in the hanging wall

Line 15 Suggestion: "Correspondingly, the 12-10 Ma phase also marksõ "

Changed

Line 16 The first bracket should be "(Figs. 5c, d and 6j)", the second "(Fig. 6k)".

Line 17 Suggestion: along the eastern side of the RSF tend to predate, but still are in agreement withõ "

Changed

Line 18 Suggestion: "In the vicinity of sample VANI 6 south of the RSF (Figs, 2, 5a), ZFT ages show a scatter õ "

Changed

Line 20 The statement "leave the hydrothermal stability field õ" should be avoided. Fact is that you have no younger monazite. This should not be mixed up with monazite no longer being stable.

Changed

Line 20/21 Here you should refer to "Fig. 9e and f"

Line 21 Suggestion to start a new paragraph before "The number"

Changed

Line 21 The "clear age patterns within the crystals" remain completely cryptic to me (see comments further above). I do not know, how the authors have sorted out what is a cluster to be combined to a weighted mean age. They have to provide this information in the "Results" chapter to gain credibility for their study.

More detailed explanations concerning the dating of hydrothermal monazite were added at an earlier stage of the manuscript.

Line 21 Rather than "staggered", I recommend using the word "stacked" or "clustered".

Changed

Lines 24/25 The content in the bracket should be "(Figs 5b, d and 6j-l)"

Lines 27/28 I do not know why the authors here refer to results from the Mont Blanc Massif. The only reason might be that they want to point out that there is a tectonic link by the Rhone Line, which starts in their study area and reached west to the Mont Blanc Massif. If yes, they should make this link. Otherwise, I would suggest deleting the entire sentence. Alternative suggestion: "Overall, the 10-7 Ma time interval õ along the extended RSF system, as far as to the Mont Blanc Massif (õ "

Changed

Line 29 Suggestion to add "Fig. 6l" to the bracket.

Line 31 Add reference after "occurred"

Line 32 The ages are "6.4-5-4 Ma".

Changed

The part of the manuscript referred to by the following comments was rewritten to be more focused and thus better understandable.

Line 32 The reference to a "muscovite age" is cryptic. What muscovite has been dated? From the host rock or from clefts or fault gouges?

Line 33 The statement "of the area" is cryptic as well. What area?

Line 34 The statement in this line ("similarõ ") is not clear. What does this statement tell us? I do not understand your reasoning. Note, this is your last sentence of the discussion. Do not stop with a loose endõ

Lines 2/3 Is this first sentence a conclusion out of this study? My suggestions would be to start with: %Hydrothermal cleft monazite-(Ce) provides an important recordõ + You should not use the word %fissure+ here, after having used the word %cleft+ only so farõ I would have preferred %fissure+.

Lines 3/4 Suggestion: %provides a important record of shifting tectonic activity associated with the regional exhumation history.+ Delete %within the monazite stability field+ as this is a term that has not been defined.

Lines 5/6 The statement %that age clusters within individual crystals from a simple exhuming area have a less clear age distribution than samples from fault zone areas+ is not clear to me. First of all, you state that the Lepontine Dome has a %complex metamorphic and tectonic history+ (see above), thus you should not refer to %a simple exhuming area+. Second, this statement, if quoted in the %Conclusions+ chapter, should have been prepared in the %Discussion+ chapter, but I cannot see such a discussion.

Line 6 Here, the authors talk about %fast exhuming area+. What is considered to be %fast+? The authors did not distinguish between slow and fast exhumation before. Why do they do this in the %Conclusions+ chapter?

Line 6 Do you really mean %recrystallization+? I realize that the authors have not explained where the monazites are crystallized and where they underwent recrystallization. Therefore, the statement remains cryptic to me. The authors should carefully define the different processes and explain, what pattern they generate and what this means for the age interpretation.

Line 7 If you state %in these areas+, the reader does not know where. Try to be more specific.

Line 7-9 These statements have not been discussed in the previous chapter.

Line 10 This is the only place where the authors use the term %232Th-208Pb monazite-(Ce)+, Why only here?

Line 10 double space before %19+

Line 11 The temperature range given here is not a result of this study. If yes, this should be presented with more clarity in your %Discussion+ chapter.

Line 12 double space before %19+

Line 12 The bracket content should be %(Fig. 9)+

Line 13 Suggestion: %Within the Lepontine Domeõ +

Line 14 Here you refer to the %eastern Gotthard nappe+, but on p. 15, line 17, you refer to the %southern edge+ of itõ

Line 14 What do you consider to be "slow exhumation" (see also comment further above concerning "fast exhumation")?

Line 15 The sentence here seems to be incomplete, something does not fit, at least I do not understand.

Line 19 "Swiss National Science Foundation"

Changed

Line 21 Suggestion: "in providing monazite-(Ce) material for this study."

Changed

References There are several references listed that are not in the text: Frisch 1979, Frisch et al. 2000, Glotzbach et al. 2010, Keller et al. 2006, Kralik et al. 1992, Putnis 2002, Schmid et al. 1996.
References There are some type errors, e.g. at Keller et al. 2005 and 2006, check spaces between first names, e.g. with Steiger and Jäger 1977 (What is "s.o."?), with Townsend et al. 2000 (is the name "DAndrea" correct?). Meinert Rahn, March 10, 2019

Changed

Review of Bergemann, Gnos, Berger, Janots and Whitehouse, Solid Earth Ms. Feb 2019:
Constraining metamorphic dome exhumation and fault activity through hydrothermal monazite-(Ce).
Reviewer: Fraukje Brouwer, VU Amsterdam, Netherlands, 11 March 2019

**General assessment**

The manuscript presents and extensive new dataset of cleft monazite ages that are an important addition to exiting geochronological work in the Alps. In addition, the study presents an interesting analysis of the relationship between the duration of tectonic events and the spread in ages recorded in individual monazite crystals. The paper certainly falls within the scope of Solid Earth, but has significant shortcomings in its presentation and therefore **I recommend that it undergoes major revision before being accepted for publication in Solid Earth**.

The manuscript in its present state has three major shortcomings:

1) The data are presented and grouped in multiple ways that are not always clarified to the reader, which makes it impossible for the reader to judge whether the interpretations are sound.

2) The figure numbers appear to have been switched around several times during the preparation of the manuscript leaving many incorrect references, including a on-existing figure in the electronic supplement, making it nearly impossible to find the correct data.

3) Section 5.3 is not clearly argued and organised and needs to be revised to clarify the reasoning of the authors. The abstract suggests the results of the study include major new findings, but if fact the results mostly confirm existing age information. To me, the value of the paper is more in the applicability of cleft monazite ages and the different expression of faster and slower tectonic processes in this dataset.

Note: This review was performed after the review of Dr. M. Rahn became available. I have tried to avoid duplication. I agree with most of his comments and suggestions. Numbers between brackets below (1) are marked in the appended annotated manuscript.

**Specific comments**

Throughout: figure numbers and references to them are a mess throughout the manuscript. This needs thorough checking.

The title is too general and not entirely on-topic. Metamorphic dome is rather unspecific. Please add an indication of location and perhaps time (Alpine). Given that the applicability of the method is not restricted to metamorphic domes, it may be better to rephrase the title altogether.

The title was changed to " Dating exhumation and fault activity of the Lepontine Dome and Rhone-Simplon Fault regions through hydrothermal monazite-(Ce)"

(4) It would be good to add a sentence or two at the end of the introduction that elaborates on the aims of the study.

This was added

(6) The more generally interested reader may have no idea where we are. I suggest to move Fig. 2 here and to add a reference to this figure to section 2.1. As indicated by Dr. Rahn, Figs. 1 and 2 need to be completed with coordinates, an indication of North, etc.

Changed

(10) The samples were grouped %roughly correlating to tectonic subdivisions+ is a vague statement and leaves the reader unable to judge the criteria that were applied. This might give an unfortunate impression of arbitrary grouping, which renders the paper less persuasive.

This was adjusted

(12) Regarding Figs 3 and 4, it would be good to mention briefly in the text, and not only in the figure caption what characteristic causes the zoning and how that is thought to be related to age information.

More detailed explanations were added to the text

(13) See annotated manuscript for necessary edits to figure 1. The term %Geological-geometric in the caption is unclear. Perhaps best replaced by %Geometry+.

Pages 7 and 8

(15) The images and all lettering in figures 3 and 4 should be enlarged so the reader is better able to assess the placement of the spots. Dr. Rahn mentions justified concerns regarding the placement of spots across boundaries between compositional domains. Some of the spots within one apparent compositional zone have different colours and it is not clear why that is the case (e.g., grains Duro2 and Klem1). The

caption mentions ‰he color of the frame+but it is not entirely clear what that refers to. Is it the box around each weighted mean age result? Please clarify.

Changed

Pages 10-12

(18) In addition to Dr Rahng comments. Please add spot numbers so the ages can be matched to the spots in figures 3 and 4. Enlarge lettering for readability; 6 pts at full size printing is usually considered minimal. I printed the pdf to A4 and most figures are too small in one way or another. The meaning of grey bands in these figures is not clear to me. Are the colours matched with those in figs 3-4?

Spot age numbers were added and colors unified

(19) The content of section 5.1 is more fitting for the introduction than for the discussion.
(22) The decisions behind the groupings are not really explained and therefor the reason has no way to judge whether these decisions are sound, or not. In addition, as indicated in figure 3 some spots within the same apparent chemical domain (based on BSE, other compositional data that may have been used is not available to the reader) are marked with different colours and therefore apparently assigned to different groups for reasons not indicated. The groupings need to be argued more clearly to convince the reader.
(23) ‰o calculate, whenever possible, weighted mean domain ages (Fig. 7).+Should this be figure 8? It is unclear to me what determines whether a weighted domain age can be calculate or, in fact, how this is done. This needs more explanation. It seems that some of this explanation is actually in the paragraph following this reference. It would be better to first explain the procedure and then present the calculated ages.
(24) ‰t appears that if dissolution-precipitation may largely preserve the chemical composition of an affected crystal part, this would mean that areas with different chemical compositions may have reprecipitated simultaneously.+What is the basis for the assumption of preservation? Has this been shown in the literature? Or do the data somehow suggest this? This needs to be explained better. For the second part of the sentence, I do not understand the reasoning either. I am not an expert on monazite dating, but if the authors want the reader to trust the validity of their interpretations, they need to argue their assumptions and decisions more clearly.
(26) There is no figure in the appendix. Has this figure been moved to the inset of Figure 8? Please correct accordingly.

The chapter was moved, reorganized and expanded to provide a better explanation tot the reader.

(29) This is certainly not clear from Fig 2 or 7, and perhaps refers to Fig 8. If so, the statement that the age ranges within grains are generally longer in the Eastern and Southern domain does not appear to be supported. This could also refer to figs 5-7 (I now note that the panels are numbered continuously through figures 5-7, which is rather confusing), but there I do not see a consistency in the graphs to support this statement either. This leaves me at a loss as to the basis of this this statement. This needs to be clarified.

Changed

(31) The shadings in Figure 9 render the ages illegible and this figure needs editing for clarity. It also seems that the age ranges are idealised to an extent: in 9b a 13.6 +/- 0.4 age is included in the 15-14 Ma range and in 9c a 13.4 +/- 0.3 age further West is included in the 13-11 Ma range. The 13-11 Ma area in 9c includes the area coloured in 9b, which contains almost exclusively ages >13 Ma.
The colouring is persuasive but the averaged ages do not appear to match the areas all that closely. From the caption it seems that the shaded areas are based on all ages from each sample, but the weighted mean average ages are based on a selection of those. Such, presumably unintentional juggling with the data makes it almost impossible for the reader to judge the value of the results and interpretations, which is very unfortunate. The authors need to do a better job in presenting their results to convince me that their interpretations are valid and can be used to underpin a tectonic scenario.

The figure was changed and explanations added in caption and text.

(36) The first sentence of the conclusions is a bit awkward. Please rephrase.

Changed

(37) ‰age clusters within individual crystals from a simply exhuming area have a less clear age distribution than samples from fault zone areas, or fast exhuming areas.+ This apparently main conclusion is new here and was not that clearly presented in the discussion. It would be good to add a couple of sentences specifying the argument and its conclusions. The same goes for the next sentence.

Taken out, this was an unfortunate phrasing.

(38) The conclusions presented here paint a much clearer picture than section 5.3. The regional references (to the various faults and domes) are less clear in 5.3. Section 5.3 needs a thorough rewrite, and perhaps splitting in two sections to present the arguments more clearly. The first part could argue the conclusions about slow vs. punctuated events leading to broader and narrower age ranges, respectively, whilst the second part would present the tectonometamorphic development of the study area (leading to the conclusions in the second paragraph of section 6).

The paper was reorganized and should hopefully be more focused now.

**Technical corrections**

Many suggested corrections for spelling and grammar and indicated in the annotated manuscript. In addition, please consider the following numbered comments.

The following techical suggestions were implemented in the manuscript.

(1) Earlier in the Abstract the authors argue that using cleft Mz is superior to other dating techniques because it is not cooling based. It then seems somewhat inconsistent to highlight cooling in line 8. Better to say exhumation only.
(2) The final sentence is very general and it would be better to be more specific as to what kind of information can be derived by dating of cleft-Mz.
(3) Alpine is used a lot here (once also without capital), but the processes considered in the study are not likely to be restricted to Alpine orogenesis or the Alps orogeny. It would be better to phrase this a bit more generally.
(5) This sentence is very vague. Either be more specific, or leave out.
(7) I am not sure what is meant by 'staggered' exhumation.
(8) This reference to Central Lepontine is unclear, because the next sentences refer to more specific areas that are indicated in Fig. 2.
(9) Number figures in order of mention in the text. The current figures 1 and 2 should be swapped. This is consistent with comment (6) above.
(11) Please add the groupings to Table 1.
(14) Please check figure references. This should probably Figs 5 through 7.
Pages 7 and 8
(16) On both pages colour and color is used in the same sentence. Please use either British or American English spelling consistently.
(17) The caption of Table 2 does not describe its content. Please correct. The text that is now in the caption is in fact a note.
(20) Suggestion to rephrase: "... existing grain. Alternatively, dissolution-reprecipitation may cause precipitation of a secondary monazite-(Ce) phase from the fluid film at the surface of the primary phase."
(21) The reference to Grand Homme et al., 2018 is not in the bibliography.
(25) It would be good to add a reference supporting these statements.
(27) "another reason" is confusing here, because in lines 17-18 prolonged tectonic activity is already mentioned as a possible reason for age spread. In addition, the description "... prolonged phases of low-intensity tectonic activity of multiple small deformation events" is very vague. Please revise to address both these issues.
(28) Correct references. Presumably Fig. 8 and Table 2.

(30) Clarify the location of the Rhine-Rhone line in the text and give it the same font size in figure 9 as all other faults

(32) panels a and b are in figure 5, j and k are in 6. All figure references need to be thoroughly checked and corrected.

(33) "clear age patterns within the crystals" is a very vague criterion, which can not be judged by the reader. Please be as specific as possible.

(34) Again, "staggered" is used in a sense that is not entirely clear to me. It would help if the authors clarify to which part of Fig 8 this refers.

(35) The mention of hydrothermal gold mineralisation is very random and appears to have little relationship with the rest of the study. Consider leaving this out.

Pages 19 and further - Bibliography

Missing from the reference list:

Milnes (1974)

Not referenced in the paper:

Frisch (1979)

Frisch et al (2000)

Glotzbach

Keller et al (2006)

Kralik et al

Putnis 2002 and 2009

Schmid et al (1996)

Possible mistakes:

Grand'Homme et al. 2016 or 2018?

Steiger and Jaeger: Title is "Subcommittee on Geochronology: Convention õ .+

Wiederkehr et al. 2008 or 2009?

---

## Author Comment (AC2) · 15 Jul 2019

The comment was uploaded in the form of a supplement:
https://www.solid-earth-discuss.net/se-2019-10/se-2019-10-AC2-supplement.pdf

---

## Author Comment (AC3) · 15 Jul 2019

| 1   | 1      |                                                                                                              |
|-----|--------|--------------------------------------------------------------------------------------------------------------|
| 2   | Con    | straining metamorphic dome exhumation and fault activity                                                     |
| 3   | thr    | cough hydrothermal monazite-(Ce)                                                                             |
| -   | 2 Dat  | ing exhumation and fault activity of the Lenontine Dome and                                                  |
|     | 2 Dat  | cimples Pault activity of the hepotethe bone and                                                             |
|     | 5 Rho  | ine-simpion Fault regions through hydrothermal monazite-(te)                                                 |
| 4   | 4 Chr  | ristian A. Bergemann                                                                                         |
| 5   | 51,    | 2                                                                                                            |
| 6   | 6 , E  | dwin Gnos                                                                                                    |
| 7   | 72     |                                                                                                              |
| 8   | 8, A   | Alfons Berger                                                                                                |
| 9   | 93     |                                                                                                              |
| 10  | 10 F   | milie Janots                                                                                                 |
| 11  | 11 4   |                                                                                                              |
| 11  | 11 4   |                                                                                                              |
| 12  | 12, a  | nd Martin J. Whitehouse                                                                                      |
| 13  | 13 5   |                                                                                                              |
| 14  | 14 1   |                                                                                                              |
| 15  | 15 Uni | versity of Geneva, Geneva, Switzerland                                                                       |
| 16  | 16 2   |                                                                                                              |
| 17  | 17 Nat | ural History Museum of Geneva, Switzerland                                                                   |
| 18  | 18 3   |                                                                                                              |
| 1 9 | 19 Uni | versity of Bern Switzerland                                                                                  |
| 20  | 20 4   | versity of bern, Switzerland                                                                                 |
| 20  | 20 4   |                                                                                                              |
| 21  | 21 IST | Perre University of Grenoble, France                                                                         |
| 22  | 22 5   |                                                                                                              |
| 23  | 23 Swe | dish Museum of Natural History, Stockholm, Sweden                                                            |
| 24  | 24 Cor | respondence:Christian Bergemann (christian.bergemann@unige.ch)                                               |
| 25  | Abs    | stract.Zoned monazite-(Ce) from Alpine fissures/clefts is used to gain new insights into the exhumation      |
|     | his    | story of the                                                                                                 |
| 26  | Cen    | stral Alpine Lepontine metamorphic dome, and timing of deformation along the Rhone-Simplon fault             |
| 20  |        | relat Alphie Beponetine metamorphic dome, and timing of deformation along the knone Simpion fault            |
|     | 20     | me on the domers                                                                                             |
|     | 25 Abs | stract.Zoned hydrothermal monazite-(Ce) from                                                                 |
|     | fis    | sures/clefts is used to gain new insights into the exhumation history                                        |
|     | 26 of  | the Lepontine Dome in the Central Alps, and timing of deformation along the Rhone-Simplon Fault              |
|     | zo     | one on the dome's                                                                                            |
| 27  | 27 wes | stern termination. These hydrothermal monazites-(Ce) directly date deformation and changes in physiochemical |
|     | con    | ditions                                                                                                      |
| 28  | thr    | cough crystallization                                                                                        |
|     | age    | es, in contrast to commonly employed cooling-based methods. The 480 SIMS measurement ages                    |
| 20  | fro    | 20 individual anutals moved again the interval between 10 and t                                              |
| 29  | TIO    | in 20 Individual crystals fecold ages over a time interval between 19 and 5                                  |
|     | Ma,    | with individual grains recording ages over as                                                                |
| 30  | lif    | etime of 2 to 7.5 Ma. The                                                                                    |
|     | age    | e range combined with age distribution and internal crystal structure help to distinguish between            |
|     | are    | eas whose deformational history was dominated by distinct tectonic events or continuous exhumation. The comb |
| 31  | ina    | ution of                                                                                                     |
| 32  | thi    | s age data with geometrical considerations and spatial distribution give a more precise exhumation/cooling   |
|     | hi     | story for the                                                                                                |
| 33  | 2 2 2  | In the east and couth of the study region, the units underwort menagiter (Co)                                |
| 55  | are    | a. In the east and south of the study region, the units underwent monazite-(te)                              |
|     | gro    | wth at 19-12.5 and 16.5-10.5 Ma, followed                                                                    |
| 34  | by     | a central group of monazite-(Ce) ages at 15-10 Ma and the movements and related cleft monazites-(Ce) are     |
|     | you    | ingest at the10                                                                                              |
| 35  | wes    | tern border with 13-7 Ma. A last phase around <mark>8-7</mark> Ma is limited to clefts of                    |
|     | th     | ne Simplon normal fault and related strike                                                                   |
| 36  | sli    | p faults as the Rhone and Rhine-Rhone faults. The large data-set                                             |
|     | spr    | read over significant metamorphic structures shows that                                                      |
| 27  | +bo    | anophing of alofta, fluid flow and monagita (Co) atability is direct linked to the monduments                |
| 51  | che    | - opening of creres, fruid from and monazice-(ce) stability is direct finked to the geodynamic               |
|     | ev     | rolution in space and time.                                                                                  |
| 38  | 1      | Introduction                                                                                                 |
| 39  | Met    | amorphic domes like the Lepontine area of the Central Alps often experienced a complex tectono-metamorphic   |
|     | evo    | lution.15                                                                                                    |
| 40  | In     | this case an interplay between exhumation and deformation during doming and activity of                      |
|     |        |                                                                                                              |

|    |    | large fault systems that dominate                                                                                       |
|----|----|-------------------------------------------------------------------------------------------------------------------------|
| 41 |    | the western parts of the area. Although much of the (thermo)chronological history of the area is well known,            |
|    |    | hydrothermal                                                                                                            |
| 42 |    | monazite-(Ce) ages complement existing cooling ages of                                                                  |
|    |    | zircon fission track, Rb-Sr in biotite and apatite fission track/apatite                                                |
| 43 |    | U-Th/He by providing crystallization and dissolution-precipitation ages that date low-T tectonic evolution.             |
|    | 28 | through crystallization                                                                                                 |
|    |    | ages. 480 SIMS spot analyses from 20 individual crystals, including co-type material of the monazite-                   |
|    | 29 | (Nd) type locality, record ages over a time interval $of \sim 19$ and 2.7 Ma, with individual grains recording          |
|    |    | age ranges of 2 to5                                                                                                     |
|    | 30 | 7.5 Myr. The combination of                                                                                             |
|    | 31 | exhumation bistory for the area. In the north-coast and south-west of the Longatine Dome units                          |
|    | JT | underwent monazite-(Ce)                                                                                                 |
|    | 32 | growth at 19-12.5 and 16.5-10.5                                                                                         |
|    |    | Ma respectively, followed by crystallization of monazite-(Ce) in the central part at 15-10 Ma.                          |
|    | 33 | Fissure monazites-(Ce) are younger at the western limit of the dome with 13-7 Ma. A last age group around 8-5           |
|    |    | Ma is limited to                                                                                                        |
|    | 34 | fissures/clefts associated with the Simplon normal fault and related strike-slip faults such as the Rhone               |
|    |    | Fault. The large data-set10                                                                                             |
|    | 35 | spread over significant metamorphic structures shows that the fissure mineral crystal-rock interaction,                 |
|    |    | fluid flow and monazite-                                                                                                |
|    | 36 | (Ce) stability are directly linked to the Lepontine Dome's evolution in space and                                       |
|    |    | time. A comparison between hydrothermal                                                                                 |
|    | 37 | monazite-(Ce) thermo-chronometric data suggests that hydrothermal monazite-(Ce) dating could allow to identify          |
|    | 38 | aleas of                                                                                                                |
|    | 39 | 1 Introduction15                                                                                                        |
|    | 40 | Metamorphic domes often experience a complex tectono-metamorphic                                                        |
|    |    | evolution (e.g.Schmidet al., 2004; Stecket al., 2013).                                                                  |
|    | 41 | For the Lepontine Dome of the European Alps, this evolution is                                                          |
|    |    | an interplay between exhumation and deformation during                                                                  |
|    | 42 | doming and motion along large fault systems that dominate the western regions of the dome.                              |
|    |    | Although much of the retrograde                                                                                         |
|    | 43 | orogenic evolution of the area is well known, hydrothermal                                                              |
|    | лл | chronometers by providing crystallization and dissolution-precipitation ages that directly                              |
|    |    | date low-T tectonic activity.20                                                                                         |
| 44 | 45 | Monazite, (LREE, Th, U) PO                                                                                              |
| 45 | 46 | 4                                                                                                                       |
| 46 |    | , is considered an excellent mineral for dating of geologic processes (e.g., Parrish, 1990) that is20                   |
| 47 |    | highly resistant to radiation damage <mark>(e.g., Meldrum et</mark> al., 1998, 1999, 2000) and shows negligible Pb loss |
|    |    | through diffusion                                                                                                       |
|    |    | (Cherniak et al., 2004; Cherniak and Pyle, 2008). Nonetheless, monazite remains geologically reactive after cr          |
| 48 |    | ystallization. It                                                                                                       |
|    | 47 | , is considered an excellent mineral for the dating of geologic processes (e.g.Parrish, 1990).                          |
|    | 48 | It is nightly resistant to radiation damage (e.g.Meldrumet al., 1998, 1999, 2000) and shows negligible PD loss          |
| 49 | 49 | 1                                                                                                                       |
| 50 | 10 | Solid Earth Discuss., https://doi.org/10.5194/se-2019-10                                                                |
| 51 |    | Manuscript under review for journal Solid Earth                                                                         |
| 52 |    | Discussion started: 5 February 2019                                                                                     |
| 53 |    | c                                                                                                                       |
| 54 |    | ©Author(s) 2019. CC BY 4.0 License.                                                                                     |
| 55 | 50 | err everyteres discolution reconstallization, thereby recording new area through mediation of budyous fluids            |
| 56 |    | (e a . Sevioux-                                                                                                         |
| 57 |    | Guillaume et al., 2012; Janots et al., 2012; Grand                                                                      |
|    |    | (Cherniaket al., 2004; Cherniak and Pyle, 2008). Nonetheless, monazite remains reactive after crystallization,          |
|    | 51 | as it can expe-                                                                                                         |
|    |    | rience dissolution-recrystallization facilitated through hydrous fluids (e.g.Seydoux-Guillaumeet al., 2012; Ja          |
|    | 52 | notset al., 2012;                                                                                                       |
|    | 53 | Grand                                                                                                                   |
| 58 | 54 | ,                                                                                                                       |
| JJ |    | Homme et al., 2010).                                                                                                    |

| 60                         | Alpine fissures and clefts occasionally containing monazite-(Ce) are voids partially filled by crystals                                                                                                                                                                                                                                                                                                                                                                                                                                                                                                                                                                                                                                                                                                                                                                                                                                                                                                                                                                                                                                                                                                                                                                                                                                                                                                                                                                                                                                                                                                                                                                                                                                                                                                                         |              |
|----------------------------|---------------------------------------------------------------------------------------------------------------------------------------------------------------------------------------------------------------------------------------------------------------------------------------------------------------------------------------------------------------------------------------------------------------------------------------------------------------------------------------------------------------------------------------------------------------------------------------------------------------------------------------------------------------------------------------------------------------------------------------------------------------------------------------------------------------------------------------------------------------------------------------------------------------------------------------------------------------------------------------------------------------------------------------------------------------------------------------------------------------------------------------------------------------------------------------------------------------------------------------------------------------------------------------------------------------------------------------------------------------------------------------------------------------------------------------------------------------------------------------------------------------------------------------------------------------------------------------------------------------------------------------------------------------------------------------------------------------------------------------------------------------------------------------------------------------------------------|--------------|
|                            | that crystallized on                                                                                                                                                                                                                                                                                                                                                                                                                                                                                                                                                                                                                                                                                                                                                                                                                                                                                                                                                                                                                                                                                                                                                                                                                                                                                                                                                                                                                                                                                                                                                                                                                                                                                                                                                                                                            |              |
| 61                         | the cleft walls from hydrous fluids during late stage Alpine metamorphism (Mullis et al., 1994; Mullis,                                                                                                                                                                                                                                                                                                                                                                                                                                                                                                                                                                                                                                                                                                                                                                                                                                                                                                                                                                                                                                                                                                                                                                                                                                                                                                                                                                                                                                                                                                                                                                                                                                                                                                                         |              |
|                            | 1996). Dating such                                                                                                                                                                                                                                                                                                                                                                                                                                                                                                                                                                                                                                                                                                                                                                                                                                                                                                                                                                                                                                                                                                                                                                                                                                                                                                                                                                                                                                                                                                                                                                                                                                                                                                                                                                                                              |              |
| c 0                 | mineralization is often difficult due to later overprinting along with multiple stages of fluid activity (Pu:                                                                                                                                                                                                                                                                                                                                                                                                                                                                                                                                                                                                                                                                                                                                                                                                                                                                                                                                                                                                                                                                                                                                                                                                                                                                                                                                                                                                                                                                                                                                                                                                                                                                                                                   | rd           |
| 62                         | y
and obsider (1072) F                                                                                                                                                                                                                                                                                                                                                                                                                                                                                                                                                                                                                                                                                                                                                                                                                                                                                                                                                                                                                                                                                                                                                                                                                                                                                                                                                                                                                                                                                                                                                                                                                                                                                                                                                                                                       |              |
| c 2                 | and Stalder, 1973).5                                                                                                                                                                                                                                                                                                                                                                                                                                                                                                                                                                                                                                                                                                                                                                                                                                                                                                                                                                                                                                                                                                                                                                                                                                                                                                                                                                                                                                                                                                                                                                                                                                                                                                                                                                                                            |              |
| 03                         | Alpine fissures in some metasediments and metagranitoids have long been known to contain                                                                                                                                                                                                                                                                                                                                                                                                                                                                                                                                                                                                                                                                                                                                                                                                                                                                                                                                                                                                                                                                                                                                                                                                                                                                                                                                                                                                                                                                                                                                                                                                                                                                                                                                        |              |
| C A                        | erustale (Niggli et al. 1040) but it is only recently that some of these                                                                                                                                                                                                                                                                                                                                                                                                                                                                                                                                                                                                                                                                                                                                                                                                                                                                                                                                                                                                                                                                                                                                                                                                                                                                                                                                                                                                                                                                                                                                                                                                                                                                                                                                                        |              |
| 04                         | crystars (Niggir et al., 1940), but it is only recently that some of these                                                                                                                                                                                                                                                                                                                                                                                                                                                                                                                                                                                                                                                                                                                                                                                                                                                                                                                                                                                                                                                                                                                                                                                                                                                                                                                                                                                                                                                                                                                                                                                                                                                                                                                                                      |              |
| 65                         | 2012) Although other minerals like mices and adularia are common in alnine figsures, they are often                                                                                                                                                                                                                                                                                                                                                                                                                                                                                                                                                                                                                                                                                                                                                                                                                                                                                                                                                                                                                                                                                                                                                                                                                                                                                                                                                                                                                                                                                                                                                                                                                                                                                                                             |              |
| 00                         | affected by overpres-                                                                                                                                                                                                                                                                                                                                                                                                                                                                                                                                                                                                                                                                                                                                                                                                                                                                                                                                                                                                                                                                                                                                                                                                                                                                                                                                                                                                                                                                                                                                                                                                                                                                                                                                                                                                           |              |

[revised manuscript text omitted]

55 Hommeet al., 2016).
56 Fissures and clefts occasionally contain hydrothermal monazite-(Ce). They represent
voids partially filled by crystals that                                                                                                                                                                                                                                                                                                                                                                                                                                                                                                                                                                                                                                                                                                                                                                                                                                                                                                                                                                  | A
M
ne |
| 86
87
88
89
90 | <pre>Allo central23
Alps with the Lepontine metamorphic dome have consequently had a complex tectonic and metamorphic history.
Early high-pressure metamorphism in the Western Alpine Sesia-Lanzo Zone during subduction below the Southern
lps is
dated at 75-65 Ma (e.g.Ruffet et al., 1997; Duchêne et al., 1997; Rubatto et al., 1998). This was followed by
underthrusting and
nappe stacking from ca. 42 Ma on during continental collision linked with a transition from high-P to high-T
etamorphism
(e.g. Köppel and Grünenfelder, 1975; Markleyet al., 1998; Herwartz et al., 2011; Boston et al., 2017). Peak r
tamorphic30
conditions in the Lepontine area in excess of 650
55 Hommeet al., 2016).
56 Fissures and clefts occasionally contain hydrothermal monazite-(Ce). They represent
voids partially filled by crystals that
57 precipitated on the fissure walls from hydrous fluids during late stage metamorphism (Mulliset
al. 1994, Mullis, 1996) 5</pre>                                                                                                                                                                                                                                                                                                                                                                                                                                                                                                                                                                                                                                                                                                                                                                                    | A
m
ne |
| 86
87
88
89
90 | <pre>Allo Centrarys
Alps with the Lepontine metamorphic dome have consequently had a complex tectonic and metamorphic history.
Early high-pressure metamorphism in the Western Alpine Sesia-Lanzo Zone during subduction below the Southern
lps is
dated at 75-65 Ma (e.g.Ruffet et al., 1997; Duchêne et al., 1997; Rubatto et al., 1998). This was followed by
underthrusting and
nappe stacking from ca. 42 Ma on during continental collision linked with a transition from high-P to high-T
etamorphism
(e.g. Köppel and Grünenfelder, 1975; Markleyet al., 1998; Herwartz et al., 2011; Boston et al., 2017). Peak r
tamorphic30
conditions in the Lepontine area in excess of 650
55 Hommeet al., 2016).
56 Fissures and clefts occasionally contain hydrothermal monazite-(Ce). They represent
voids partially filled by crystals that
57 precipitated on the fissure walls from hydrous fluids during late stage metamorphism (Mulliset
al., 1994; Mullis, 1996).5
58 hating such</pre>                                                                                                                                                                                                                                                                                                                                                                                                                                                                                                                                                                                                                                                                                                                                                                | A
M
ne |
| 86
87
88
89
90 | <pre>Allos with the Lepontine metamorphic dome have consequently had a complex tectonic and metamorphic history.
Early high-pressure metamorphism in the Western Alpine Sesia-Lanzo Zone during subduction below the Southern
lps is
dated at 75-65 Ma (e.g.Ruffet et al., 1997; Duchêne et al., 1997; Rubatto et al., 1998). This was followed by
underthrusting and
nappe stacking from ca. 42 Ma on during continental collision linked with a transition from high-P to high-T
etamorphism
(e.g. Köppel and Grünenfelder, 1975; Markleyet al., 1998; Herwartz et al., 2011; Boston et al., 2017). Peak mathematication
(e.g. Köppel and Grünenfelder, 1975; Markleyet al., 1998; Herwartz et al., 2011; Boston et al., 2017). Peak mathematication
(e.g. Köppel and Grünenfelder, 1975; Markleyet al., 1998; Herwartz et al., 2011; Boston et al., 2017). Peak mathematication
(e.g. Köppel and Grünenfelder, 1975; Markleyet al., 1998; Herwartz et al., 2011; Boston et al., 2017). Peak mathematication
(e.g. Köppel and Grünenfelder, 1975; Markleyet al., 1998; Herwartz et al., 2011; Boston et al., 2017). Peak mathematication
(e.g. Köppel and Grünenfelder, 1975; Markleyet al., 1998; Herwartz et al., 2011; Boston et al., 2017). Peak mathematication
(conditions in the Lepontine area in excess of 650
55 Hommeet al., 2016).
56 Fissures and clefts occasionally contain hydrothermal monazite-(Ce). They represent
voids partially filled by crystals that
57 precipitated on the fissure walls from hydrous fluids during late stage metamorphism (Mulliset
al., 1994; Mullis, 1996).5
58 Dating such
mineralization is often difficult due to later overprinting along with multiple stages of fluid activity (Pur</pre> | A
m
ne |
| 86
87
88
89
90 | Alps with the Lepontine metamorphic dome have consequently had a complex tectonic and metamorphic history.
Early high-pressure metamorphism in the Western Alpine Sesia-Lanzo Zone during subduction below the Southern
lps is
dated at 75-65 Ma (e.g.Ruffet et al., 1997; Duchène et al., 1997; Rubatto et al., 1998). This was followed by
underthrusting and
nappe stacking from ca. 42 Ma on during continental collision linked with a transition from high-P to high-T
etamorphism
(e.g. Köppel and Grünenfelder, 1975; Markleyet al., 1998; Herwartz et al., 2011; Boston et al., 2017). Peak r
tamorphic30
conditions in the Lepontine area in excess of 650
55 Hommeet al., 2016).
56 Fissures and clefts occasionally contain hydrothermal monazite-(Ce). They represent
voids partially filled by crystals that
57 precipitated on the fissure walls from hydrous fluids during late stage metamorphism (Mulliset
al., 1994; Mullis, 1996).5
58 Dating such
mineralization is often difficult due to later overprinting along with multiple stages of fluid activity (Pu:
V                                                                                                                                                                                                                                                                                                                                                                                                                                                                                                                                                                                                                                                                                       | A
m
ne |
| 86
87
88
89
90 | Alps with the Lepontine metamorphic dome have consequently had a complex tectonic and metamorphic history.
Early high-pressure metamorphism in the Western Alpine Sesia-Lanzo Zone during subduction below the Southern
lps is
dated at 75-65 Ma (e.g.Ruffet et al., 1997; Duchêne et al., 1997; Rubatto et al., 1998). This was followed by
underthrusting and
nappe stacking from ca. 42 Ma on during continental collision linked with a transition from high-P to high-T
etamorphism
(e.g. Köppel and Grünenfelder, 1975; Markleyet al., 1998; Herwartz et al., 2011; Boston et al., 2017). Peak r
tamorphic30
conditions in the Lepontine area in excess of 650
55 Hommeet al., 2016).
56 Fissures and clefts occasionally contain hydrothermal monazite-(Ce). They represent
voids partially filled by crystals that
57 precipitated on the fissure walls from hydrous fluids during late stage metamorphism (Mulliset
al., 1994; Mullis, 1996).5
58 Dating such
mineralization is often difficult due to later overprinting along with multiple stages of fluid activity (Pu:
y
and                                                                                                                                                                                                                                                                                                                                                                                                                                                                                                                                                                                                                                                                                | A
m
ne |

|     | 59         | Stalder, 1973). Fissures and clefts in some metasediments and metagranitoids have long been known to contain     |
|-----|------------|------------------------------------------------------------------------------------------------------------------|
|     | c 0 | well-developed                                                                                                   |
|     | 60         | monazite-(Ce) crystals (Niggliet al., 1940), but it is only recently that some of these                          |
|     | C 1        | Were dated (e.g.Gasquetet al., 2010;                                                                             |
|     | ΟŢ         | Danotset al.,                                                                                                    |
|     |            | 2012). Although other minerals like micas and adularia are common in alpine fissures, they are often affected    |
|     | 62         | by overpressure/excess                                                                                           |
|     |     | argon, (e.g., Furdy and Stalder, 1973), and it is not always clear if these ages represent crystallizationic     |
|     | 63         | or cooling (e.g., Rauchenstein-Martinek, 2014). The                                                              |
|     | C A        | rissures and clerts in the Lepontine region formed after the metamorphic                                         |
|     | 04         | peak, in relation to extensional tectonic activity. Accordingly, fissures and clefts are oriented roughly        |
|     | 65         | lineation                                                                                                        |
|     | 00         | and foliation of the bost rock. The fluid that intruded during figure formation (300-500                         |
| 91  | 66         | and forfactor of the nost fock. The finite that included during fissure formation (500-500                       |
| 92  | 00         | C in some regions were reached diachronously from south to north in time                                         |
|     |            | around 30-19 Ma and accompanied by limited magmatic activity from 33 Ma down to ca. 22 Ma (yon Blanckenburg et   |
| 93  |            | al., 1991;                                                                                                       |
|     |            | Romer et al., 1996; Schärer et al., 1996; Oberli et al., 2004; Rubatto et al. 2009; Janots et al., 2009). Prog   |
| 94  |            | rade metamorphism                                                                                                |
|     | 67         | C; Mulliset al., 1994;                                                                                           |
|     |            | Mullis, 1996) interacted with the wall rock. This triggered dissolution and precipitation of minerals in both    |
|     | 68         | host rock and                                                                                                    |
|     |            | fissure, leading to the formation of a porous alteration halo in the surrounding wall rock. Complex growth dom   |
|     | 69         | ains are common15                                                                                                |
|     |            | in hydrothermal monazite-(Ce) from such fissures showing both, dissolution and secondary growth (e.g.Janotset    |
|     | 70         | al., 2012;                                                                                                       |
|     |            | Bergemannet al., 2017, 2018), as well as dissolution-reprecipitation reactions resulting in patchy grains (e.    |
|     | 71         | g.Gnoset al.,                                                                                                    |
|     |            | 2015). In contrast to metamorphic rocks, where monazite-(Ce) rarely exceeds 100µm, fissure monazite-(Ce) is co   |
|     | 72         | mmonly                                                                                                           |
|     |            | mm-sized, with large individual growth domains. This permits dating individual domains precisely by using seco   |
|     | 73         | ndary ion                                                                                                        |
|     |            | mass spectrometry (SIMS), resolve growth duration and identify phases and single events of tectonic activity     |
|     | 74         | (e.g.Janotset al.,20                                                                                             |
|     | 75         | 2012; Bergeret al., 2013; Bergemannet al., 2017, 2018, 2019).                                                    |
|     |            | The aim of this study is to illustrate that hydrothermal monazite-(Ce) dating provides information about the t   |
|     | 76         | ectonic evolution                                                                                                |
|     | 77         | of the Lepontine Dome.                                                                                           |
|     | 78         | 2 Geological setting                                                                                             |
|     | 79         | 2.1 Evolution of the study area25                                                                                |
|     |            | The formation of the nappe stack of the European Alps caused by the collision of the European and Adriatic pla   |
|     | 80         | tes was followed                                                                                                 |
|     | 0.1        | by the development of several metamorphic areas (Tauern, Rechnitz in the Eastern Alps, and Lepontine in the Ce   |
|     | 8 T        | ntral Alps;e.g.                                                                                                  |
|     | 0.0        | Schmidet al., 2004). Their formation was related to crustal shortening associated with coeval orogen-parallel    |
|     | 82         | extension (e.g.                                                                                                  |
|     | 0.2        | Mancktelow, 1992; Ratschbacheret al., 1989; Ratschbacheret al., 1991). The Western and Central Alps with the L   |
|     | 00         | epontine                                                                                                         |
|     | 04         | For have consequencily had a complex tectoric and metamorphic history.50                                         |
|     | 85         | larry high pressure metamorphism in the western kipine sesta banzo zone auting subduction below the Southern k   |
|     | 00         | is dated at 75-65 Ma (e o Ruffetet al . 1997: Rubattoet al . 1998: Regiset al . 2014). This was followed by un   |
|     | 86         | derthrusting                                                                                                     |
|     | 20         | and nappe stacking fromca.42 Ma on during continental collision linked with a transition from high-P/low-T to    |
|     | 87         | barrow type                                                                                                      |
| 95  | 88         | 2                                                                                                                |
| 96  |            | Solid Earth Discuss., https://doi.org/10.5194/se-2019-10                                                         |
| 97  |            | Manuscript under review for journal Solid Earth                                                                  |
| 98  |            | Discussion started: 5 February 2019                                                                              |
| 99  |            | c and a second |
| 100 |            | ©Author(s) 2019. CC BY 4.0 License.                                                                              |
| 101 | 89         |                                                                                                                  |
| 102 |            | was followed by staggered exhumation in the Ticino and Toce culminations of the Lepontine dome.                  |
|     |            | Accelerated Cooling Delow                                                                                        |

| 103                                                                                                                                                                                                                        |    | 500                                                                                                                                                                                                                                                                                                                                                                                                                                                                                                                                                                                                                                                                                                                                                                                                                                                                                                                                                                                                                                                                                                                                                                                                                                                                                                                                                                                                                                                                                                                                                                                                                                                                                                                                                                                                                                                                                                                                                                                                                |
|----------------------------------------------------------------------------------------------------------------------------------------------------------------------------------------------------------------------------|----|--------------------------------------------------------------------------------------------------------------------------------------------------------------------------------------------------------------------------------------------------------------------------------------------------------------------------------------------------------------------------------------------------------------------------------------------------------------------------------------------------------------------------------------------------------------------------------------------------------------------------------------------------------------------------------------------------------------------------------------------------------------------------------------------------------------------------------------------------------------------------------------------------------------------------------------------------------------------------------------------------------------------------------------------------------------------------------------------------------------------------------------------------------------------------------------------------------------------------------------------------------------------------------------------------------------------------------------------------------------------------------------------------------------------------------------------------------------------------------------------------------------------------------------------------------------------------------------------------------------------------------------------------------------------------------------------------------------------------------------------------------------------------------------------------------------------------------------------------------------------------------------------------------------------------------------------------------------------------------------------------------------------|
|                                                                                                                                                                                                                            |    | Figure 1.Map of the Lepontine Dome, modified from Stecket al.(2013) and Schmidet al.(2004). Colored areas mark                                                                                                                                                                                                                                                                                                                                                                                                                                                                                                                                                                                                                                                                                                                                                                                                                                                                                                                                                                                                                                                                                                                                                                                                                                                                                                                                                                                                                                                                                                                                                                                                                                                                                                                                                                                                                                                                                                     |
|                                                                                                                                                                                                                            | 90 | the areal division in                                                                                                                                                                                                                                                                                                                                                                                                                                                                                                                                                                                                                                                                                                                                                                                                                                                                                                                                                                                                                                                                                                                                                                                                                                                                                                                                                                                                                                                                                                                                                                                                                                                                                                                                                                                                                                                                                                                                                                                              |
|                                                                                                                                                                                                                            | 91 | the context of this study.                                                                                                                                                                                                                                                                                                                                                                                                                                                                                                                                                                                                                                                                                                                                                                                                                                                                                                                                                                                                                                                                                                                                                                                                                                                                                                                                                                                                                                                                                                                                                                                                                                                                                                                                                                                                                                                                                                                                                                                         |
|                                                                                                                                                                                                                            |    | metamorphism (medium P/T;e.g.Köppel and Grünenfelder, 1975; Markleyet al., 1998; Herwartzet al., 2011; Bostone                                                                                                                                                                                                                                                                                                                                                                                                                                                                                                                                                                                                                                                                                                                                                                                                                                                                                                                                                                                                                                                                                                                                                                                                                                                                                                                                                                                                                                                                                                                                                                                                                                                                                                                                                                                                                                                                                                     |
|                                                                                                                                                                                                                            | 92 | t al.,                                                                                                                                                                                                                                                                                                                                                                                                                                                                                                                                                                                                                                                                                                                                                                                                                                                                                                                                                                                                                                                                                                                                                                                                                                                                                                                                                                                                                                                                                                                                                                                                                                                                                                                                                                                                                                                                                                                                                                                                             |
|                                                                                                                                                                                                                            | 93 | 2017). Peak metamorphic conditions in the Lepontine area (in excess of 650                                                                                                                                                                                                                                                                                                                                                                                                                                                                                                                                                                                                                                                                                                                                                                                                                                                                                                                                                                                                                                                                                                                                                                                                                                                                                                                                                                                                                                                                                                                                                                                                                                                                                                                                                                                                                                                                                                                                         |
|                                                                                                                                                                                                                            | 94 |                                                                                                                                                                                                                                                                                                                                                                                                                                                                                                                                                                                                                                                                                                                                                                                                                                                                                                                                                                                                                                                                                                                                                                                                                                                                                                                                                                                                                                                                                                                                                                                                                                                                                                                                                                                                                                                                                                                                                                                                                    |
|                                                                                                                                                                                                                            | 95 | C in some regions) were reached diachronously                                                                                                                                                                                                                                                                                                                                                                                                                                                                                                                                                                                                                                                                                                                                                                                                                                                                                                                                                                                                                                                                                                                                                                                                                                                                                                                                                                                                                                                                                                                                                                                                                                                                                                                                                                                                                                                                                                                                                                      |
|                                                                                                                                                                                                                            | 96 | from south to north around 30-19 Ma (e.g.Schäreret al., 1996). Barrovian metamorphism was followed by                                                                                                                                                                                                                                                                                                                                                                                                                                                                                                                                                                                                                                                                                                                                                                                                                                                                                                                                                                                                                                                                                                                                                                                                                                                                                                                                                                                                                                                                                                                                                                                                                                                                                                                                                                                                                                                                                                              |
|                                                                                                                                                                                                                            |    | exhumation starting                                                                                                                                                                                                                                                                                                                                                                                                                                                                                                                                                                                                                                                                                                                                                                                                                                                                                                                                                                                                                                                                                                                                                                                                                                                                                                                                                                                                                                                                                                                                                                                                                                                                                                                                                                                                                                                                                                                                                                                                |
|                                                                                                                                                                                                                            | 97 | in the east and                                                                                                                                                                                                                                                                                                                                                                                                                                                                                                                                                                                                                                                                                                                                                                                                                                                                                                                                                                                                                                                                                                                                                                                                                                                                                                                                                                                                                                                                                                                                                                                                                                                                                                                                                                                                                                                                                                                                                                                                    |
|                                                                                                                                                                                                                            |    | migrating to the west of the Lepontine Dome, with vertical displacement along the Insubric Line starting as                                                                                                                                                                                                                                                                                                                                                                                                                                                                                                                                                                                                                                                                                                                                                                                                                                                                                                                                                                                                                                                                                                                                                                                                                                                                                                                                                                                                                                                                                                                                                                                                                                                                                                                                                                                                                                                                                                        |
| 1.0.4                                                                                                                                                                                                                      | 98 | early as 30 Ma (e.g.Hurford, 1986; Steck and Hunziker, 1994). Accelerated cooling below 500                                                                                                                                                                                                                                                                                                                                                                                                                                                                                                                                                                                                                                                                                                                                                                                                                                                                                                                                                                                                                                                                                                                                                                                                                                                                                                                                                                                                                                                                                                                                                                                                                                                                                                                                                                                                                                                                                                                        |
| 104                                                                                                                                                                                                                        | 99 | •                                                                                                                                                                                                                                                                                                                                                                                                                                                                                                                                                                                                                                                                                                                                                                                                                                                                                                                                                                                                                                                                                                                                                                                                                                                                                                                                                                                                                                                                                                                                                                                                                                                                                                                                                                                                                                                                                                                                                                                                                  |
| 102                                                                                                                                                                                                                        |    | c occurred at 20 Ma first in the central Lepontine (Huriord, 1986). This was followed in the                                                                                                                                                                                                                                                                                                                                                                                                                                                                                                                                                                                                                                                                                                                                                                                                                                                                                                                                                                                                                                                                                                                                                                                                                                                                                                                                                                                                                                                                                                                                                                                                                                                                                                                                                                                                                                                                                                                       |
| 106                                                                                                                                                                                                                        |    | cooling of the Ticino dome between 22 and 17 Ma (Steck and Hunziker, 1994; Pubatto et                                                                                                                                                                                                                                                                                                                                                                                                                                                                                                                                                                                                                                                                                                                                                                                                                                                                                                                                                                                                                                                                                                                                                                                                                                                                                                                                                                                                                                                                                                                                                                                                                                                                                                                                                                                                                                                                                                                              |
| TOO                                                                                                                                                                                                                        |    | al 2009) after which exhumation                                                                                                                                                                                                                                                                                                                                                                                                                                                                                                                                                                                                                                                                                                                                                                                                                                                                                                                                                                                                                                                                                                                                                                                                                                                                                                                                                                                                                                                                                                                                                                                                                                                                                                                                                                                                                                                                                                                                                                                    |
| 107                                                                                                                                                                                                                        |    | slowed down. To the west, the Toce dome experienced phases of accelerated cooling somewhat later in the time                                                                                                                                                                                                                                                                                                                                                                                                                                                                                                                                                                                                                                                                                                                                                                                                                                                                                                                                                                                                                                                                                                                                                                                                                                                                                                                                                                                                                                                                                                                                                                                                                                                                                                                                                                                                                                                                                                       |
|                                                                                                                                                                                                                            |    | of 18-15 Ma                                                                                                                                                                                                                                                                                                                                                                                                                                                                                                                                                                                                                                                                                                                                                                                                                                                                                                                                                                                                                                                                                                                                                                                                                                                                                                                                                                                                                                                                                                                                                                                                                                                                                                                                                                                                                                                                                                                                                                                                        |
|                                                                                                                                                                                                                            |    | and 12-10 Ma (Campani et al., 2014). The later cooling phase was related to detachment along the Rhone-Simplon                                                                                                                                                                                                                                                                                                                                                                                                                                                                                                                                                                                                                                                                                                                                                                                                                                                                                                                                                                                                                                                                                                                                                                                                                                                                                                                                                                                                                                                                                                                                                                                                                                                                                                                                                                                                                                                                                                     |
| 108                                                                                                                                                                                                                        |    | Fault (Steck5                                                                                                                                                                                                                                                                                                                                                                                                                                                                                                                                                                                                                                                                                                                                                                                                                                                                                                                                                                                                                                                                                                                                                                                                                                                                                                                                                                                                                                                                                                                                                                                                                                                                                                                                                                                                                                                                                                                                                                                                      |
| 109                                                                                                                                                                                                                        |    | and Hunziker, 1994; Campani et al., 2014).                                                                                                                                                                                                                                                                                                                                                                                                                                                                                                                                                                                                                                                                                                                                                                                                                                                                                                                                                                                                                                                                                                                                                                                                                                                                                                                                                                                                                                                                                                                                                                                                                                                                                                                                                                                                                                                                                                                                                                         |
|                                                                                                                                                                                                                            |    | While most of the Lepontine area is marked by doming and associated deformation events, the western and southw                                                                                                                                                                                                                                                                                                                                                                                                                                                                                                                                                                                                                                                                                                                                                                                                                                                                                                                                                                                                                                                                                                                                                                                                                                                                                                                                                                                                                                                                                                                                                                                                                                                                                                                                                                                                                                                                                                     |
| 110                                                                                                                                                                                                                        |    | estern                                                                                                                                                                                                                                                                                                                                                                                                                                                                                                                                                                                                                                                                                                                                                                                                                                                                                                                                                                                                                                                                                                                                                                                                                                                                                                                                                                                                                                                                                                                                                                                                                                                                                                                                                                                                                                                                                                                                                                                                             |
| 111                                                                                                                                                                                                                        |    | limits of the study area are dominated by the Rhone-Simplon Fault system, its <mark>extensions</mark>                                                                                                                                                                                                                                                                                                                                                                                                                                                                                                                                                                                                                                                                                                                                                                                                                                                                                                                                                                                                                                                                                                                                                                                                                                                                                                                                                                                                                                                                                                                                                                                                                                                                                                                                                                                                                                                                                                              |
|

---

## Referee Report (RR1)

**Dating exhumation and fault activity of the Lepontine Dome and Rhone-Simplon Fault regions through hydrothermal monazite-(Ce)**

Christian A. Bergemann, Edwin Gnos, Alfons Berger, Emilie Janots, Martin J. Whitehouse

by Meinert Rahn

This review is an update of a previous version of the manuscript. The study of Bergemann and coworkers presents 480 single spot ages and 33 weighted mean ages from 19 locations and their cleft monazites within the northern Lepontine Dome (and areas adjacent to the dome). Here, I first pass through the major issues of criticism from the previous review to check what the authors have done with it.

- Title of the manuscript. No doubt, the title has been changed, but the content has also been changed. I feel that the title still is very ambiguous with respect to the data which cannot (like many methods with a well-defined closure temperature) be linked to a specific moment of cooling in the exhumation history. I below show that the temperature range the authors argue with can at least be questioned. I would give this point a "partly fulfilled".
- 2. Mess with figure numbers: No doubt, this has been sorted out, and this issue can be closed.
- 3. Short chapter on "Results": The chapter on the result has been extended, but many issues still remain vague and are not clear. With respect to shortness, the issue is solved, with respect to clarity, the issue remains open.
- 4. Methodical descriptions: The authors have intensified their description on how, the data have been produced, but I feel that aspects have been blown up that do not play a role in your final data interpretation. Together with figure 5, the authors develop a distinction into several sample groups, based on the internal structures, which I am unable to follow and which in the end have no relevance for the age data:
  - a. "Monazite stability field": The authors claim at several places a range of 200-350 °C to be the temperature range of hydrothermal monazite formation (which is clearly different from its formation as rock forming mineral with a U/Pb closure temperature above 400°C). However, in figure 6, it becomes evident that the range given above does not fit, if the monazite ages are compared to the ages of thermochronometers with well-defined closure temperatures. Find a figure on that issue below, where I have compared three localities (TAMB 1, VALS and VANI 6)

The orange age range is given by the monazite, while the purple envelope covers the existing thermochronology data I know from these localities. For the locality VANI 6, the resulting formation range (indicated by the dashed red vertical arrows) corresponds well to the quoted temperature range of 200-350°C, however for the two others, the range is at distinctly lower temperatures. In the end, the range does not really matter, as the range, if I understand the authors correctly, is not comparable to a closure temperature. But it illustrates that fissure monazites can form over wide temperature range. Since monazite formation in fissures is controlled by the opening of fissures and the evolution of the fluid, the temperature within the fissure should not be taken as a measure for the temperature of the surrounding rock. One part of disequilibrium with fissures is that there is a thermal gradient from fissure to rock, which leads to fast cooling and precipitation of minerals in the fissure. This brings me

to another unsolved issue:

"Disequilibrium": I do not find the discussion about chemical equilibrium and disequilibrium very fruitful. Can we agree on this: Mineral precipitation within fissures is the result of disequilibrium and the system's approach to reach equilibrium? It does not matter whether this mean chemical or physical disequilibrium as they commonly develop simultaneously. Accordingly, monazites in fissures are not a sign of equilibrium, but a disturbed system that reacted by precipitating this mineral. The current discussion on equilibrium and disequilibrium is at odds with such a description.

- b. BSE images: Yes, figure 5 has now a key function, and I like it very much to gain an overview. However, it is also this figure that causes most of the methodical trouble: I do not understand your grouping of the samples (not the regional one, but the one on internal structures). I can only follow some of your descriptions, as they are not clear and they are not transparent to the reader. You are using several terms that are not clear (or at least not clear to me): zonation, sector zonation, sector-like zonation, oscillatory zonation, oscillatory-complex zonation, ring zonation, complex zonation, alteration, alteration features, diffuse pattern. Even with the help of figures 3 and 5, for most of these features I cannot follow your description. It does not help to speak about "signs of alteration", if the reader does not know, what "signs" you are talking about. I understand that the authors tried to date areas of different colours within the BSE images. But I do not understand, why you have then placed analytical spots into oscillatory zoning patterns. As stated in the previous review, the division into different groups (separated by colour) of the analytical spots is not clear at all.
- c. Avoiding measurements next to cracks and holes: This issue has not been solved. I would expect that the authors have one of several arguments, based on which they know whether an analysis spot is a good spot or not (like in microprobe analysis, e.g. if the sum strongly differs from 100%). But they do not tell. It is evident from figure 5 (e.g. VANI 4, VALS) that the authors have discarded some analyses, but the arguments on which they were discarded, is not mentioned. Since the BSE images are only 2D, but some analysis spots could be bad due to 3D effects (e.g. holes not visible from the surface), discarding single spots should be a normal procedure, but based on arguments that are applied when processing the data (and not afterwards).
- d. Choice of weighted mean ages: This issue has not been addressed at all. If you take some investigated crystals such as e.g. KLEM 3 or LUCO 1, then the choice of which analytical spot has what colour and which of the spots are combined into a weighted mean age is not clear to me. Since this is the basis for the age interpretation afterwards (in particular figure 7), I do not know, whether I can trust the authors' choice. Without any further information, the calculation of these mean ages is cryptic to me. Issue not solved.
- e. Age groups: This issue is not solved either: On p. 12 (lines 12/13), you are stating that weighted mean ages are "precise ages", while "spot ages" are "approximate ages". I would agree with this classification. But the authors do not stick to it. In the Discussion chapter you mainly use the term "age", sometimes "spot age", sometimes "mean age", but in many cases, it is not clear whether we now talk about a "precise" or an "approximate" age information. The same is true for figure 4, in which a two colour classification is introduced, but seemingly without any consequence for the later interpretation.
- f. Figure 8 is now figure 4 and has not been changed. We recommended to give clear definitions for the boundaries from white to faint and faint to strong colour. The authors only provide a vague definition by stating that strong colour means peaks and plateaus, but for the blue curve I see a plateau from 7 to 10 Ma, while the authors do not. Issue not solved for me.
- g. Figure 8 and previous literature data: This point has not been taken up as far as I can see.
- h. Regional groups: There is no new information in this issue, therefore the issue cannot be closed. Furthermore, the distinction into different patterns in the description of the samples along with figure 5, does cause additional confusion. Why have the authors not stated that their division has been e.g. made initially and that this division seems to work well? It is never clear whether the division is an initial feature or a result of the study.
- i. Comparison with other thermochronometers. I definitely like figure 6. But as shown in my figure above, I would draw other conclusions from it. I consider the fact that monazites are formed over a wide temperature range as a methodical advantage, not a disadvantage. Let us backtrack for a second: Monazites in fissures have no closure temperature, the ages are formation ages, we do not date cooling or exhumation, but we date fluid activity (and in most cases then tectonic activity). As such, figure 6 has an important methodical impact, but to me not the same as for the authors.

- j. Chapter on hydrothermal monazite "crystallisation : This information has now been shifted to the intro part of the paper. Issue closed.
- 5. New information about exhumation and tectonics within the Lepontine Dome: To me this issue is only partly solved. First of all, the authors do not consider the possibility that fluid pathways do not need to be changed to precipitate monazite. Imagine e.g. a dehydrating mineral reaction in depth that releases a lot of hot fluid, which then migrates upwards along existing pathways and leads to monazite formation. No tectonic "event" is needed, but only progressive burial. In the view of the authors, all monazite formation must ultimately be linked to tectonic activity (which in many, but not necessarily all cases may be true). But: In the end (and this is the important message!), the authors present some interesting age pattern across the Lepontine dome (even if we neglect any weighted mean ages and only stick to the range of the spot ages). This is what remains to me if I subtract all doubtful clustering of the ages. And it probably results in a much more vague interpretation of the ages. Which would be ok with me, as such an interpretation would be based on sound data that nobody can criticise as I do above.

I conclude, that many issues have not been solved and the paper still lacks convincing straightforwardness. Since above, we have mainly looked backwards, I now bring up my concerns with respect to the in a shortened form. All issues rely on the many detailed comments as listed below.

As stated within my previous review, there is no doubt to me that the provided data are an interesting data set. The authors have failed though to convincingly explain on how the single spot analyses can be used to derive mean ages to derive age groups and to derive conclusions for the structural evolution of the Lepontine dome.

My major concerns on the revised version are the following:

- 1. Methodical meaning of the ages. The monazite ages are formation ages, if we assume in particular that one spot is placed on one monazite generation and there were no mixing processes afterwards. Formation is due to fluid disequilibrium, caused by PT changes, fluid changes, and fluid composition changes. Part of the changes may be caused by tectonic activity. This is my general frame. It seems that the authors differ from it, but I do not understand why. I find the methodical description (e.g. chapter 3.1) not yet very clear. In addition, the paper strongly promotes that monazite formation means tectonic activity, and such a simplification is not exclusively true. The authors miss to express their caution about this. The authors could also discuss the effect of alteration on the ages, if there is a systematic effect; but they do not.
- 2. Grain-internal "features": The authors use a large range of expressions to describe the internal patterns they see. In chapter 4.1, they present a division into 5 categories that is based on the internal structures, but this categories seem to have no influence afterwards and I do not understand your applied terminology. Methodically they present figure 3 to explain, however, with many descriptions to the BSE images in figure 5 I disagree or, at least, am not able to see the same. In the end, I wonder what the purpose of the classification into 5 feature categories is. The authors somehow have used these patterns to place their analysis spots. But this is far from being transparent to the reader.
- 3. Group formation and weighted mean age calculation: I do not understand how the different analyses were grouped (colour groups in figure 5), and how you decided on which spots are grouped forming a weighted mean age (see many questions below). To me the logical consequence is that I can only trust your single spot ages and the resulting spot age range (see statement on p. 12, lines 10-12), while the "weighted mean ages" in figure 7 are not explained in a transparent way.
- 4. Temperature range of fissure monazite formation: As shown in the figure above, I have doubts about the temperature range of 200-350°C. If we assume that fluid temperatures in fissures can substantially differ from the temperatures of the surrounding rock, then these two parameters become decoupled and the comparison with thermochronology data is relatively useless.
- 5. Estimation of exhumation rate: If the temperature range, in which the fissure monazites are formed is very large (see my figure above), any comparison with thermochronological ages to estimate exhumation rates is difficult. In your chapter 5 you refer to some examples from areas of slow exhumation, but you do not show any example for fast exhumation to show any difference. In addition, one of the assumptions in the estimation is that the temperature of the fluid in the fissures is identical or close to the temperature of the host rock. I am sure that this is not always the case (monazite formation out of disequilibrium (e.g. causing fast cooling of the fluid). Thus I have serious doubts about the conclusions of chapter 5.
- 6. Repetitive structure in Discussion and Summary chapter. Chapter 6 and 7 seem to be repetitive and I do not know why the authors have chosen such a repetitive structure.

In summary, I note that still fundamental issues have not been solved. I simply realise that the list of detailed comments below is very long again (nearly 11 pages...). The issues mainly cover methodical aspects, it is not due to a bad data set. Any interpretation of the data depends on the data treatment. For the time being I feel unable to understand the procedure the authors have applied. I have less concerns about the general interpretation of the data pattern and its evolution in time (in particular Fig. 7). But the data presentation would need a fundamental revision to clarify the procedure and by this provide a transparent base for later data interpretation. This would still require again major revisions of the manuscript. This must be disappointing for the authors who have already done a strong revision, but evidently not sufficiently focussed on the comments of this reviewer.

**Detailed comments:**

The following comments are sorted according to page numbers and text lines (in the pdf provided). They were continuously gathered while reading. Many of them have later been clustered to major comments (see above). With respect to text quality, I have found very few typos, thus the text seems to be in an advanced stage. Many text comments should be seen as suggestions (as indicated), e.g. to clarify a statement.

| Page 1      |                                                                                                                                                                                                                                                                              |  |  |  |
|-------------|------------------------------------------------------------------------------------------------------------------------------------------------------------------------------------------------------------------------------------------------------------------------------|--|--|--|
| Line 2      | Suggestion: " in the Central Alps and the timing of deformation"                                                                                                                                                                                                             |  |  |  |
| Line 3      | Suggestion: Start sentence with "Hydrothermal monazites-(Ce)"                                                                                                                                                                                                                |  |  |  |
| Line 3      | "physicochemical"                                                                                                                                                                                                                                                            |  |  |  |
| Line 4      | "through their crystallization ages"                                                                                                                                                                                                                                         |  |  |  |
| Lines 4/5   | Delete "includingtype locality," this information is not as important as to be mentioned in the abstract.                                                                                                                                                                    |  |  |  |
| Line 6      | Suggestion: "geometric"                                                                                                                                                                                                                                                      |  |  |  |
| Line 6      | It is not clear what you mean with "spatial distribution". Do you mean the spatial distribution of the analysis spots on the crystals or the spatial distribution of the samples within the Lepontine dome?                                                                  |  |  |  |
| Line 8      | The "followed" is a fuzzy term, as time wise "15-10 Ma" do not follow "16.5-10.5 Ma", but have a large overlap.                                                                                                                                                              |  |  |  |
| Line 9      | Suggestion: "along the western limit"                                                                                                                                                                                                                                        |  |  |  |
| Line 9      | Suggestion: "A youngest age group"                                                                                                                                                                                                                                           |  |  |  |
| Line 10     | Suggestion: delete "large", not necessary, perhaps in 20 years with automated analytical machines, this number will be small                                                                                                                                                 |  |  |  |
| Line 11     | Suggestion: delete "spread over significant metamorphic structures". The paper does not discuss metamorphism in the Lepontine dome (or steps in metamorphic overprint e.g. across the RSL).                                                                                  |  |  |  |
| Lines 11/12 | I do not think, the formation of monazite has got to do with stability (do you mean
thermodynamic stability?). At these relatively low temperatures, probably, the fluid
composition and kinetics are much more relevant. Thus, I would avoid the term
"stability". |  |  |  |
| Lines 12-14 | I do not believe in this estimation and I would suggest to complete delete this last sentence.                                                                                                                                                                               |  |  |  |
| Line 16     | Suggestion: Use "multi-phase" instead of "complex"                                                                                                                                                                                                                           |  |  |  |
|             |                                                                                                                                                                                                                                                                              |  |  |  |

- Lines 18/19 If "much of the retrograde evolution of the area is well known", you should be able to cite some of the relevant references.
- Line 19 In this early stage of the paper, it may be more cautions and state "monazite-(Ce) ages may be able to complement …". If this finally is true you have to prove.

- Line 1 I am not sure, whether "reactive" is the correct word. In some cases, dissolution of existing monazite may occur, which, from a monazite point of view, is not an "active" process at all...
- Lines 10/11 It would be interesting to add, why the monazite system is different and does not have such methodical flaws. Why do monazite do a better job?
- Line 18 Suggestion: "where newly formed monazite-(Ce) ..."
- Line 27 I do not think that the statement "development of several metamorphic areas" (and the given examples) is correct. What is a "metamorphic area? The Tauern Window is a tectonic window, showing higher metamorphic grade than the surrounding and covering units, but this does not mean that the window has experience a different metamorphic history, but only a different exhumation/denudation history.
- Lines 29/30 I do not know, what the statement "The Western and Central Alps ... complex tectonic and metamorphic history." should add to the content of the paper. The term "complex" is fuzzy. It simply means that there is not sufficient data to fully understand it. Is this of importance to this study?

**Page 3**

- Figure 1 ok, no comments.
- Line 2 Is the bracket "in excess of 650°C in some regions" important? The information is not very specific. It would help much more to state that the Lepontine Dome consists of medium-grade metamorphic rocks (meaning minimum Alpine metamorphic temperatures of 500°C, which of course is not true for the samples from the Gotthard massif, TAMB 1, VALS and VANI 6). One might even think about showing the 500°C metamorphic isotherm in figure 1.
- Line 10 Suggestion: "margins" instead of "limits". The border is not very sharp and not very obvious from figure 1.
- Line 12 "contemporaneously"
- Page 4
- Figure 2 One wonders about a combination of figure 1 and 2, as there is a large overlap in their content. In figure 2, the corner top right and bottom centre of the map both show and area of light grey colour, which is not defined and in the first case is simply geologically wrong. Check with the Tectonic map of Switzerland. In the figure caption you are mentioning that the different profiles have different scales among each other and compared to the map. But for each of the profiles a km-scale should be added. "Profile (b) should have SW NE instead of S-N at the endpoints.
- Figure caption Line 2: Suggestion: "across" instead of "over". In the figure caption you should add an explanation about A and A', you should refer to figure 6, in which the line A-A' is used.

**Page 5**

Table 1For sample "DUTH 2", the name of the lake is "Lago Scuro". It is not my job to check
this...

**Page 6**

Line 1 Change title to "2.2 Study area"

Line 2 Suggestion: "comprises the northern part of ..."

- Line 2 You define the study area as the part, "in which mineralized fissures/clefts are commonly found". This seems to be a strange definition, as you do not explain, why there are no mineralized fissures further south and who has made this statement/observation beforehand. The study area may simply be defined by the collected samples. If however there is a southern border that is defined geologically, it would be important to know ("stability" or field of occurrence of fissure monazite).
- Line 3 Suggestion: "across" instead of "over"
- Line 5 The statement "were divided into four groups" needs an explanation. Has this grouping been based on major tectonic boundaries? If yes, why were the Gotthard samples not placed into a separate group?
- Line 5 "to the east" instead of "to the west". Could this be more specific? One may e.g. state "Adula nappe and further east".
- Line 6 The "Verzasca Line" is not visible in any figure, so the reader cannot assess this explanation.
- Line 8 The statement "and adjacent south-western Gotthard nappe" comes as a surprise. All other groups are defined by major tectonic lines, but here, group swaps across a major tectonic boundary. Should this not be a result of your study, instead of a prerequisite? One result could be, there is no change in monazite formation ages from Lepontine Dome to Gotthard Massif.
- Line 12 Suggestion: Start paragraph with "The formation of hydrothermal ..."
- Lines 15-24 I beg to disagree with this text. Fissures are areas of disequilibrium. If there is equilibrium, nothing happens. Due of this disequilibrium, however, minerals are precipitated. Thus, I see the formation of monazite not as a sign of equilibrium. The text here is misleading and would need rewriting. I also disagree that "chemical disequilibrium is generally triggered by tectonic activity (line 17). The disequilibrium could be reached by new fluid flushing through the system, and this does not "generally" be based on tectonic activity, but may be related to e.g. a dehydrating mineral reaction further down. Finally, what is so important to state that this is a "chemical" disequilibrium? The disequilibrium may mainly come from large temperature differences between fissure and rock fluid. In the abstract you are more general, referring to "physicochemical" conditions that change.
- Line 26 Avoid the term "reactive (see comment p. 2, line 1).
- Line 28 Suggestion: "may cause the crystal …". It is not clear at all that precipitation (and in particular precipitation of monazite) occurs. Thus I would make a more cautious statement. Note the implication of your statements here. Monazite within the fissures MAY record tectonic events that can be dated. But monazite formation is not guaranteed at all, and important tectonic events MAY be missed, and also other fluid forming events MAY also cause monazite formation. The reduced formula "monazite formation = tectonic event" is simply wrong.
- Page 7
- Figure 3 I find this figure difficult. If forms the basis for the interpretation of all the other BSE images inn figure 5. For such purpose, the three samples do a rather moderate job. Image B is difficult to interpret, as the colour differences are weak and disturbed by the combination of the different photos that have different colours and colour contrasts. I cannot follow the description of this image. In addition, here starts the mess with the different terms such as zonation, sector zonation sector-like zonation etc. If these descriptions and classifications are important, you have to more carefully describe what you mean by these terms.
- Figure caption Line 1: "BSE images …". Line 3: I would not use the word "pristine", as it implicates that you know that this is pristine. How do you know? Lines 3/4: Suggestion: "(B) Partially.preserved sector-like zonation…". Lines 6/7: Suggestion: "(C)" Multiple rims combined with …". Line 8. Suggestion: "core" instead of "central part". Line 9: Is "inner rim" equal to "central part"/core?

- Lines 5/6 Here, you mention a temperature window and stability field for fissure monazite. My question is: Does it matter, if such a temperature window (similar to a closure temperature) is defined. Would it not be more favourable to the method to leave it open, in particular, if you assume that there might be a temperature disequilibrium between fissure fluid and surrounding rock wall? I find it rather contradictory, if you speak about a "temperature window and stability field" first and then about the monazite formation to be largely "temperature independent".
- Line 8 Suggestion: "catalysed" or "triggered" instead of "aided"
- Line 9 Whatever means "primary or secondary" is not defined in your discussion about the BSE images or data. You only talk about formation and alteration, which may not be the same.

- Line 3 You refer to figure 3c, but strictly speaking you should state "as assumed for the crystal in figure 3c".
- Line 6 "possess"
- Lines 9/10 Suggestion: Most of the samples were provided by mineral collectors (see Table 1 for location details", as monazite-(Ce) is uncommon in clefts and often ..." I would avoid the term "hydrothermal cleft", these clefts have formed due to tectonic movements in a brittle rock regime, but not originated from hydrothermal activity. Furthermore: I note that there seems to be no information about the orientation of the fissures that these samples have been taken from. Thus, any arguments from such fissure orientation has to remain speculative, as this observation has not been collected (or at least not systematically).
- Line 11 Suggestion: "level of a central section across the grain"
- Line 12 Suggestion: "were obtained, using ..."
- Line 15 Suggestion: Start sentence with "Spot measurements next to cracks ...". If your spot analysis includes a crack or inclusion, how do you know? What would be the methodical procedure to sort out analyses with cross contamination with a volume that is not a monazite? I just want to make it clear. If there are spot analyses that contain cracks or inclusions they should be sorted out before calculating ages, as these ages may have no geologic meaning.
- Lines 22-29 I am not a U/Th/Pb specialist, and I do not understand your careful explanation on how common Pb was corrected. Similar to me, however, I expect all non Pb-specialists to get to know what the issue about the common Pb is. Later on (p. 13) you mention that some grain analyses have high to very high common Pb values, and the reader wants to know whether this means anything with respect to precision or reliability of the ages. Thus your detailed explanation here is good, but what is the consequence with respect of the ages afterwards?
- Line 31 "possess"
- Page 9
- Table 2no comments

- Line 4 Suggestion: "from comparison"
- Line 6 add space before "to capture"

- Lines 7/8 Suggestion: Delete sentence "The number of domains ...". This is not a scientific issue, but an organisational, and I am sure that Martin Whitehouse is not amused to be critized...
- Line 9 If the high Th/U ratios are the argument, then it does not matter whether this high ratio results from low U contents. I would therefore delete "at low U content"
- Line 10 "of" instead of "oft"
- Line 15 Suggestion: "showing the compositional ..."
- Line 19 "as these could be shown to date tectonic activity": I have not gone through the references papers, but I would simply be more cautious about the relationship between tectonic activity and monazite formation. How about: "as it has been shown that monazite formation is closely related to tectonic activity"?
- Line 29 I would not speak about "a problem" but address the issue directly: "Large parts of the Lepontine Dome …"
- Line 30 Why "more than two"? Suggestion: "several", this leaves the number more open.
- Line 31 Suggestion: "a reason for large age scatter"
- Lines 31/32 Further above you state correctly that the monazite ages are formation ages. Here you talk about "age resetting", which is a term coming from the closure temperature concept. If alteration changes the age (and we do not know how), the age is geologically meaningless.
- Line 35 The statement here differs from the statement on p. 6, lines 23/24. I consider this statement much better. This is what actually happens: precipitation of monazite due to disequilibrium and the system trying to re-gain equilibrium.

Figure 4 I still do not understand the arguments on the basis of which the different colour bars are chosen on the basis of the probability density plots. I understand that you apply a threshold value at 1 age (which cuts out the youngest bumps for the red, green and blue curves). But then: For me, the blue curve has a "plateau" at 7 to 10 Ma in my eyes, but obviously not in your terminology. This needs more clarity in your figure caption (lines 3 and 4).

**Page 12**

- Line 4 the statement "especially in larger grains" cannot be assessed. How/where do I see this and why is this the case.
- Line 5 If you state that "the entire dataset of each region was plotted according to ...", this might also be interpreted in such a way that you simply smear out geologically meaningless ages. The choice what is a good analysis (providing a good age) and what is not a good analysis (providing a useless or doubtful age) has to be made before aggregating the ages.
- Line 6 add comma, after "Fig. 4"
- Line 7 What means "a significant number of ages"?
- Lines 10-12 I like this statement, but of course such a statement only works, if only reliable analyses were used. If everything fails and all weighted mean ages are artificial clusters, then this is a statement that would still be true.
- Lines 12/13 The distinction is only proposed here. But the proposal has not been put into action, if one e.g. compares with p. 20, lines 22-24. Thus you do not distinguish between "precise" and "approximate ages".
- Line 17 Suggestion: delete "covering the time"

- Lines 19/20 Something seems to be missing here. I do not understand this sentence.
- Line 22 Is a low Th content an argument to discard an analysis? This is one of the critical points: How do you discard an analysis (see e.g. figures 5l, 5t). You have to specify this somewhere. Otherwise, the reader thinks that the quoted ages are the only "reasonable ages" (leaving it to the authors to decide what is "reasonable"...)
- Lines 25-27 Suggestion: Cut long sentence in two: Line 25: Start with "Th contents are generally..." and cut at line 27 after "ppm)", then restart "This results ..."
- Line 27 Can you be more specific: "up to several hundred, e.g. by quoting the highest value?
- Lines 29/30 Suggestion: Delete "above 70%" and shorten to "contents up to a maximum ..."
- Line 30 "With the exception of sample BLAS 1": P. 13, line 1 states that GRAESER 3 also shows "no clear sign of alteration"...
- Line 32 Start new line with "(1) Sector like zonation". For the now coming list of all samples, divided into 5 categories, I have four general comments: (1) I wonder, whether this list would better to be placed in a table. At the moment this is a long list that the reader cannot really assess. (2) I would disagree with many of your descriptions or simply cannot see what you mean. And this has got to do with (3): Many if the terms you use, such as zonation, sector zonation, sector like zonation, oscillatory zonation, oscillatorycomplex zonation, ring zonation, complex zonation, alteration features, diffuse pattern are not clear to me. Figure 3 does only partly help the reader to understand. Finally (4): Does it all matter? Does this grouping into 5 categories help anything? The reader simply wants to know: Do I understand how the authors have grouped their single spot analyses and derived weighted mean ages from. The five categories do not help in this respect
- Line 33 Here it starts: What are "some signs of alteration"? Where? What am I supposed to see? What do you mean by "complex zonation"? Which is the "inner part of the grain"? What is the difference between "primary" and "secondary"?
- Page 13

| Line 1 | What would b | be "clear s | ians of | alteration"? |
|--------|--------------|-------------|---------|--------------|
|        |              |             |         |              |

- Line 2 What is the consequence of the high common Pb contents to the age?
- Line 5 What are these "strong signs of alteration"? Where is "in places"?
- Line 7 Suggestion: "... are relatively low (1600-10800 ppm)." Comma after "elevated".
- Line 8 delete "that"? What do you mean by "strong alteration features"?
- Lines 9/10 What is the effect of this "common Pb" to the ages and its credibility?
- Line 10 comma after "elevated"
- Line 11 Text should be shifted to the left.
- Line 12 BLAS 1 is not the only grain that has the feature of young ages in the core. This is an issue for several grains. However, it is not discussed anywhere. It must be discussed, otherwise the reader will think that your method does not work.
- Line 13 What means "minor signs of alteration"?
- Line 14 What means "shows signs of alteration"? Which ones? I do not understand or see what you mean by "the zonation is diffuse in places like the centre and part of the rim".
- Line 15 I do not understand what you mean by "primary zonation is cut in places". What means primary zonation? Are there examples of "secondary zonation"?
- Line 16 Text should be shifted to the left. What are "altered zones" and how do I detect them?

- Line 17 What means "featureless"? I see many features... and you state that they are altered areas. How can you see alteration, if there are no features?
- Line 18 What means "oscillatory-complex zonation"? To me, I would expect that zoning is either oscillatory or complex. In addition I see no oscillation in figure 5a.
- Line 19 What "feature" are we talking about? I do not understand what you mean by "secondary zonation".
- Line 20 Text should be shifted to the left. What "alteration features" do you mean? If you talk about "features", does this always refer to alteration?
- Line 22 What do you mean by "diffuse pattern"? I see only areas of homogeneous colour.
- Line 23 Not clear what you mean by "sector-like" and by "complex zonation". These terms have not been defined elsewhere.
- Line 24 Text should be shifted to the left.
- Line 25 The description says nothing about the alteration. Where do I see alteration and is this alteration interpreted to fall into this category?
- Line 26 What do you mean by "diffuse pattern"? Does this concern zonation or alteration? I do not understand, why DUTH 3 is in category 4.

Line 27 For this sample you state "strong zonation in the altered parts". What kind of alteration do you mean? Up to here, the reader has the idea that zonation is something that has got to do with monazite growth (i.e. it should be a primary feature), but here zonation seems to form out of alteration...

- Line 29 It should be noted that this sample is not found in figure 5, but in the appendix (see table 2). What do you mean by "remnants of sector zonation"?
- Line 31 What do you mean by "diffuse internal structure"? If diffuse, how do you know that this sample has to be put into the group of "weak zonation with strong alteration features"?
- Line 32 Suggestion: "This may lead to ...
- Line 32 Shouldn't it be "<3300 ppm"? For this grain, the authors do not describe the internal features, but only discuss the

**Pages 14**

- Figure 5 In the following, I want to illustrate why I have so strong difficulties in understanding the authors' choice of colour grouping among the spot analyses and derivation of weighted mean ages. We need clear rules on how the distinctions were made and how then the spots for the weighted mean ages were chosen. Since the authors do not (as required in the first review) provide any clue, how the grouping and weighted mean age calculation was done methodically, I tried to find any hints to understand. In the following I formulate many questions to illustrate why I do not understand your method. Graphically: The mean age boxes are too small to read.
- Figure 5a Why are the 7 orange spots not grouped to another weighted mean age?
- Figure 5b Why are the yellow spot analyses not in an ascending order (right diagram)?
- Figure 5c How do you separate chemically between yellow, green and blue sports (could e.g. spot 12 be blue or 13 be yellow)?
- Figure 5d This sample and your choice of grouping are transparent!

Figure 5e Why are the red spot analyses not listed in ascending order (right diagram)?

- Figure 5f Why is the spot with the highest U content green? Why has spot 29 not a blue colour?
- Figure 5g This sample and your choice of grouping are absolutely clear!
- Figure 5h Why has spot 20 been discarded? Why did you not calculate a weighted mean age for the 6 yellow spots? Why no mean age for the four red spots?

- Figure 5i Why are spots 3 to 5 not blue (in particular 5)? Why are these three excluded from the weighted mean age for the yellow samples? If they are excluded, shouldn't they have a different colour?
- Figure 5j Why is spot 15 not blue?
- Figure 5k Why is spot 3 not blue? Why is spot 14 not included in the weighted mean age calculation of the yellow spots?
- Figure 5I Why are the spot analyses 3, 14 and 16 discarded from further age calculation? Why is there no weighted mean age for the orange spots?

**Page 17**

- Figure 5m This figure has a mess between the BSE image and the diagram to the right. Spot 6 is red in the image, but yellow in the diagram. Spot 7 is blue in the image, but red in the diagram. Spot 10 is yellow in the image, but blue in the diagram. Spots 19 and 20 are blue in the image, but red in the diagram...
- Figure 5n The label for spot 2 is shifted to the left in the diagram to the right. The here quoted weighted mean age was not found in figure 7
- Figure 50 Why is spot 5 red and not blue? Why do you not calculate a weighted mean age for the blue spots?
- Figure 5p Why is spot 8 not yellow? Why are the red spots not in ascending order in the diagram to the right? Why are the spots 9 and 10 not a separate group?
- Page 18
- Figure 5q How do you explain, why the yellow spots (distributed all over the crystal) form one group and e.g. spot 6 does not belong to this group? Why are the red spots not in ascending order in the diagram to the right? Why haven't you calculated a weighted mean age for the orange group?
- Figure 5r Why is spot 13 not yellow? Why is spot 19 not yellow? Why is spot 20 not yellow? Based on what argument do your define the green group?
- Figure 5s Based on what do you separate the blue spots? Why are the spots 20 and 24 not yellow?
- Figure 5t Why don't you define a weighted mean age for the orange group? Why not for the yellow group as well? Why are the spots 29 and 32 not red?

Figure 6 Nice figure!

Figure captions Lines 3/4: Suggestion: "Note that rock-forming monazites give considerably higher Tmax than..."

- Line 2 "occur typically in a temperature window of…": My figure above shows that this is not true (and this is also visible in figure 6), if you really want to compare with thermochronology data. I see the fissure temperature relatively separate from the temperature in the surrounding rock. One part of the "disequilibrium" that occurs in fissures is the fact that the surrounding rock has a lower temperature and the fissure fluid will be cooled quickly and therefore precipitate minerals out of the fluid.
- Line 3 "independent of the local cooling rate": Interestingly, this statement comes after quoting references. So what is the reference to this statement?
- Line 4 Suggestion: "are slightly older or equal to the ZFT ages"
- Lines 6/7 Delete "based on Steck et al. (2013)", as this reference is already quoted in the figure caption.
- Line 7 No brackets needed for "Fig. 6"
- Line 8 "starts to crystallize" is wrong for the rock-forming monazite, as the latter are crystallized much earlier, but their isotopic system closes earlier than the fissure monazite.
- Line 11 I disagree. The authors assume that the fluid temperature in the fissure is always in equilibrium with the temperature of the surrounding host rock. This definitely is not the case.

- Lines 1/2 I disagree. If you look at figure 5, some fissure monazite ages are even in the range of AFT ages.
- Line 2 The "(1)" is not necessary (also delete "(2)" in line 5 and "(3)" in line 10.
- Line 3 space before "2011"
- Lines 8/9 Perhaps one may here refer to figure 6, where the jump in cooling ages is visible. I agree with the authors that the jump get much smaller toward the south, but here data are two variable to definitely state that there is no jump.
- Line 9 Suggestion: "across" instead of "along"
- Line 14 "exhumation/cooling rates were low": What means low? Can you quantify? The Alps exhume low with respect to Taiwan or Scandinavia, but there are still faster areas than others...
- Line 15 I suggest to shift the references to the end of the sentence.
- Lines 16-20 What I am missing is a counter example. Stating that this is true for areas of slow exhumation should be proven by a case, where it is different and exhumation is fast. The absence of evidence is not the evidence of absence...

I turn your argument around: If the fissure monazite ages are close to the K/Ar ages of white micas, and we assume that fissure formation only occurs in a brittle regime (i.e. below 300 °C, where quartz becomes brittle, then having similar age this would rather mean fast cooling from white mica closure (at 350°C) to fissure formation (not earlier than at 300°C).

- Line 24 Do we talk about a spot age or a weighted mean age? From here on, this should carefully be distinguished (as stated on p. 12, lines 12/13).
- Lines 27/28 Again: Do we talk about spot ages or weighted mean ages?
- Line 31 Suggestion: "age record with the oldest ages being diachronous ..."
- Line 33 Delete "After this," and start sentence with "The age record...

- Line 5 Be more precise: "... at the SW edge of the ..."
- Line 5 Suggestion: "separated from the Western region
- Line 6 Suggestion: "shows an age range similar to the eastern region". Why is it here "eastern", but one line above "Western"? Upper case or lower case?
- Lines 8-10 I am missing here the age jump across the RSF, furthermore the possibility of an age offset across the Forcola fault. Your figure 6 also suggests a "jump in cooling ages" across the Forcola Fault, not only the RSF.
- Lines 11-33 From here on, all the regions are discussed again, and it is not obvious why. This part seems to be repetitive and I would suggest to combine it editorially with the former paragraphs. First discuss the single regions, then discuss the general trends.
- Line 16 "Forcola" instead of "Forcoloa"
- Line 17 Rather than only quoting the method "AFT", you should also quote the age.
- Line 18 "late circulation of hot fluids": Here the possibility of hot fluids circulating is mentioned, but this is all the time the case. Fissures fill because hot fluid enters the fissures and get cooled and this causes precipitation. Why is this only mentioned here? Simply because this far outside of the temperature range quoted e.g. on page 20, line 17 (and twice before)?
- Line 20 Suggestion: "and then runs ..."
- Line 20 What does "runs parallel to that of ..." mean? "Parallel" with respect to what? If you mean "cooling", then state that "it cools with the same rate than..."
- Lines 22/23 Suggestion: "After this time, temperature conditions must have decreased due to exhumation ..."
- Line 23 "around 16-15 in the Gotthard nappe". Can you add a reference? What ages do you refer to?
- Line 25 You are referring to ZHe ages of Price et al., but it is not clear what area you are referring to. Price et al. have not produced ZHe ages from the Lepontine Dome nor from the Gotthard massif (northern Adula nappe is not within the Lepontine Dome).
- Line 26 I am not sure how the Price et al. data can date the "decoupling of the Gotthard nappe". Your "regions" are combining the northern Lepontine Dome with the southern Gotthard nappe (Figure 7, thus they are treated as being "coupled"). If in the end, your data sets show a difference between Lepontine Dome and the Gotthard nappe fissure ages, then you should show this in figure 7 and revise your data regions (which, as a result out of your dating, would be ok).
- Line 28 "until" instead of "to"
- Lines 28/29 "as the samples ... 9 Ma (Fig. 7d)": I do not see the link. If it was fast, why should it show continued age spread? Needs clarification.
- Lines 30/31 Suggestion: Start sentence with "In some areas it records a change from..."
- Lines 33/34 In this statement you argue with the direction of the opened fissures/clefts. However, you do not provide any information about the fissure/cleft direction of your samples (see e.g. tables 1 and 2). Thus, this discussion looks odd to the reader: Why should we discuss something, we have no information on? In addition, I got lost in these two lines. Is there something missing? Perhaps you have to simply the long sentences.
- Page 22
- Figure 7 Figure has been much improved. I found two errors: Sample TAMB 1has two ages in figure b, but the younger age is erroneously again quoted in figure c. In addition, the mean age of BLAS 1 is missing in figure c.

Figure captions.. Line 1: What means "age"? Spot age or mean age? Line 4: Suggestion: "that could be identified within the corresponding grain. Ages are ordered into six time intervals." Otherwise, one may misinterpret that you have ordered your spot ages according to age and then calculated weighted mean ages...

- Line 2 The reference should be "Figs. 7e-f".
- Lines3-5 I note that the different ages are quoted with different precision: 11 Ma (based on three spot ages of 12, 12 and 10.5 Ma, should you therefore rather quote an age range?), 16.8 Ma and 11-10 Ma. In addition, the 11-10 Ma is based on spot ages that range up to 11.5 Ma (Fig. 5d). Be more careful with the numbers you are quoting, and be evenly precise when you quote them.
- Line 8 Can you be more precise with the reference? Which part of figure 5? Which sample?
- Lines 10-24 Again, there is a repetitive paragraph that again runs through the same results. There is basically nothing new that comes in this paragraph.
- Lines 10/11 I do not know what the statement "is related to the exhumed deeper part of the Simplon Fault" means. What is the "deeper part of the Simplon Fault"?
- Line 11 The reference "Hartel and Herwegh 2012" is missing in the reference list.
- Line 12 If you talk about a "switch in deformation style", then you should also state from which to which state the system switched.
- Line 19 "where strong late-stage hydrothermal activity occurred": What means "strong"? Can you add a reference to this statement?
- Line 21 "indicate a resetting ...". Do you mean a local resetting? This would be in line with hydrothermal activity...
- Lines 21/22 "These phases of strike-slip deformation": This comes with surprise that you talk about a direction of fault movement. Where does this information come from? Is the direction of motion important to this study? Why is hydrothermal activity related to "strike-slip"?
- Line 28 "regional" instead of "regionsal"
- Line 31 "record" instead of "records"
- Line 32 The here again quoted (and to my taste wrong) temperature range is not a result of this paper. You show yourself in figure 6 that the range is larger and discuss it on p. 21, lines 17-19.

Lines 1-6 Here again, similar to the Discussion chapter: This part is repetitive to previous statements in the summary. This part could be shortened.

I have not checked the appendices and the references, as the major revision of the paper will also imply serious changes in these parts.

Meinert Rahn, August 15, 2019

---

## Referee Report (RR2)

Review of Bergemann, Gnos, Berger, Janots and Whitehouse, Solid Earth Ms. Dating tectonic activity in the Lepontine Dome and Rhone-Simplon Fault regions through hydrothermal monazite-(Ce). - version 4, 28 October 2019.
Reviewer: Fraukje Brouwer, VU Amsterdam, Netherlands, 18 December 2019

Note: This review concerns version 4 of the manuscript that prepared in response to the review of Dr. M. Rahn of version 3, and focuses mainly on whether his comments were addressed satisfactorily. The response accompanying the revised version includes many questions throughout to Dr. Rahn. In some cases, the related comments were not addressed and the editor might want to check whether he feels further communication between the authors and Dr Rahn is necessary to address those points and could further improve the manuscript.

**General assessment**
In this version, the authors do a better job of presenting their data in a way that is intelligible to the reader and the manuscript has benefitted markedly from the revisions directed by the earlier reviews. I am disappointed by the quality of the English of the manuscript and recommend a thorough round of editing to improve its presentation. I started making specific comments for the Abstract but stopped after that, because it would take a considerable amount of time and I do not want to further delay my input.
Below, I list comments specifically related to the latest review of Dr. Rahn and the authors' response, Two additional, if related, comments of my own and finally a couple of pages of technical comments, which I emphasise, are far from exhaustive.
because of the interesting and extensive dataset, as well as the argumentation for its interpretation, I feel this manuscript is certainly worthy of further consideration for publication in Solid Earth, and I sincerely hope that with a final round of revisions, the authors address all remaining concerns and render the manuscript publishable. **I recommend that it undergoes minor revision before being accepted for publication in Solid Earth**.

**Specific comments referring to the response accompanying v4 - remaining issues**
4a   The first part of the response asks further clarification and perhaps use of the figures from the reviewer. This is not a discussion I can or want to weigh in on and I suggest the editor decides what he feels is appropriate.
4i   The response reflects confusion as the authors do not understand the reviewer's comments. I agree that they appear to have addressed the issue at hand under 4a.
5   The authors state that fluid flow requires an 'event' and infer that such an event is most likely deformation. Dr. Rahn does not agree, and I feel that he is at least partly correct in his arguments. The authors make a strong case in the response that deformation is likely to play an important role, and this argument is now made more clearly in the manuscript (3.1) than in previous versions. In my view, two significant problems persist:
   **a)** the authors fail to consider any alternative causes for fluid flow that might result in Mz precipitation, and
   **b)** the position of this section in the manuscript, at the beginning of the methods section, is very strange. Section 3.1 in my view would be better placed in the introduction, because it presents the arguments supporting the basic premise of the study (all cleft Mz crystalisation is triggered by deformation-induced fluid flow). In the present form this remains an implicit assumption, rather than a well-argued premise. Alternatively, a sentence in the introduction could foreshadow this argument, and then 3.1 should be placed in the discussion of the manuscript.

10 There is quite a fundamental difference in opinion here between the authors and Dr. Rahn. I feel that the text has been changed enough that the reader may form their own opinion on this matter and that the manuscript does not suggest this interpretation is fact.

11 By their very nature the discussion and summary (or conclusions) must be repetitive, which means I disagree with Dr. Rahn on this point. I feel the present structure is clear and do not mind the repetition. It might be clearer yet to combine 3.1, 5 and 6 into a single discussion (section 5), with three or more subsections.

Page 3, fig. 1: adding the 500°C isotherm is an excellent suggestion and make the location and shape of the Lepontine dome much clearer to the reader less familiar with the study area. I feel the authors should adopt this suggestion of Dr. Rahn.

**Additional comments on v4.**

- (Lines 12-13) 'The data-set shows that the fissure mineral crystal-rock interaction, fluid flow and resulting monazite-(Ce) age record are directly linked to the Lepontine Dome's evolution in space and time.' There is still a risk of circular argument here: if all ages are assumed to reflect deformation events, this is an unavoidable consequence.

- Related to point 5, above. The implicit assumption of the abstract and introduction is that all fluid activity in veins that resulted in Mz growth was deformation induced. This is a basic premise of the paper and should be argued explicitly and convincingly in the Introduction. I don't necessarily disagree with the assumptions, but find the paper less convincing than it could be because the interpretations rely fully on this basic principle. In the end, this issue is the source of much of the criticism thus far.

**Technical corrections**

The numbers below refer to line numbers in v4 of the Ms.

(5) Replace 'and' by 'to'.

(6) 'this' should be 'these', as data is always plural.

(7) 'In' should be 'at' or 'near', as an edge is a 2D, rather than 3D feature.

(8-9) '…started in the eastern Lepontine Dome later at 15-10 Ma.' should be '…in the eastern Lepontine Dome started later, at 15-10 Ma.'

(9) To my understanding mineral names should in almost all cases be used in singular, rather than plural (one wouldn't write 'quartzes', where it now says 'monazites'). The exception is when the word refers to different forms (e.g. solid solutions) of a mineral (garnets could be pyrope and almandine, as opposed to multiple grains). Therefore, in line 9 'Fissure monazites-(Ce) are younger…' should be replaced by 'Fissure monazite-(Ce) is younger…'.

(10) 'A youngest…' should be 'The youngest…'. This statement is a bit confusing given the earlier statements that ages range to 2.7 Ma.

(11) 'data set', instead of 'data-set'.

(12) 'fissure mineral crystal-rock interaction' - it is not clear to me what the authors are trying to say. This needs to be rephrased again.

(13) 'and' is missing: monazite-(Ce) AND thermo-chronometric data

As indicated above, I have stopped suggesting text edits after the abstract. Below are a few points that are simply wrong or unclear, but many more improvements can and should be made to the text throughout.

(13) 'Compared to this…' should be 'In contrast, …'

Table 1 - It is unclear what the significance is of the two reference numbers in the table header. Three localities are not all aligned to the column (e.g., DURO; should this be DORU, like the locality itself)

(3-4)'…Fault in the west/southwest south of the Centovalli Fault,…' - confusing use of directions. Please rewrite more clearly.
(22-23) A verb is lacking in the part '…meaning that….deformation of the system.'

(2-4)There's an overload of parentheses in this section, which do not make the text clearer. I'd suggest '…fractures in monazite-(Ce) crystallized during the initial formation of the grain (primary Mz) and monazite-(Ce) formed at a later time or recrystallized/reprecipitated (secondary), induced by…'. Note that there is a typo in initial.
(9)   ()Grand0Homme et al., 2018). Correct parentheses.

(21) The use of phases for deformation is confusing. It would be better use 'deformation phases', 'stages', instead, to avoid confusion with phases in the thermodynamic sense.

(25) …younger than or equal to…
(32) …age record whereas ZFT ages…

(16) delete first comma
(24) … window or if the analyzed…

(23) '…may indicate a localized resetting…' - addition needed to address Dr. Rahn's comment.

---

## Author Response (AR3)

Dear editor, dear reviewers,

All technical comments were implemented in the manuscript. The issue of deformation or fluid flow as the reason for monazite (re)crystallization was the main point Dr. Brouwer felt had not been addressed in sufficient detail. The part explaining our reasoning why the analyzed grains date deformation now provides more details, was moved and is now included in the introduction.
We would like to express our gratitude to the reviewers for their thorough work that greatly improved the quality of the manuscript.

Yours sincerely,
Christian Bergemann

Review of Bergemann, Gnos, Berger, Janots and Whitehouse, Solid Earth Ms. Dating tectonic activity in the Lepontine Dome and Rhone-Simplon Fault regions through hydrothermal monazite-(Ce). - version 4, 28 October 2019.

Reviewer: Fraukje Brouwer, VU Amsterdam, Netherlands, 18 December 2019

Note: This review concerns version 4 of the manuscript that prepared in response to the review of Dr. M. Rahn of version 3, and focuses mainly on whether his comments were addressed satisfactorily. The response accompanying the revised version includes many questions throughout to Dr. Rahn. In some cases, the related comments were not addressed and the editor might want to check whether he feels further communication between the authors and Dr Rahn is necessary to address those points and could further improve the manuscript.

**General assessment**

In this version, the authors do a better job of presenting their data in a way that is intelligible to the reader and the manuscript has benefitted markedly from the revisions directed by the earlier reviews. I am disappointed by the quality of the English of the manuscript and recommend a thorough round of editing to improve its presentation. I started making specific comments for the Abstract but stopped after that, because it would take a considerable amount of time and I do not want to further delay my input.

The English was improved as much as possible by ourselves and given to another colleague for proof-reading (although also not a native speaker). We hope that upon reading the manuscript you will find the level of English acceptable.

Below, I list comments specifically related to the latest review of Dr. Rahn and the authors' response, Two additional, if related, comments of my own and finally a couple of pages of technical comments, which I emphasise, are far from exhaustive.
because of the interesting and extensive dataset, as well as the argumentation for its interpretation, I feel this manuscript is certainly worthy of further consideration for publication in Solid Earth, and I sincerely hope that with a final round of revisions, the authors address all remaining concerns and render the manuscript publishable. **I recommend that it undergoes minor revision before being accepted for publication in Solid Earth**.

**Specific comments referring to the response accompanying v4 - remaining issues**

4a The first part of the response asks further clarification and perhaps use of the figures from the reviewer. This is not a discussion I can or want to weigh in on and I suggest the editor decides what he feels is appropriate.

4i The response reflects confusion as the authors do not understand the reviewer's comments. I agree that they appear to have addressed the issue at hand under 4a.

5 The authors state that fluid flow requires an 'event' and infer that such an event is most likely deformation. Dr. Rahn does not agree, and I feel that he is at least partly correct in his arguments. The authors make a strong case in the response that deformation is likely to play an important role, and this argument is now made more clearly in the manuscript (3.1) than in previous versions. In my view, two significant problems persist:

**a)** the authors fail to consider any alternative causes for fluid flow that might result in Mz precipitation, and

**b)** the position of this section in the manuscript, at the beginning of the methods section, is very strange. Section 3.1 in my view would be better placed in the introduction, because it presents the arguments supporting the basic premise of the study (all cleft Mz crystalisation is triggered by deformation-induced fluid flow). In the present form this remains an implicit assumption, rather than a well-argued premise. Alternatively, a sentence in the introduction could foreshadow this argument, and then 3.1 should be placed in the discussion of the manuscript.

Additional comments on v4.

(Lines 12-13) 'The data-set shows that the fissure mineral crystal-rock interaction, fluid flow and resulting monazite-(Ce) age record are directly linked to the Lepontine Dome's evolution in space and time.' There is still a risk of circular argument here: if all ages are assumed to reflect deformation events, this is an unavoidable consequence.

Related to point 5, above. The implicit assumption of the abstract and introduction is that all fluid activity in veins that resulted in Mz growth was deformation induced. This is a basic premise of the paper and should be argued explicitly and convincingly in the Introduction. I don't necessarily disagree with the assumptions, but find the paper less convincing than it could be because the interpretations rely fully on this basic principle. In the end, this issue is the source of much of the criticism thus far.

As proposed by the reviewer, chapter 3.1 was moved to the introduction and expanded to discuss the issue of deformation vs other causes for fluid activity. The introduction was expanded to better explain that fluid flow is of little impact for most of the (re)crystallization history of hydrothermal monazite. Since generally only small fluid volumes that tend to stay within the cleft/fissure are involved.

10 There is quite a fundamental difference in opinion here between the authors and Dr. Rahn. I feel that the text has been changed enough that the reader may form their own opinion on this matter and that the manuscript does not suggest this interpretation is fact.

11 By their very nature the discussion and summary (or conclusions) must be repetitive, which means I disagree with Dr. Rahn on this point. I feel the present structure is clear and do not mind the repetition. It might be clearer yet to combine 3.1, 5 and 6 into a single discussion (section 5), with three or more subsections.

As mentioned above, section 3.1 was expanded and moved to the introduction. Chapters 5 and 6 were kept separate, but are now subsections of the discussion chapter 5.

Page 3, fig. 1: adding the 500°C isotherm is an excellent suggestion and make the location and shape of the Lepontine dome much clearer to the reader less familiar with the study area. I feel the authors should adopt this suggestion of Dr. Rahn.

The 500 °C isotherm available for the area has been added to the map. In lack of this for the south-western part of the study region, the albit-oligoclase mineral zone boundary was used.

**Technical corrections**

The numbers below refer to line numbers in v4 of the Ms.

(5) Replace 'and' by 'to'.

Changed

(6) 'this' should be 'these', as data is always plural.

Changed

(7) 'In' should be 'at' or 'near', as an edge is a 2D, rather than 3D feature.

Changed to "at"

(8-9) '…started in the eastern Lepontine Dome later at 15-10 Ma.' should be '…in the eastern Lepontine Dome started later, at 15-10 Ma.'

Changed

(9) To my understanding mineral names should in almost all cases be used in singular, rather than plural (one wouldn't write 'quartzes', where it now says 'monazites'). The exception is when the word refers to different forms (e.g. solid solutions) of a mineral (garnets could be pyrope and almandine, as opposed to multiple grains). Therefore, in line 9 'Fissure monazites-(Ce) are younger…' should be replaced by 'Fissure monazite-(Ce) is younger…'.

Changed

(10) 'A youngest…' should be 'The youngest…'. This statement is a bit confusing given the earlier statements that ages range to 2.7 Ma.

Changed to "A younger…"

(11) 'data set', instead of 'data-set'.

Changed

(12) 'fissure mineral crystal-rock interaction' - it is not clear to me what the authors are trying to say. This needs to be rephrased again.

Changed to "interaction between fissure mineral and host rock"

(13) 'and' is missing: monazite-(Ce) AND thermo-chronometric data

Changed

As indicated above, I have stopped suggesting text edits after the abstract. Below are a few points that are simply wrong or unclear, but many more improvements can and should be made to the text throughout.

(13) 'Compared to this…' should be 'In contrast, …'

Changed

Table 1 - It is unclear what the significance is of the two reference numbers in the table header. Three localities are not all aligned to the column (e.g., DURO; should this be DORU, like the locality itself)

The reference numbers were moved to the the analytical techniques section in the form of the sentence "Sample GRAESER 1 was provided by the Natural History Museum of Basel (identification number NMBa 10226) and VALS was provided by the Natural History Museum of Bern (identification number NMBE43124).". The issue concerning the locality names in the table was solved.

(3-4) '…Fault in the west/southwest south of the Centovalli Fault,…' - confusing use of directions.

Please rewrite more clearly.

Changed to "…across the central Lepontine Dome to the Simplon Fault in the west/southwest, to south of the Simplon Fault, …"

(22-23) A verb is lacking in the part '…meaning that…deformation of the system.'

Sentence changed to "The fissure/cleft remains fluid filled and behaves for considerable parts of its history as a closed system, meaning that during deformation of the system, repeatedly recycled small volumes of fluid suffice for the (re)precipitation of large mineral volumes (Sharp et al., 2005)."

(2-4) There's an overload of parentheses in this section, which do not make the text clearer. I'd suggest '…fractures in monazite-(Ce) crystallized during the initial formation of the grain (primary Mz) and monazite-(Ce) formed at a later time or recrystallized/reprecipitated (secondary), induced by…'. Note that there is a typo in initial.

Changed

(9) ()Grand0Homme et al., 2018). Correct parentheses.

Changed

(21) The use of phases for deformation is confusing. It would be better use 'deformation phases', 'stages', instead, to avoid confusion with phases in the thermodynamic sense.

Changed

(25) …younger than or equal to…

Changed

(32) …age record whereas ZFT ages…

(16) delete first comma

Changed

(24) … window or if the analyzed…

Changed

(23) '…may indicate a localized resetting…' - addition needed to address Dr. Rahn's comment.

Changed